# How to Fix a Broken Confidence Estimator: Evaluating Post-hoc Methods for Selective Classification with Deep Neural Networks

**Luís Felipe P. Cattelan**[1]        **Danilo Silva**[1]

[1]Department of Electrical and Electronic Engineering,
Federal University of Santa Catarina (UFSC), Florianópolis, Brazil ,
lfp.cattelan@gmail.com, danilo.silva@ufsc.br

## Abstract

This paper addresses the problem of selective classification for deep neural networks, where a model is allowed to abstain from low-confidence predictions to avoid potential errors. We focus on so-called post-hoc methods, which replace the confidence estimator of a given classifier without modifying or retraining it, thus being practically appealing. Considering neural networks with softmax outputs, our goal is to identify the best confidence estimator that can be computed directly from the unnormalized logits. This problem is motivated by the intriguing observation in recent work that many classifiers appear to have a "broken" confidence estimator, in the sense that their selective classification performance is much worse than what could be expected by their corresponding accuracies. We perform an extensive experimental study of many existing and proposed confidence estimators applied to 84 pretrained ImageNet classifiers available from popular repositories. Our results show that a simple $p$-norm normalization of the logits, followed by taking the maximum logit as the confidence estimator, can lead to considerable gains in selective classification performance, completely fixing the pathological behavior observed in many classifiers. As a consequence, the selective classification performance of any classifier becomes almost entirely determined by its corresponding accuracy. Moreover, these results are shown to be consistent under distribution shift.

## 1 INTRODUCTION

Consider a machine learning classifier that does not reach the desired performance for the intended application, even after significant development time. This may occur for a variety of reasons: the problem is too hard for the current technology; more development resources (data, compute or time) are needed than what is economically feasible for the specific situation; or perhaps the target distribution is different from the training one, resulting in a performance gap. In this case, one is faced with the choice of deploying an underperforming model or not deploying a model at all.

A better tradeoff may be achieved by using so-called selective classification [Geifman and El-Yaniv, 2017, El-Yaniv and Wiener, 2010]. The idea is to run the model on all inputs but reject predictions for which the model is least confident, hoping to increase the performance on the accepted predictions. The rejected inputs may be processed in the same way as if the model were not deployed, for instance, by a human specialist or by the previously existing system. This offers a tradeoff between performance and *coverage* (the proportion of accepted predictions) which may be a better solution than any of the extremes. In particular, it could shorten the path to adoption of deep learning in safety-critical applications, such as medical diagnosis and autonomous driving, where the consequences of erroneous decisions can be severe [Zou et al., 2023, Neumann et al., 2018].

A key element in selective classification is the confidence estimator that is thresholded to decide whether a prediction is accepted. In the case of neural networks with softmax outputs, the natural baseline to be used as a confidence estimator is the maximum softmax probability (MSP) produced by the model, also known as the softmax response [Geifman and El-Yaniv, 2017, Hendrycks and Gimpel, 2016]. Several approaches have been proposed attempting to improve upon this baseline, which generally fall into two categories: approaches that require retraining the classifier, by modifying some aspect of the architecture or the training procedure, possibly adding an auxiliary head as the confidence estimator [Geifman and El-Yaniv, 2019, Liu et al., 2019, Huang et al., 2020]; and post-hoc approaches that do not require retraining, thus only modifying or replacing the confidence estimator based on outputs or intermediate features produced by the model [Corbière et al., 2022, Granese et al.,

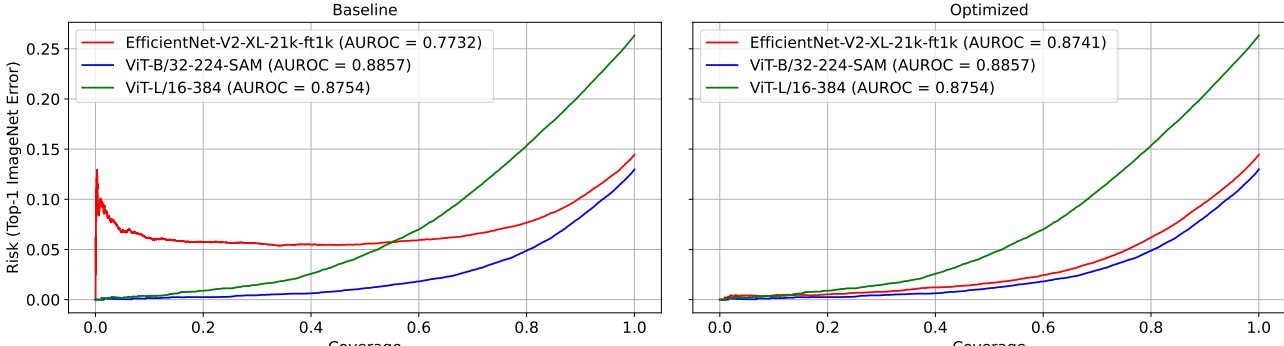

Figure 1: A comparison of RC curves made by three models selected in [Galil et al., 2023], including examples of highest (ViT-L/16-384) and lowest (EfficientNet-V2-XL) AUROC. An RC curve shows the tradeoff between risk (in this case, error rate) and coverage. The initial risk for any classifier is found at the 100% coverage point, where all predictions are accepted. Normally, the risk can be reduced by reducing coverage (which is done by increasing the selection threshold); for instance, a 2% error rate can be obtained at 36.2% coverage for the ViT-B/32-224-SAM model and at 61.9% coverage for the ViT-L/16-38 model. However, for the EfficientNet-V2-XL model, this error rate is not achievable at any coverage, since its RC curve is lower bounded by 5% risk. Moreover, this RC curve is actually non-monotonic, with an increasing risk as coverage is reduced, for low coverage. Fortunately, this apparent pathology in EfficientNet-V2-XL completely disappears after a simple post-hoc tuning of its confidence estimator (without the need to retrain the model), resulting in significantly improved selective classification performance. In particular, a 2% error rate can then be achieved at 55.3% coverage.

2021, Shen et al., 2022, Galil et al., 2023]. The latter is arguably the most practical scenario, especially if tuning the confidence estimator is sufficiently simple.

In this paper, we focus on the simplest possible class of post-hoc methods, which are those for which the confidence estimator can be computed directly from the network unnormalized *logits* (pre-softmax output). Our main goal is to identify the methods that produce the largest gains in selective classification performance, measured by the area under the risk-coverage curve (AURC); however, as in general these methods can have hyperparameters that need to be tuned on hold-out data, we are also concerned with data efficiency. Our study is motivated by an intriguing problem reported in [Galil et al., 2023] and illustrated in Fig. 1: some state-of-the-art ImageNet classifiers, despite attaining excellent predictive performance, nevertheless exhibit appallingly poor performance at detecting their own mistakes. Can such pathologies be fixed by simple post-hoc methods?

To answer this question, we consider every such method to our knowledge, as well as several variations and novel methods that we propose, and perform an extensive experimental study using 84 pretrained ImageNet classifiers available from popular repositories. Our results show that, among other close contenders, a simple $p$-norm normalization of the logits, followed by taking the maximum logit as the confidence estimator, can lead to considerable gains in selective classification performance, completely fixing the pathological behavior observed in many classifiers, as illustrated in Fig. 1. As a consequence, the selective classification performance of any classifier becomes almost entirely determined

by its corresponding accuracy.

The main contributions of this work are summarized as follows:

- We perform an extensive experimental study of many existing and proposed confidence estimators, obtaining considerable gains for most classifiers. In particular, we find that a simple post-hoc estimator can provide up to 62% reduction in normalized AURC using no more than one sample per class of labeled hold-out data;

- We show that, after post-hoc optimization, the selective classification performance of any classifier becomes almost entirely determined by its corresponding accuracy, eliminating the seemingly existing tradeoff between these two goals reported in previous work.

- We also study how these post-hoc methods perform under distribution shift and find that the results remain consistent: a method that provides gains in the in-distribution scenario also provides considerable gains under distribution shift.

## 2 RELATED WORK

Selective prediction is also known as learning with a reject option (see [Zhang et al., 2023, Hendrickx et al., 2021] and references therein), where the rejector is usually a thresholded confidence estimator.[1] Essentially the same problem

---

[1] An interesting application is enabling efficient inference with model cascades [Lebovitz et al., 2023], although the literature on those topics appears disconnected.

is studied under the equivalent terms misclassification detection [Hendrycks and Gimpel, 2016], failure prediction [Corbière et al., 2022, Zhu et al., 2022], and (ordinal) ranking [Moon et al., 2020, Galil et al., 2023]. Uncertainty estimation is a more general term that encompasses these tasks (where confidence may be taken as negative uncertainty) as well as other tasks where uncertainty might be useful, such as calibration and out-of-distribution (OOD) detection, among others [Gawlikowski et al., 2022, Abdar et al., 2021]. These tasks are generally not aligned: for instance, optimizing for calibration may harm selective classification performance [Ding et al., 2020, Zhu et al., 2022, Galil et al., 2023]. Our focus here is on in-distribution selective classification, although we also study robustness to distribution shift.

Most approaches to selective classification consider the base model as part of the learning problem [Geifman and El-Yaniv, 2019, Huang et al., 2020, Liu et al., 2019], which we refer to as training-based approaches. While such an approach has a theoretical appeal, the fact that it requires retraining a model is a significant practical drawback. Alternatively, one may keep the model fixed and only modify or replace the confidence estimator, which is known as a post-hoc approach. Such an approach is practically appealing and perhaps more realistic, as it does not require retraining. Some papers that follow this approach construct a *meta-model* that feeds on intermediate features of the base model and is trained to predict whether or not the base model is correct on hold-out samples [Corbière et al., 2022, Shen et al., 2022]. However, depending on the size of such a meta-model, its training may still be computationally demanding.

A popular tool in the uncertainty literature is the use of ensembles [Lakshminarayanan et al., 2017, Teye et al., 2018, Ayhan and Berens, 2018], of which Monte-Carlo dropout Gal and Ghahramani [2016] is a prominent example. While constructing a confidence estimator from ensemble component outputs may be considered post-hoc if the ensemble is already trained, the fact that multiple inference passes need to be performed significantly increases the computational burden at test time. Moreover, recent work has found evidence that ensembles may not be fundamental for uncertainty but simply better predictive models [Abe et al., 2022, Cattelan and Silva, 2022, Xia and Bouganis, 2022]. Thus, we do not consider ensembles here.

In this work we focus on simple post-hoc confidence estimators for softmax networks that can be directly computed from the logits. The earliest example of such a post-hoc method used for selective classification in a real-world application seems to be the use of LogitsMargin in [Le Cun et al., 1990]. While potentially suboptimal, such methods are extremely simple to apply on top of any trained classifier and should be natural choice to try before any more complex technique. In fact, it is not entirely obvious how a training-based approach should be compared to a post-hoc

method. For instance, Feng et al. [2023] has found that, for some state-of-the-art training-based approaches to selective classification, *after* the main classifier has been trained with the corresponding technique, better selective classification performance can be obtained by discarding the auxiliary output providing confidence values and simply use the conventional MSP as the confidence estimator. Thus, in this sense, the MSP can be seen as a strong baseline.

Post-hoc methods have been widely considered in the context of calibration, among which the most popular approach is temperature scaling (TS). Applying TS to improve calibration (of the MSP confidence estimator) was originally proposed in [Guo et al., 2017] based on the negative log-likelihood. Optimizing TS for other metrics has been explored in [Mukhoti et al., 2020, Karandikar et al., 2021, Clarté et al., 2023] for calibration and in [Liang et al., 2023] for OOD detection, but had not been proposed for selective classification. A generalization of TS is adaptive TS (ATS) [Balanya et al., 2023], which uses an input-dependent temperature based on logits. The post-hoc methods we consider here can be seen as a special case of ATS, as logit norms may be seen as an input-dependent temperature; however Balanya et al. [2023] investigate a different temperature function and focuses on calibration. (For more discussion on this and other post-hoc methods inspired by calibration, please see Appendix H.) Other logit-based confidence estimators proposed for calibration and OOD detection include [Liu et al., 2020, Tomani et al., 2022, Rahimi et al., 2022, Neumann et al., 2018, Gonsior et al., 2022].

Normalizing the logits with the $L_2$ norm before applying the softmax function was used in [Kornblith et al., 2021] and later proposed and studied in [Wei et al., 2022] as a training technique (combined with TS) to improve OOD detection and calibration. A variation where the logits are normalized to unit variance was proposed in [Jiang et al., 2023] to accelerate training. In contrast, we propose to use logit normalization as a post-hoc method for selective classification, extend it to general $p$-norm, consider a tunable $p$ with AURC as the optimization objective, and allow it to be used with confidence estimators other than the MSP, all of which are new ideas which depart significantly from previous work.

Benchmarking of models in their performance at selective classification/misclassification detection has been done in [Galil et al., 2023, Ding et al., 2020], however these works mostly consider the MSP as the confidence estimator. In particular, a thorough evaluation of potential post-hoc estimators for selective classification as done in this work had not yet appeared in the literature. The work furthest in that direction is the paper by Galil et al. [2023], who empirically evaluated ImageNet classifiers and found that TS-NLL improved selective classification performance for some models but degraded it for others. In the context of calibration, Wang et al. [2021] and Ashukha et al. [2020]

have argued that models should be compared after simple post-hoc optimizations, since models that appear worse than others can sometimes easily be improved by methods such as TS. Here we advocate and provide further evidence for this approach in the context of selective classification.

# 3 BACKGROUND

## 3.1 SELECTIVE CLASSIFICATION

Let $P$ be an unknown distribution over $\mathcal{X} \times \mathcal{Y}$, where $\mathcal{X}$ is the input space and $\mathcal{Y} = \{1, \ldots, C\}$ is the label space, and $C$ is the number of classes. The *risk* of a *classifier* $h : \mathcal{X} \to \mathcal{Y}$ is $R(h) = E_P[\ell(h(x), y)]$, where $\ell : \mathcal{Y} \times \mathcal{Y} \to \mathbb{R}^+$ is a loss function, for instance, the 0/1 loss $\ell(\hat{y}, y) = \mathbb{1}[\hat{y} \neq y]$, where $\mathbb{1}[\cdot]$ denotes the indicator function. A *selective classifier* [Geifman and El-Yaniv, 2017] is a pair $(h, g)$, where $h$ is a classifier and $g : \mathcal{X} \to \mathbb{R}$ is a *confidence estimator* (also known as *confidence score function* or *confidence-rate function*), which quantifies the model's confidence on its prediction for a given input. For some fixed threshold $t$, given an input $x$, the selective model makes a prediction $h(x)$ if $g(x) \geq t$, otherwise the prediction is rejected. A selective model's *coverage* $\phi(h, g) = P[g(x) \geq t]$ is the probability mass of the selected samples in $\mathcal{X}$, while its *selective risk* $R(h, g) = E_P[\ell(h(x), y) \mid g(x) \geq t]$ is its risk restricted to the selected samples. In particular, a model's risk equals its selective risk at *full coverage* (i.e., for $t$ such that $\phi(h, g) = 1$). These quantities can be evaluated empirically given a given a test dataset $\{(x_i, y_i)\}_{i=1}^{N}$ drawn i.i.d. from $P$, yielding the *empirical coverage* $\hat{\phi}(h, g) = (1/N) \sum_{i=1}^{N} \mathbb{1}[g(x_i) \geq t]$ and the *empirical selective risk*

$$\hat{R}(h, g) = \frac{\sum_{i=1}^{N} \ell(h(x_i), y_i) \mathbb{1}[g(x_i) \geq t]}{\sum_{i=1}^{N} \mathbb{1}[g(x_i) \geq t]}. \quad (1)$$

Note that, by varying $t$, it is generally possible to trade off coverage for selective risk, i.e., a lower selective risk can usually (but not necessarily always) be achieved if more samples are rejected. This tradeoff is captured by the *risk-coverage (RC) curve* [Geifman and El-Yaniv, 2017], a plot of $\hat{R}(h, g)$ as a function of $\hat{\phi}(h, g)$. While the RC curve provides a full picture of the performance of a selective classifier, it is convenient to have a scalar metric that summarizes this curve. A commonly used metric is the *area under the RC curve* (AURC) [Ding et al., 2020, Geifman et al., 2019], denoted by AURC$(h, g)$. However, when comparing selective models, if two RC curves cross, then each model may have a better selective performance than the other depending on the operating point chosen, which cannot be captured by the AURC. Another interesting metric, which forces the choice of an operating point, is the *selective accuracy constraint* (SAC) [Galil et al., 2023], defined as the maximum coverage allowed for a model to achieve a specified accuracy.

Closely related to selective classification is misclassification detection [Hendrycks and Gimpel, 2016], which refers to the problem of discriminating between correct and incorrect predictions made by a classifier. Both tasks rely on ranking predictions according to their confidence estimates, where correct predictions should be ideally separated from incorrect ones. A usual metric for misclassification detection is the area under the ROC curve (AUROC) [Fawcett, 2006] which, in contrast to the AURC, is blind to the classifier performance, focusing only on the quality of the confidence estimates. Thus, it may also be used to evaluate confidence estimators for selective classification [Galil et al., 2023].

## 3.2 CONFIDENCE ESTIMATION

From now on we restrict attention to classifiers that can be decomposed as $h(x) = \arg\max_{k \in \mathcal{Y}} z_k$, where $\mathbf{z} = f(x)$ and $f : \mathcal{X} \to \mathbb{R}^C$ is a neural network. The network output $\mathbf{z}$ is referred to as the (vector of) *logits* or *logit vector*, due to the fact that it is typically applied to a softmax function to obtain an estimate of the posterior distribution $P[y|x]$. The softmax function is defined as

$$\sigma : \mathbb{R}^C \to [0, 1]^C, \quad \sigma_k(\mathbf{z}) = \frac{e^{z_k}}{\sum_{j=1}^{C} e^{z_j}}, \quad k \in \{1, \ldots, C\} \quad (2)$$

where $\sigma_k(\mathbf{z})$ denotes the $k$th element of the vector $\sigma(\mathbf{z})$.

The most popular confidence estimator is arguably the *maximum softmax probability* (MSP) [Ding et al., 2020], also known as *maximum class probability* [Corbière et al., 2022] or *softmax response* [Geifman and El-Yaniv, 2017]

$$g(x) = \text{MSP}(\mathbf{z}) \triangleq \max_{k \in \mathcal{Y}} \sigma_k(\mathbf{z}) = \sigma_{\hat{y}}(\mathbf{z}) \quad (3)$$

where $\hat{y} = \arg\max_{k \in \mathcal{Y}} z_k$. However, other functions of the logits can be considered. Some examples are the *softmax margin* [Belghazi and Lopez-Paz, 2021, Lubrano et al., 2023], the *max logit* [Hendrycks et al., 2022], the *logits margin* [Streeter, 2018, Lebovitz et al., 2023], the *negative entropy*[2] [Belghazi and Lopez-Paz, 2021], and the *negative Gini index* [Granese et al., 2021, Gomes et al., 2022], defined, respectively, as

$$\text{SoftmaxMargin}(\mathbf{z}) \triangleq \sigma_{\hat{y}}(\mathbf{z}) - \max_{k \in \mathcal{Y}:k \neq \hat{y}} \sigma_k(\mathbf{z}) \quad (4)$$

$$\text{MaxLogit}(\mathbf{z}) \triangleq z_{\hat{y}} \quad (5)$$

$$\text{LogitsMargin}(\mathbf{z}) \triangleq z_{\hat{y}} - \max_{k \in \mathcal{Y}:k \neq \hat{y}} z_k \quad (6)$$

$$\text{NegativeEntropy}(\mathbf{z}) \triangleq \sum_{k \in \mathcal{Y}} \sigma_k(\mathbf{z}) \log \sigma_k(\mathbf{z}) \quad (7)$$

$$\text{NegativeGini}(\mathbf{z}) \triangleq -1 + \sum_{k \in \mathcal{Y}} \sigma_k(\mathbf{z})^2. \quad (8)$$

---

[2]Note that any uncertainty estimator can be used as a confidence estimator by taking its negative.

Note that, in the scenario we consider, DOCTOR's $D_\alpha$ and $D_\beta$ discriminators [Granese et al., 2021] are equivalent to the negative Gini index and MSP confidence estimators, respectively, as discussed in more detail in Appendix A.

It is worth mentioning that, as shown by Chow [1970] and Franc et al. [2023], if indeed $\sigma_y(\mathbf{z}) = P[y|x]$ for all $y \in \mathcal{Y}$, then the MSP is the optimal confidence estimator for the 0/1 loss, known in this case as Chow's rule. Thus, in the general case, it emerges as a natural baseline.

# 4 METHODS

## 4.1 TUNABLE LOGIT TRANSFORMATIONS

In this section, we introduce a simple but powerful framework for designing post-hoc confidence estimators for selective classification. The idea is to take any parameter-free logit-based confidence estimator, such as those described in Section 3.2, and augment it with a logit transformation parameterized by one or a few hyperparameters, which are then tuned (e.g., via grid search) using a labeled hold-out dataset not used during training of the classifier (i.e. validation data). Moreover, this hyperparameter tuning is done using as objective function not a proxy loss but rather the exact same metric that one is interested in optimizing, for instance, AURC or AUROC. This approach forces us to be conservative about the hyperparameter search space, which is important for data efficiency.

### 4.1.1 Temperature Scaling

Originally proposed in the context of post-hoc calibration, temperature scaling (TS) [Guo et al., 2017] consists in transforming the logits as $\mathbf{z}' = \mathbf{z}/T$, before applying the softmax function. The parameter $T > 0$, which is called the temperature, is then optimized over hold-out data.

The conventional way of applying TS, as proposed in [Guo et al., 2017] for calibration and referred to here as TS-NLL, consists in optimizing $T$ with respect to the negative log-likelihood (NLL) [Murphy, 2022]. Here we instead optimize $T$ using AURC and the resulting method is referred to as TS-AURC.

Note that TS does not affect the ranking of predictions for MaxLogit and LogitsMargin, so it is not applied in these cases.

### 4.1.2 Logit Normalization

Inspired by Wei et al. [2022], who show that logits norms are directly related to overconfidence and propose logit normalization during training, we propose logit normalization as a post-hoc method. Additionally, we extend the normal-

ization from the 2-norm to a general $p$-norm, where $p$ is a tunable hyperparameter and, similarly to the method proposed in [Jiang et al., 2023], we propose to *centralize* the logits before normalization. (For more context on logit normalization, as well as intuition and theoretical justification for our proposed modifications, see the Appendix B. For an ablation study on the centralization, see Appendix G.) Thus, (centralized) logit $p$-normalization is defined as the operation

$$\mathbf{z}' = \frac{\mathbf{z} - \mu(\mathbf{z})}{\tau \|\mathbf{z} - \mu(\mathbf{z})\|_p} \tag{9}$$

where $\|\mathbf{z}\|_p \triangleq (|z_1|^p + \cdots + |z_C|^p)^{1/p}$, $p \in \mathbb{R}$, is the $p$-norm of $\mathbf{z}$, $\mu(\mathbf{z}) = \frac{1}{C} \sum_{j=1}^{C} z_j$ is the mean of the logits, and $\tau > 0$ is a temperature scaling parameter. Note that, when the softmax function is used, this transformation becomes a form of adaptive TS [Balanya et al., 2023], with an input-dependent temperature $\tau \|\mathbf{z} - \mu(\mathbf{z})\|_p$.

Logit $p$-normalization introduces two hyperparameters, $p$ and $\tau$, which should be jointly optimized; in this case, we first optimize $\tau$ for each value of $p$ considered and then pick the best value of $p$. This transformation, together with the optimization of $p$ and $\tau$, is here called pNorm. The optimizing metric is always AURC and therefore it is omitted from the nomenclature of the method.

Note that, when the underlying confidence estimator is MaxLogit or LogitsMargin, the parameter $\tau$ is irrelevant and is ignored.

One key benefit of centralization is that it enables logit $p$-normalization to be applied even if we only have access to the softmax probabilities instead of the original logits. This can be done by computing the logits as $\tilde{\mathbf{z}} = \log(\sigma(\mathbf{z})) = \mathbf{z} - c$, where $c = \log(\sum_{j=1}^{C} e^{z_j})$. Then we have $\tilde{\mathbf{z}} - \mu(\tilde{\mathbf{z}}) = \mathbf{z} - c - \mu(\mathbf{z} - c) = \mathbf{z} - \mu(\mathbf{z})$ from which (9) can be computed.

## 4.2 EVALUATION METRICS

### 4.2.1 Normalized AURC

A common criticism of the AURC metric is that it does not allow for meaningful comparisons across problems [Geifman et al., 2019]. An AURC of some arbitrary value, for instance, 0.05, may correspond to an ideal confidence estimator for one classifier (of much higher risk) and to a completely random confidence estimator for another classifier (of risk equal to 0.05). The excess AURC (E-AURC) was proposed by Geifman et al. [2019] to alleviate this problem: for a given classifier $h$ and confidence estimator $g$, it is defined as E-AURC$(h, g) = $ AURC$(h, g) - $ AURC$(h, g^*)$, where $g^*$ corresponds to a hypothetically optimal confidence estimator that perfectly orders samples in decreasing order of their losses. Thus, an ideal confidence estimator always has zero E-AURC.

Unfortunately, E-AURC is still highly sensitive to the classifier's risk, as shown by Galil et al. [2023], who suggested the use of AUROC instead. However, using AUROC for comparing confidence estimators has an intrinsic disadvantage: if we are using AUROC to evaluate the performance of a tunable confidence estimator, it makes sense to optimize it using this same metric. However, as AUROC and AURC are not necessarily monotonically aligned [Ding et al., 2020], the resulting confidence estimator will be optimized for a different problem than the one in which we were originally interested (which is selective classification). Ideally, we would like to evaluate confidence estimators using a metric that is a monotonic function of AURC.

We propose a simple modification to E-AURC that eliminates the shortcomings pointed out in [Galil et al., 2023]: normalizing by the E-AURC of a random confidence estimator, whose AURC is equal to the classifier's risk. More precisely, we define the normalized AURC (NAURC) as

$$\text{NAURC}(h, g) = \frac{\text{AURC}(h, g) - \text{AURC}(h, g^*)}{R(h) - \text{AURC}(h, g^*)}. \quad (10)$$

Note that this corresponds to a min-max scaling that maps the AURC of the ideal classifier to 0 and the AURC of the random classifier to 1. The resulting NAURC is suitable for comparison across different classifiers and is monotonically related to AURC.

### 4.2.2 MSP Fallback

A useful property of MSP-TS-AURC (but not MSP-TS-NLL) is that, in the infinite-sample setting, it can never have a worse performance than the MSP baseline, as long as $T = 1$ is included in the search space. It is natural to extend this property to every confidence estimator, for a simple reason: it is very easy to check whether the estimator provides an improvement to the MSP baseline and, if not, then use the MSP instead. Formally, this corresponds to adding a binary hyperparameter indicating an MSP fallback.

Equivalently, when measuring performance across different models, we simply report a (non-negligible) positive gain in NAURC whenever it occurs. More precisely, we define the *average positive gain* (APG) in NAURC as

$$\text{APG}(g) = \frac{1}{|\mathcal{H}|} \sum_{h \in \mathcal{H}} [\text{NAURC}(h, \text{MSP}) - \text{NAURC}(h, g)]_\epsilon^+ \quad (11)$$

where $[x]_\epsilon^+$ is defined as $x$ if $x > \epsilon$ and is 0 otherwise, $\mathcal{H}$ is a set of classifiers and $\epsilon > 0$ is chosen so that only non-negligible gains are reported.

## 5 EXPERIMENTS

All experiments[3] in this section were performed using PyTorch [Paszke et al., 2019] and all of its provided classifiers pre-trained on ImageNet [Deng et al., 2009]. Additionally, some models of the Wightman [2019] repository were used, particularly the ones highlighted by Galil et al. [2023]. In total, 84 ImageNet classifiers were used. The list of all models, together with all the results per model are presented in Appendix J. The ImageNet validation set was randomly split into 5000 hold-out images for post-hoc optimization (which we also refer to as the *tuning set*) and 45000 images for performance evaluation (the test set). To ensure that the results are statistically significant, we repeat each experiment (including post-hoc optimization) for 10 different random splits and report mean and standard deviation.

To give evidence that our results are not specific to ImageNet, we also performed experiments on CIFAR-100 [Krizhevsky, 2009] and Oxford-IIIT Pet [Parkhi et al., 2012] datasets, which are presented in the Appendix D.

### 5.1 COMPARISON OF METHODS

We start by evaluating the NAURC of each possible combination of a confidence estimator listed in Section 3.2 with a logit transformation described in Section 4.1, for specific models. Table 1 shows the results for EfficientNet-V2-XL (trained on ImageNet-21K and fine tuned on ImageNet-1K) and VGG16, respectively, the former chosen for having the worst confidence estimator performance (in terms of AUROC, with MSP as the confidence estimator) of all the models reported in [Galil et al., 2023] and the latter chosen as a representative example of a lower accuracy model for which the MSP is already a good confidence estimator.

As can be seen, on EfficientNet-V2-XL, the baseline MSP is easily outperformed by most methods. Surprisingly, the best method is not to use a softmax function but, instead, to take the maximum of a $p$-normalized logit vector, leading to a reduction in NAURC of 0.27 points or about 62%.

However, on VGG16, the situation is quite different, as methods that use the unnormalized logits and improve the performance on EfficientNet-V2-XL, such as LogitsMargin and MaxLogit-pNorm, actually degrade it on VGG16. Moreover, the highest improvement obtained, e.g., with MSP-TS-AURC, is so small that it can be considered negligible. (In fact, gains below 0.003 NAURC are visually imperceptible in an AURC curve.) Thus, it is reasonable to assert that none of the post-hoc methods considered is able to outperform the baseline in this case.

In Table 2, we evaluate the average performance of post-

---

[3] Our code is available at `https://github.com/lfpc/FixSelectiveClassification`.

Table 1: NAURC (mean ±std) for post-hoc methods applied to ImageNet classifiers

| Classifier | Conf. Estimator | Logit Transformation | | | |
| | | Raw | TS-NLL | TS-AURC | pNorm |
|---|---|---|---|---|---|
| EfficientNet-V2-XL | MSP | 0.4402 ±0.0032 | 0.3506 ±0.0039 | 0.1957 ±0.0027 | 0.1734 ±0.0030 |
| | SoftmaxMargin | 0.3816 ±0.0031 | 0.3144 ±0.0034 | 0.1964 ±0.0046 | 0.1726 ±0.0026 |
| | MaxLogit | 0.7680 ±0.0028 | - | - | **0.1693** ±0.0018 |
| | LogitsMargin | 0.1937 ±0.0023 | - | - | 0.1728 ±0.0020 |
| | NegativeEntropy | 0.5967 ±0.0031 | 0.4295 ±0.0057 | 0.1937 ±0.0023 | 0.1719 ±0.0022 |
| | NegativeGini | 0.4486 ±0.0032 | 0.3517 ±0.0040 | 0.1957 ±0.0027 | 0.1732 ±0.0030 |
| VGG16 | MSP | **0.1839** ±0.0006 | 0.1851 ±0.0006 | 0.1839 ±0.0007 | 0.1839 ±0.0007 |
| | SoftmaxMargin | 0.1900 ±0.0006 | 0.1892 ±0.0006 | 0.1888 ±0.0006 | 0.1888 ±0.0006 |
| | MaxLogit | 0.3382 ±0.0009 | - | - | 0.2020 ±0.0012 |
| | LogitsMargin | 0.2051 ±0.0005 | - | - | 0.2051 ±0.0005 |
| | NegativeEntropy | 0.1971 ±0.0007 | 0.2055 ±0.0006 | 0.1841 ±0.0006 | 0.1841 ±0.0006 |
| | NegativeGini | 0.1857 ±0.0007 | 0.1889 ±0.0005 | 0.1840 ±0.0006 | 0.1840 ±0.0006 |

Table 2: APG-NAURC (mean ±std) of post-hoc methods across 84 ImageNet classifiers

| Conf. Estimator | Logit Transformation | | | |
| | Raw | TS-NLL | TS-AURC | pNorm |
|---|---|---|---|---|
| MSP | 0.0 ± 0.0 | 0.03665 ±0.00034 | 0.05769 ±0.00038 | 0.06796 ±0.00051 |
| SoftmaxMargin | 0.01955 ±0.00008 | 0.04113 ±0.00022 | 0.05601 ±0.00041 | 0.06608 ±0.00052 |
| MaxLogit | 0.0 ± 0.0 | - | - | **0.06863** ±0.00045 |
| LogitsMargin | 0.05531 ±0.00042 | - | - | 0.06204 ±0.00046 |
| NegativeEntropy | 0.0 ± 0.0 | 0.01570 ±0.00085 | 0.05929 ±0.00032 | 0.06771 ±0.00052 |
| NegativeGini | 0.0 ± 0.0 | 0.03636 ±0.00042 | 0.05809 ±0.00037 | 0.06800 ±0.00054 |

hoc methods across all models considered, using the APG-NAURC metric described in Section 4.2.2, where we assume $\epsilon = 0.01$. Figure 2 shows the gains for selected methods for each model, ordered by MaxLogit-pNorm gains. It can be seen that the highest gains are provided by MaxLogit-pNorm, NegativeGini-pNorm, MSP-pNorm and NegativeEntropy-pNorm, and their performance is essentially indistinguishable whenever they provide a non-negligible gain over the baseline. Moreover, the set of models for which significant gains can be obtained appears to be consistent across all methods.

Although several post-hoc methods provide considerable gains, they all share a practical limitation which is the requirement of hold-out data for hyperparameter tuning. In Appendix E, we study the data efficiency of some of the best performing methods. MaxLogit-pNorm, having a single hyperparameter, emerges as a clear winner, requiring fewer than 500 samples to achieve near-optimal performance on ImageNet ($< 0.5$ images per class on average) and fewer than 100 samples on CIFAR-100 ($< 1$ image per class on average). These requirements are clearly easily satisfied in practice for typical validation set sizes.

Details on the optimization of $T$ and $p$, additional results

showing AUROC values and RC curves, and results on the insensitivity of our conclusions to the choice of $\epsilon$ are provided in Appendix C. In addition, the benefits of a tunable versus fixed $p$ and a comparison with other tunable methods that do not fit into the framework of Section 4.1 are discussed, respectively, in Appendices F and H. Finally, an investigation of the calibration performance of some methods can be found in Appendix I.

## 5.2 POST-HOC OPTIMIZATION FIXES BROKEN CONFIDENCE ESTIMATORS

From Figure 2, we can distinguish two groups of models: those for which the MSP baseline is already the best confidence estimator and those for which post-hoc methods provide considerable gains (particularly, MaxLogit-pNorm). In fact, most models belong to the second group, comprising 58 of the 84 models considered.

Figure 3 illustrates two noteworthy phenomena. First, as previously observed by Galil et al. [2023], certain models exhibit superior accuracy than others but poorer uncertainty estimation, leading to a trade-off when selecting a classifier for selective classification. Second, post-hoc optimization

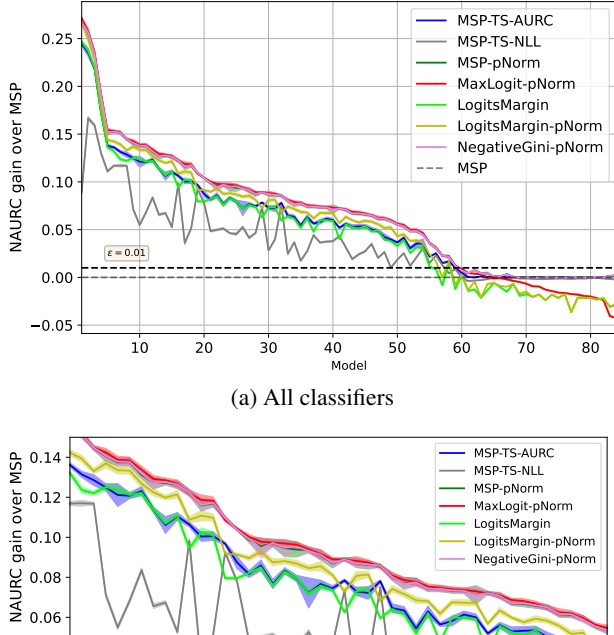

(a) All classifiers

(b) Close up

Figure 2: NAURC gains for post-hoc methods across 84 ImageNet classifiers. Lines indicate the average of 10 random splits and the filled regions indicate $\pm 1$ standard deviation. The black dashed line denotes $\epsilon = 0.01$.

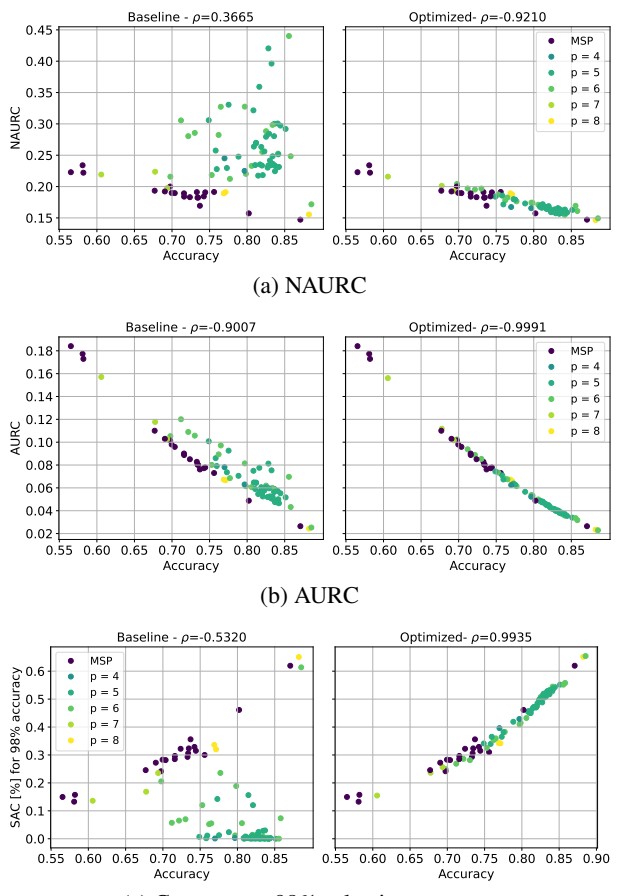

(a) NAURC

(b) AURC

(c) Coverage at 98% selective accuracy

Figure 3: NAURC, AURC and SAC of 84 ImageNet classifiers with respect to their accuracy, before and after post-hoc optimization. The baseline plots use MSP, while the optimized plots use MaxLogit-pNorm. The legend shows the optimal value of $p$ for each model, where MSP indicates MSP fallback (no significant positive gain). $\rho$ is the Spearman correlation between a metric and the accuracy. In (c), models that cannot achieve the desired selective accuracy are shown with $\approx 0$ coverage.

can fix any "broken" confidence estimators. This can be seen in two ways: In Figure 3a, after optimization, all models exhibit a much more similar level of confidence estimation performance (as measured by NAURC), although a dependency on accuracy is clearly seen (better predictive models are better at predicting their own failures). In Figure 3b, it is clear that, after optimization, the selective classification performance of any classifier (measured by AURC) becomes almost entirely determined by its corresponding accuracy. Indeed, the Spearman correlation between AURC and accuracy becomes extremely close to 1. The same conclusions hold for the SAC metric, as shown in Figure 3c. This implies that any "broken" confidence estimators have been fixed, and consequently, total accuracy becomes the primary determinant of selective performance even at lower coverage levels.

### 5.3 PERFORMANCE UNDER DISTRIBUTION SHIFT

We now turn to the question of how post-hoc methods for selective classification perform under distribution shift. Previous works have shown that calibration can be harmed under distribution shift, especially when certain post-hoc methods—such as TS—are applied [Ovadia et al., 2019]. To find out whether a similar issue occurs for selective

classification, we evaluate selected post-hoc methods on ImageNet-C [Hendrycks and Dietterich, 2018], which consists in 15 different corruptions of the ImageNet's validation set, and on ImageNetV2 [Recht et al., 2019], which is an independent sampling of the ImageNet test set replicating the original dataset creation process. We follow the standard approach for evaluating robustness with these datasets, which is to use them only for inference; thus, the post-hoc methods are optimized using only the 5000 hold-out images from the uncorrupted ImageNet validation dataset. To avoid data leakage, the same split is applied to the ImageNet-C dataset, so that inference is performed only on the 45000 images originally selected as the test set.

First, we evaluate the performance of MaxLogit-pNorm on ImageNet and ImageNetV2 for all classifiers considered.

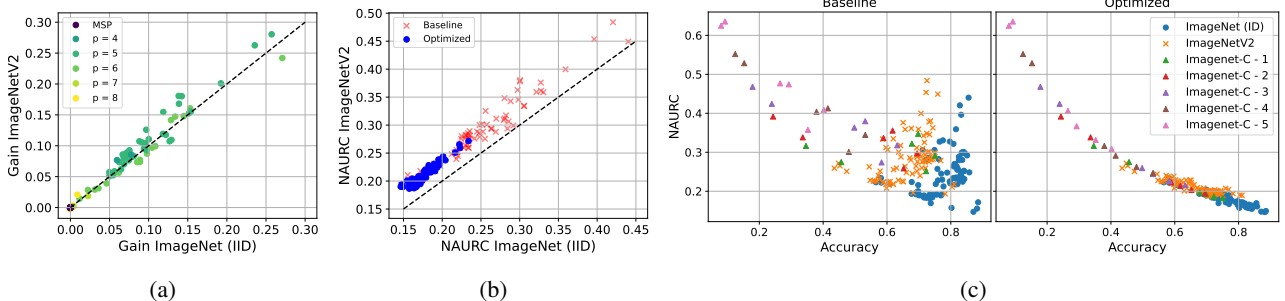

(a)  (b)  (c)

Figure 4: (a) NAURC gains (over MSP) on ImageNetV2 versus NAURC gains on the ImageNet test set. (b) NAURC on ImageNetV2 versus NAURC on the ImageNet test set. (c) NAURC versus accuracy for ImageNetV2, ImageNet-C and the IID dataset. All models are optimized using MaxLogit-pNorm (with MSP fallback).

Table 3: Selective classification performance (achievable coverage for some target selective accuracy; mean $_{\pm\text{std}}$) for a ResNet-50 on ImageNet under distribution shift. For ImageNet-C, each entry is the average across all corruption types for a given level of corruption. The target accuracy is the one achieved for corruption level 0.

| | | \multicolumn{6}{c}{Corruption level} | |
| | Method | 0 | 1 | 2 | 3 | 4 | 5 | V2 |
|---|---|---|---|---|---|---|---|---|
| Accuracy[%] | - | 80.84 | 67.81 $_{\pm0.05}$ | 58.90 $_{\pm0.04}$ | 49.77 $_{\pm0.04}$ | 37.92 $_{\pm0.03}$ | 26.51 $_{\pm0.03}$ | 69.77 $_{\pm0.10}$ |
| Coverage (SAC) [%] | MSP | 100 | 72.14 $_{\pm0.11}$ | 52.31 $_{\pm0.13}$ | 37.44 $_{\pm0.11}$ | 19.27 $_{\pm0.07}$ | 8.53 $_{\pm0.12}$ | 76.24 $_{\pm0.22}$ |
| | MSP-TS-AURC | 100 | 72.98 $_{\pm0.23}$ | 55.87 $_{\pm0.27}$ | 40.89 $_{\pm0.21}$ | 24.65 $_{\pm0.19}$ | 12.52 $_{\pm0.05}$ | 76.22 $_{\pm0.41}$ |
| | MaxLogit-pNorm | 100 | **75.24** $_{\pm0.15}$ | **58.58** $_{\pm0.27}$ | **43.67** $_{\pm0.37}$ | **27.03** $_{\pm0.36}$ | **14.51** $_{\pm0.26}$ | **78.66** $_{\pm0.38}$ |

Figure 4a shows that the NAURC gains (over the MSP baseline) obtained for ImageNet translate to similar gains for ImageNetV2, showing that this post-hoc method is quite robust to distribution shift. Then, considering all models after post-hoc optimization with MaxLogit-pNorm, we investigate whether selective classification performance itself (as measured by NAURC) is robust to distribution shift. As can be seen in Figure 4b, the results are consistent, following an affine function (with Pearson's correlation equal to 0.983); however, a significant degradation in NAURC can be observed for all models under distribution shift. While at first sight this would suggest a lack of robustness, a closer look reveals that it can actually be explained by the natural accuracy drop of the underlying classifier under distribution shift. Indeed, we have already noticed in Figure 3a a negative correlation between the NAURC and the accuracy; in Figure 4c these results are expanded by including the evaluation on ImageNetV2 and also (for selected models AlexNet, ResNet50, WideResNet50-2, VGG11, EfficientNet-B3 and ConvNext-Large, sorted by accuracy) on ImageNet-C, where we can see that the strong correlation between NAURC and accuracy continues to hold.

Finally, to give a more tangible illustration of the impact of selective classification, Table 3 shows the SAC metric for a ResNet50 under distribution shift, with the target accuracy as the original accuracy obtained with the in-distribution test data. As can be seen, the original accuracy can be restored

at the expense of coverage; meanwhile, MaxLogit-pNorm achieves higher coverages for all distribution shifts considered, significantly improving coverage over the MSP baseline.

## 6 DISCUSSION

Our work has identified two broad classes of trained models (which comprise 31% and 69% of our sample, respectively): models for which the MSP is apparently an already optimal confidence estimator, in the sense that is not improvable by any of the post-hoc methods we evaluated; and models for which the MSP is suboptimal, in which case all of the best methods evaluated produce highly correlated gains. As a consequence, a few questions naturally arise.

**Why is the MSP such a strong baseline in many cases but easily improvable in many others?** As mentioned in Section 3.2, the MSP is the optimal confidence estimator if the softmax output provides the exact class-posterior distribution. While this is obviously not the case in general, *if the model is designed and trained to estimate this posterior*, e.g., by minimizing the NLL, then it is unlikely that a better estimate can be found by simple post-hoc optimization. For instance, the optimal temperature parameter could be easily learned during training and, more generally, any beneficial logit transformation would already be made part of

the model architecture to maximize performance. However, modern deep learning classifiers are often trained and tuned with the goal of maximizing validation accuracy rather than validation NLL, resulting in overfitting of the latter. Indeed, this was the explanation offered in Guo et al. [2017] for the emergence of overconfidence which motivated their proposal of TS. Similarly, Wei et al. [2022] identified a specific mechanism that could cause this overconfidence, namely, an increasing magnitude of logits during training, which motivated their proposal of logit normalization (see Appendix B for more details). Thus, overconfidence could be the main cause of poor selective classification performance and simple post-hoc tuning could be able to easily improve it. While our results clearly prove this second hypothesis, they actually *disprove* the first, as shown below.

**What is the cause of poor selective classification performance?** According to our experiments in Appendix I, models that produce highly confident MSPs tend to have better confidence estimators (in terms of NAURC), while models whose MSP distribution is more balanced tend to be easily improvable by post-hoc optimization—which, in turn, makes the resulting confidence estimator concentrated on highly confident values. In other words, overconfidence is not necessarily a problem for selective classification, but underconfidence may be. While the root causes of this underconfidence are currently under investigation, some natural suspects are techniques that create soft labels, such as label smoothing [Szegedy et al., 2016] and mixup augmentation [Zhang et al., 2017], which are present in modern training recipes and have already been shown in [Zhu et al., 2022] to be harmful for misclassification detection. In any case, our results reinforce the observations in previous works [Zhu et al., 2022, Galil et al., 2023] that—except in the special case where an ideal probabilistic model can be found—calibration and selective classification are distinct problems and optimizing one may harm the other. In particular, the method with best calibration performance (TS-NLL) achieves only small gains in NAURC, while the method with highest NAURC gains that still deliver probabilities (MSP-pNorm) does not significantly improve calibration and sometimes harms it.

**Why are the gains of all methods highly correlated? Why does post-hoc logit normalization improve performance at all?** One particular case of underconfidence is when the model incorrectly attributes too much posterior probability mass to the least probable classes (e.g., when all classes except the predicted one have the same probability). In this case, LogitsMargin, which effectively disregards all logits except the highest two, may be a better confidence estimator. However, as shown in Appendix B, MSP-TS with small $T$ approximates LogitsMargin, while MaxLogit-pNorm with $p = 1/T$ is closely related to the MSP-TS. Thus, all methods combat underconfidence in a similar way by focus on the largest logits and therefore give highly correlated gains.

Moreover, this explains why using a sufficiently large $p$ is essential in post-hoc $p$-norm logit normalization. On the other hand, as also shown in Appendix B, due to its unique characteristics, MaxLogit-pNorm is even more effective than MSP-TS in combatting this particular form of underconfidence, since it can effectively discard the smallest, least reliable logits without penalizing largest ones.

## 7  CONCLUSION

In this paper, we addressed the problem of selective multi-class classification for deep neural networks with softmax outputs. Specifically, we considered the design of post-hoc confidence estimators that can be computed directly from the unnormalized logits. We performed an extensive benchmark of more than 20 tunable post-hoc methods across 84 ImageNet classifiers, establishing strong baselines for future research. To allow for a fair comparison, we proposed a normalized version of the AURC metric that is insensitive to the classifier accuracy.

Our main conclusions are the following: (1) For 58 (69%) of the models considered, considerable NAURC gains over the MSP can be obtained, in one case achieving a reduction of 0.27 points or about 62%. (2) Our proposed method MaxLogit-pNorm (which does not use a softmax function) emerges as a clear winner, providing the highest gains with exceptional data efficiency, requiring on average less than 1 sample per class of hold-out data for tuning its single hyperparameter. These observations are also confirmed under additional datasets and the gains preserved even under distribution shift. (3) After post-hoc optimization, all models with a similar accuracy achieve a similar level of confidence estimation performance, even models that have been previously shown to be very poor at this task. In particular, the selective classification performance of any classifier becomes almost entirely determined by its corresponding accuracy, eliminating the seemingly existing tradeoff between these two goals reported in previous work. (4) Selective classification performance itself appears to be robust to distribution shift, in the sense that, although it naturally degrades, this degradation is not larger than what would be expected by the corresponding accuracy drop.

We have also investigated what makes a classifier easily improvable by post-hoc methods and found that the issue is related to underconfidence. The root causes of this underconfidence are currently under investigation and will be the subject of our future work.

**Acknowledgements**

The authors thank Bruno M. Pacheco for suggesting the NAURC metric.

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

# Supplementary Material

**Luís Felipe P. Cattelan**[1]                    **Danilo Silva**[1]

[1]Department of Electrical and Electronic Engineering,

Federal University of Santa Catarina (UFSC), Florianópolis, Brazil ,

lfp.cattelan@gmail.com, danilo.silva@ufsc.br

## A   ON THE DOCTOR METHOD

The paper by [Granese et al., 2021] introduces a selection mechanism named DOCTOR, which actually refers to two distinct methods, $D_\alpha$ and $D_\beta$, in two possible scenarios, Total Black Box and Partial Black Box. Only the former scenario corresponds to post-hoc estimators and, in this case, the two methods are equivalent to NegativeGini and MSP, respectively.

To see this, first consider the definition of $D_\alpha$: a sample $x$ is rejected if $1 - \hat{g}(x) > \gamma\hat{g}(x)$, where

$$1 - \hat{g}(x) = \sum_{k \in \mathcal{Y}} (\sigma(\mathbf{z}))_k (1 - (\sigma(\mathbf{z}))_k) = 1 - \sum_{k \in \mathcal{Y}} (\sigma(\mathbf{z}))_k^2 = 1 - \|\sigma(\mathbf{z})\|_2^2$$

is exactly the Gini index of diversity applied to the softmax outputs. Thus, a sample $x$ is accepted if $1 - \hat{g}(x) \leq \gamma\hat{g}(x) \iff (1 + \gamma)\hat{g}(x) \geq 1 \iff \hat{g}(x) \geq 1/(1 + \gamma) \iff \hat{g}(x) - 1 \geq 1/(1 + \gamma) - 1$. Therefore, the method is equivalent to the confidence estimator $g(x) = \hat{g}(x) - 1 = \|\sigma(\mathbf{z})\|^2 - 1$, with $t = 1/(1 + \gamma) - 1$ as the selection threshold.

Now, consider the definition of $D_\beta$: a sample $x$ is rejected if $\hat{P}_e(x) > \gamma(1 - \hat{P}_e(x))$, where $\hat{P}_e(x) = 1 - (\sigma(\mathbf{z}))_{\hat{y}}$ and $\hat{y} = \arg\max_{k \in \mathcal{Y}} (\sigma(\mathbf{z}))_k$, i.e., $\hat{P}_e(x) = 1 - \mathrm{MSP}(\mathbf{z})$. Thus, a sample $x$ is accepted if $\hat{P}_e(x) \leq \gamma(1 - \hat{P}_e(x)) \iff (1 + \gamma)\hat{P}_e(x) \leq \gamma \iff \hat{P}_e(x) \leq \gamma/(1 + \gamma) \iff \mathrm{MSP}(\mathbf{z}) \geq 1 - \gamma/(1 + \gamma) = 1/(1 + \gamma)$. Therefore, the method is equivalent to the confidence estimator $g(x) = \mathrm{MSP}(\mathbf{z})$, with $t = 1/(1 + \gamma)$ as the selection threshold.

Given the above results, one may wonder why the results in [Granese et al., 2021] show different performance values for $D_\beta$ and MSP (softmax response), as shown, for instance, in Table 1 in Granese et al. [2021]. We suspect this discrepancy is due to numerical imprecision in the computation of the ROC curve for a limited number of threshold values, as the authors themselves point out on their Appendix C.3, combined with the fact that $D_\beta$ and MSP in [Granese et al., 2021] use different parametrizations for the threshold values. In contrast, we use the implementation from the scikit-learn library (adapting it as necessary for the RC curve), which considers every possible threshold for the confidence values given and so is immune to this kind of imprecision.

## B   ON LOGIT NORMALIZATION

**Logit normalization during training.** Wei et al. [2022] argued that, as training progresses, a model may tend to become overconfident on correctly classified training samples by increasing $\|\mathbf{z}\|_2$. This is due to the fact that the predicted class depends only on $\tilde{\mathbf{z}} = \mathbf{z}/\|\mathbf{z}\|_2$, but the training loss on correctly classified training samples can still be decreased by increasing $\|\mathbf{z}\|_2$ while keeping $\tilde{\mathbf{z}}$ fixed. Thus, the model would become overconfident on those samples, since increasing $\|\mathbf{z}\|_2$ also increases the confidence (as measured by MSP) of the predicted class. This overconfidence phenomenon was confirmed experimentally in [Wei et al., 2022] by observing that the average magnitude of logits (and therefore also their average 2-norm) tends to increase during training. For this reason, Wei et al. [2022] proposed logit 2-norm normalization during training, as a way to mitigate overconfidence. However, during inference, they still used the raw MSP as confidence estimator, without any normalization.

**Post-training logit normalization.** Here, we propose to use logit $p$-norm normalization as a post-hoc method and we intuitively expected it to have a similar effect in combating overconfidence. (Note that the argument in [Wei et al., 2022] holds unchanged for any $p$, as nothing in their analysis requires $p = 2$.) Our initial hypothesis was the following: if the model has become too overconfident (through high logit norm) on certain input regions, then—since overconfidence is a form of (loss) overfitting—there would be an increased chance that the model will produce incorrect predictions on the test set along these input regions. Thus, high logit norm on the test set would indicate regions of higher inaccuracy, so that, by applying logit normalization, we would be penalizing likely inaccurate predictions, improving selective classification performance. However, this hypothesis was *disproved* by the experimental results in Appendix E, which show that overconfidence is *not* necessarily a problem for selective classification, but *underconfidence* may be.

Nevertheless, it should be clear that, despite their similarities, logit L2 normalization during training and post-hoc logit $p$-norm normalization are different techniques applied to different problems and with different behavior. Moreover, even if logit normalization during training turns out to be beneficial to selective classification (an evaluation that is, however, outside the scope of this work), it should be emphasized that post-hoc optimization can be easily applied on top of any trained model without requiring modifications to its training regime.

**Combating underconfidence with temperature scaling.** If a model is underconfident on a set of samples, with low logit norm and an MSP value smaller than its expected accuracy on these samples, then the MSP may not provide a good estimate of confidence. One particular case of underconfidence is when the model incorrectly attributes too much posterior probability mass to the least probable classes (e.g., when all classes except the predicted one have the same probability). In this case, LogitsMargin (the margin between the highest and the second highest logit), which effectively disregards all logits except the highest two, may be a better confidence estimator. Alternatively, one may use MSP-TS with a low temperature, which approximates LogitsMargin, as can be easily seen below. Let $\mathbf{z} = (z_1, \ldots, z_C)$, with $z_1 > \ldots > z_C$. Then

$$\text{MSP}(\mathbf{z}/T) = \frac{e^{z_1/T}}{\sum_j e^{z_j/T}} = \frac{1}{1 + e^{(z_2-z_1)/T} + \sum_{j>2} e^{(z_j-z_1)/T}} \tag{12}$$

$$= \frac{1}{1 + e^{-(z_1-z_2)/T}\left(1 + \sum_{j>2} e^{-(z_2-z_j)/T}\right)} \approx \frac{1}{1 + e^{-(z_1-z_2)/T}} \tag{13}$$

for small $T > 0$. Note that a strictly increasing transformation does not change the ordering of confidence values and thus maintains selective classification performance. This helps explain why TS (with $T < 1$) can improve selective classification performance, as already observed in [Galil et al., 2023].

**Logit $p$-norm normalization as temperature scaling.** To shed light on why post-hoc logit $p$-norm normalization (with a general $p$) may be helpful, we can show that it is closely related to MSP-TS. Let $g_p(\mathbf{z}) = z_1/\|\mathbf{z}\|_p$ denote MaxLogit-pNorm without centralization, which we denote here as MaxLogit-pNorm-NC. Then

$$\text{MSP}(\mathbf{z}/T) = \left(\frac{e^{z_1}}{\left(\sum_j e^{z_j/T}\right)^T}\right)^{1/T} = \left(\frac{e^{z_1}}{\|e^{\mathbf{z}}\|_{1/T}}\right)^{1/T} = \left(g_{1/T}(e^{\mathbf{z}})\right)^{1/T}. \tag{14}$$

Thus, MSP-TS is equivalent to MaxLogit-pNorm-NC with $p = 1/T$ applied to the transformed logit vector $\exp(\mathbf{z})$. This helps explain why a general $p$-norm normalization is useful, as it is closely related to TS, emphasizing the largest components of the logit vector. This also implies that any benefits of MaxLogit-pNorm-NC over MSP-TS stem from *not* applying the exponential transformation of logits.

**Logit $p$-norm normalization goes beyond temperature scaling in combatting underconfidence.** To understand why not applying this exponential transformation is beneficial, we first express MaxLogit-pNorm-NC as

$$\text{MaxLogit-pNorm-NC}(\mathbf{z}) = \frac{z_1}{\left(\sum_{j=1}^{C} |z_j|^p\right)^{1/p}} = \frac{1}{\left(\sum_{j=1}^{C} \left|\frac{z_j}{z_1}\right|^p\right)^{1/p}} = \left(\frac{1}{1 + \left|\frac{z_2}{z_1}\right|^p + \sum_{j=3}^{C} \left|\frac{z_j}{z_1}\right|^p}\right)^{1/p} \tag{15}$$

where we assume $z_1 > 0$. Now, suppose that the logits already happen to be centralized (which also ensures $z_1 > 0$). It follows that most of the logits $z_j$ for $j \gg 1$ are close to zero (except possibly the very last ones). Thus, under the summation in (15), these logits effectively disappear, which is particularly useful in the case of underconfidence discussed above. However, this would not happen if an exponential transformation were applied to the logits as in (12), unless the $T$ is very

small. On the other hand, making $T$ too small can lead to ignoring not only the smallest logits but also some of the larger ones as well, i.e., it may be too drastic a measure. These effects are illustrated in Fig. 5.

This analysis also helps explain why centralization is useful. As shown in Appendix G, for most models, the logits are already centralized, so MaxLogit-pNorm-NC already provides the highest gains. A few models, however, have logits with means significantly different from zero and precisely these models achieve significant gains when centralization is applied, which enables the above analysis to hold.

In summary, underconfidence can be mitigated by prioritizing the largest logits. This is done MaxLogit-pNorm by increasing $p$ (which is akin to lowering the temperature), by making most of the smallest logits close to zero via centralization (if needed), and by *not* using an exponential transformation, which allows these near-zero logits to be effectively discarded without penalizing largest logits.

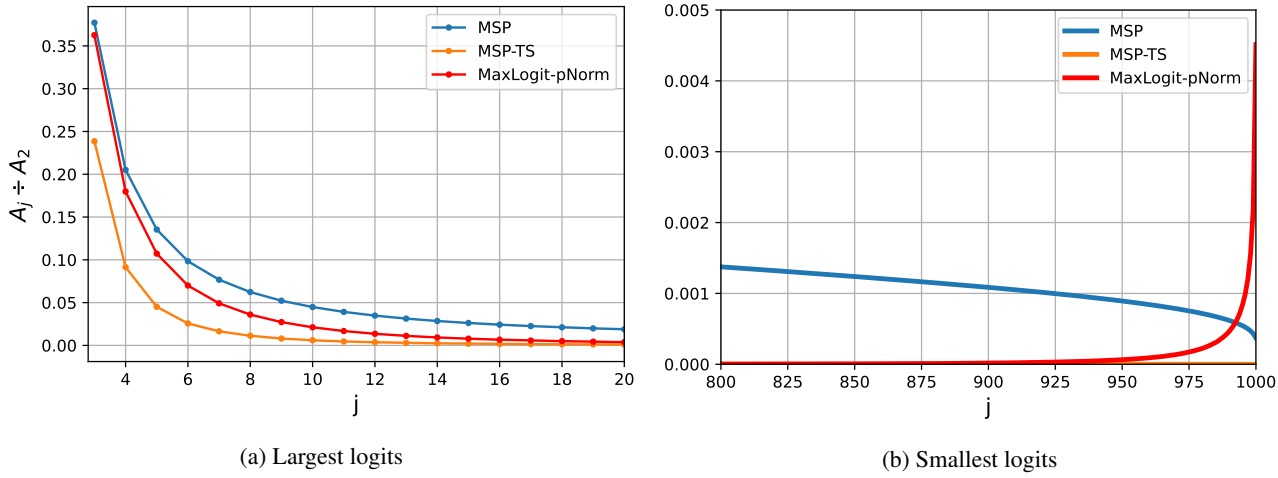

(a) Largest logits  (b) Smallest logits

Figure 5: The ratio $A_j/A_2$, where $A_j = \exp(z_j/T)$ for MSP-TS and $A_j = |z_j - \mu(\mathbf{z})|^p$ for MaxLogit-pNorm, hence reflecting the influence of intermediate logits on Equations 12 and 15. The classifier is EfficientNet-B3 evaluated on ImageNet. The sum $\sum_{j \geq 100} A_j/A_2$ is equal to 2.020 for the MSP, 0.005 for the MSP-TS and 0.024 for the MaxLogit-pNorm, showing the effectiveness of the latter two methods in discarding the smallest logits.

# C  MORE DETAILS AND RESULTS ON THE EXPERIMENTS ON IMAGENET

## C.1  HYPERPARAMETER OPTIMIZATION OF POST-HOC METHODS

Because it is not differentiable, the NAURC metric demands a zero-order optimization. For this work, the optimizations of $p$ and $T$ were conducted via grid-search. Note that, as $p$ approaches infinity, $||\mathbf{z}||_p \to \max(|\mathbf{z}|)$. Indeed, it tends to converge reasonable quickly. Thus, the grid search on $p$ can be made only for small $p$. In our experiments, we noticed that it suffices to evaluate a few values of $p$, such as the integers between 0 and 10, where the 0-norm is taken here to mean the sum of all nonzero values of the vector. The temperature values were taken from the range between 0.01 and 3, with a step size of 0.01, as this showed to be sufficient for achieving the optimal temperature for selective classification (in general between 0 and 1).

## C.2  AUROC RESULTS

Table 4 shows the AUROC results for all methods for an EfficientNetV2-XL and a VGG-16 on ImageNet, and Figure 6 shows the correlation between the AUROC and the accuracy. As it can be seen, the results are consistent with the ones for NAURC presented in Section 5.

## C.3  RC CURVES

In Figure 7 the RC curves of selected post-hoc methods applied to a few representative models are shown.

Table 4: AUROC (mean ±std) for post-hoc methods applied to ImageNet classifiers

| Classifier | Conf. Estimator | Logit Transformation | | | |
| | | Raw | TS-NLL | TS-AURC | pNorm |
|---|---|---|---|---|---|
| EfficientNet-V2-XL | MSP | 0.7732 ±0.0014 | 0.8107 ±0.0016 | 0.8606 ±0.0011 | 0.8712 ±0.0012 |
| | SoftmaxMargin | 0.7990 ±0.0013 | 0.8245 ±0.0014 | 0.8603 ±0.0012 | 0.8712 ±0.0011 |
| | MaxLogit | 0.6346 ±0.0014 | - | - | **0.8740** ±0.0010 |
| | LogitsMargin | 0.8604 ±0.0011 | - | - | 0.8702 ±0.0010 |
| | NegativeEntropy | 0.6890 ±0.0014 | 0.7704 ±0.0026 | 0.6829 ±0.0891 | 0.8719 ±0.0016 |
| | NegativeGini | 0.7668 ±0.0014 | 0.8099 ±0.0017 | 0.8606 ±0.0011 | 0.8714 ±0.0012 |
| VGG16 | MSP | 0.8660 ±0.0004 | 0.8652 ±0.0003 | 0.8661 ±0.0004 | 0.8661 ±0.0004 |
| | SoftmaxMargin | 0.8602 ±0.0003 | 0.8609 ±0.0004 | 0.8616 ±0.0003 | 0.8616 ±0.0003 |
| | MaxLogit | 0.7883 ±0.0004 | - | - | 0.8552 ±0.0007 |
| | LogitsMargin | 0.8476 ±0.0003 | - | - | 0.8476 ±0.0003 |
| | NegativeEntropy | 0.8555 ±0.0004 | 0.8493 ±0.0004 | 0.8657 ±0.0004 | 0.8657 ±0.0004 |
| | NegativeGini | 0.8645 ±0.0004 | 0.8620 ±0.0003 | 0.8659 ±0.0003 | 0.8659 ±0.0003 |

## C.4 EFFECT OF $\epsilon$

Figure 8 shows the results (in APG metric) for all methods when $p$ is optimized. As can be seen, MaxLogit-pNorm is dominant for all $\epsilon > 0$, indicating that, provided the MSP fallback described in Section 4.2.2 is enabled, it outperforms the other methods.

# D   EXPERIMENTS ON ADDITIONAL DATASETS

## D.1   EXPERIMENTS ON OXFORD-IIIT PET

The hold-out set for Oxford-IIIT Pet, consisting of 500 samples, was taken from the training set before training. The model used was an EfficientNet-V2-XL pretrained on ImageNet from Wightman [2019]. It was fine-tuned on Oxford-IIIT Pet [Parkhi et al., 2012]. The training was conducted for 100 epochs with Cross Entropy Loss, using a SGD optimizer with initial learning rate of 0.1 and a Cosine Annealing learning rate schedule with period 100. Moreover, a weight decay of 0.0005 and a Nesterov's momentum of 0.9 were used. Data transformations were applied, specifically standardization, random crop (for size 224x224) and random horizontal flip.

Figure 9 shows the RC curves for some selected methods for the EfficientNet-V2-XL. As can be seen, considerable gains are obtained with the optimization of $p$, especially in the low-risk region.

## D.2   EXPERIMENTS ON CIFAR-100

The hold-out set for CIFAR-100, consisting of 5000 samples, was taken from the training set before training. The model used was forked from `github.com/kuangliu/pytorch-cifar`, and adapted for CIFAR-100 [Krizhevsky, 2009]. It was trained for 200 epochs with Cross Entropy Loss, using a SGD optimizer with initial learning rate of 0.1 and a Cosine Annealing learning rate schedule with period 200. Moreover, a weight decay of 0.0005 and a Nesterov's momentum of 0.9 were used. Data transformations were applied, specifically standardization, random crop (for size 32x32 with padding 4) and random horizontal flip.

Figure 10 shows the RC curves for some selected methods for a VGG19. As it can be seen, the results follow the same pattern of the ones observed for ImageNet, with MaxLogit-pNorm achieving the best results.

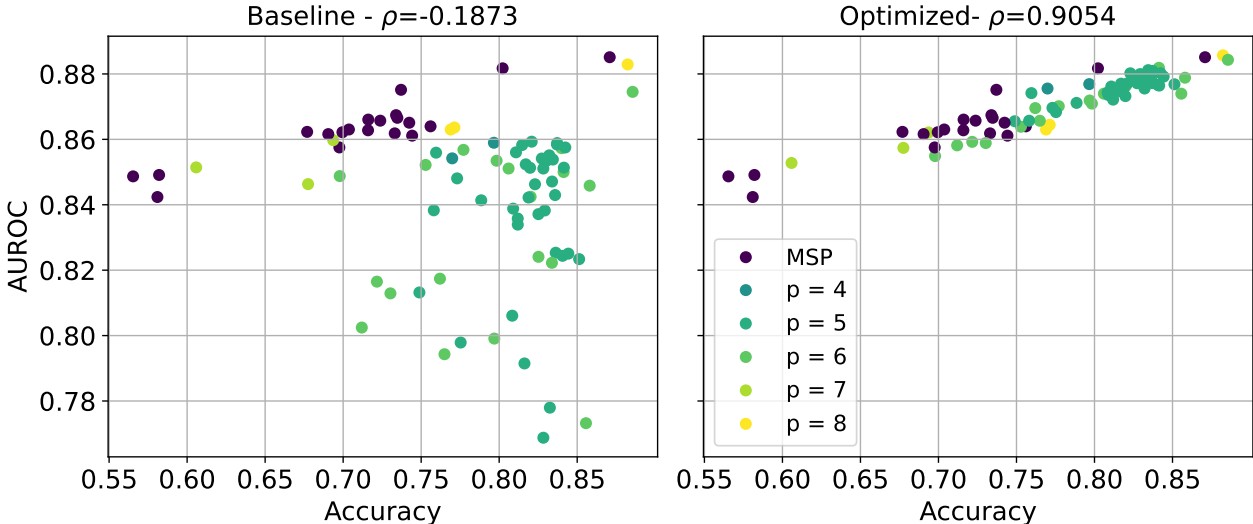

Figure 6: AUROC of 84 ImageNet classifiers with respect to their accuracy, before and after post-hoc optimization. The baseline plots use MSP, while the optimized plots use MaxLogit-pNorm. The legend shows the optimal value of $p$ for each model, where MSP indicates MSP fallback (no significant positive gain). $\rho$ is the Spearman correlation between the AUROC and the accuracy.

## E DATA EFFICIENCY

In this section, we empirically investigate the *data efficiency* [Zhang et al., 2020] of tunable post-hoc methods, which refers to their ability to learn and generalize from limited data. As is well-known from machine learning theory and practice, the more we evaluate the empirical risk to tune a parameter, the more we are prone to overfitting, which is aggravated as the size of the dataset used for tuning decreases. Thus, a method that require less hyperparameter tuning tends to be more data efficient, i.e., to achieve its optimal performance with less tuning data. We intuitively expect this to be the case for MaxLogit-pNorm, which only requires evaluating a few values of $p$, compared to any method based on the softmax function, which requires tuning a temperature parameter.

As mentioned in Section 5, the experiments conducted in ImageNet used a test set of 45000 images randomly sampled from the available ImageNet validation dataset, resulting in 5000 images for the tuning set. To evaluate data efficiency, the post-hoc optimization process was executed multiple times, using different fractions of the tuning set while keeping the test set fixed. This whole process was repeated 50 times for different random samplings of the test set (always fixed at 45000 images).

Figure 11a displays the outcomes of these studies for a ResNet50 trained on ImageNet. As observed, MaxLogit-pNorm exhibits outstanding data efficiency, while methods that require temperature optimization achieve lower efficiency.

Furthermore, this experiment was conducted on the VGG19 model for CIFAR-100, as shown in figure 11a. Indeed, the same conclusions hold for the high efficiency of MaxLogit-pNorm.

To ensure our finding generalize across models, we repeat this process for all the 84 ImageNet classifiers considered, for a specific tuning set size. This time only 10 realizations of the test set were performed, similarly to the results of Section 5.1. Table 5 is the equivalent of Table 2 for a tuning set of 1000 samples, while Table 6 corresponds to a tuning set of 500 samples. As can be seen, the results obtained are consistent with those observed previously. In particular, MaxLogit-pNorm provides a statistically significant improvement over all other methods when the tuning set is reduced. Moreover, MaxLogit-pNorm is one of the most stable among the tunable methods in terms of variance of gains.

## F ABLATION STUDY ON THE CHOICE OF $p$

A natural question regarding $p$-norm normalization (with a general $p$) is whether it can provide any benefits beyond the default $p = 2$ used by Wei et al. [2022]. Table 7 shows the APG-NAURC results for the 84 ImageNet classifiers when different values of $p$ are kept fixed and when $p$ is optimized for each model (tunable).

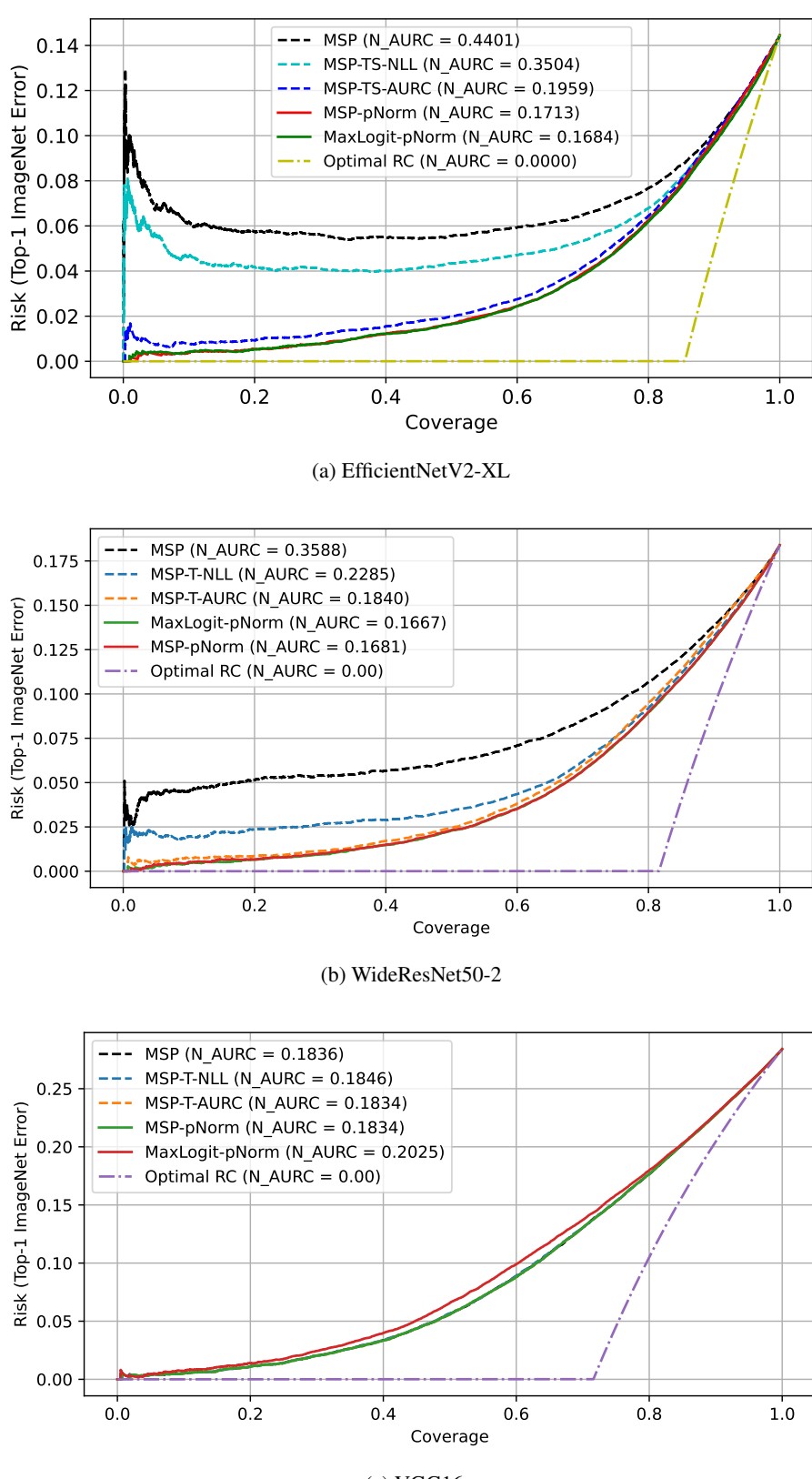

(a) EfficientNetV2-XL

(b) WideResNet50-2

(c) VGG16

Figure 7: RC curves for selected post-hoc methods applied to ImageNet classifiers.

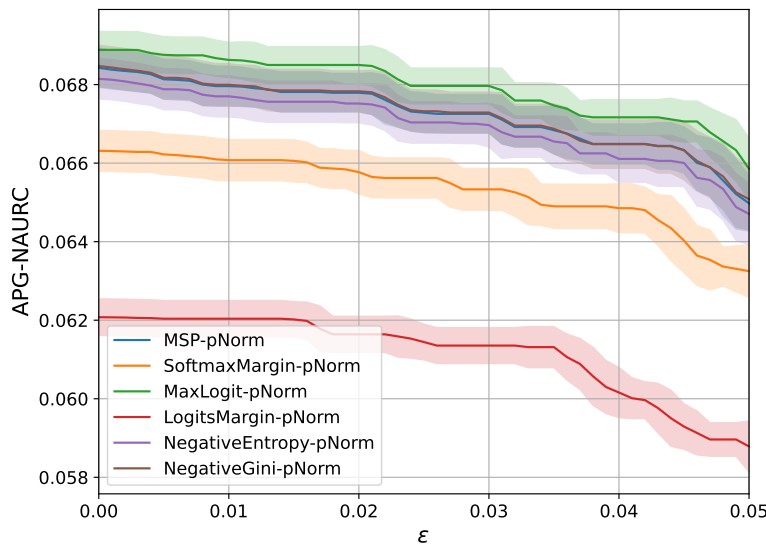

Figure 8: APG as a function of $\epsilon$

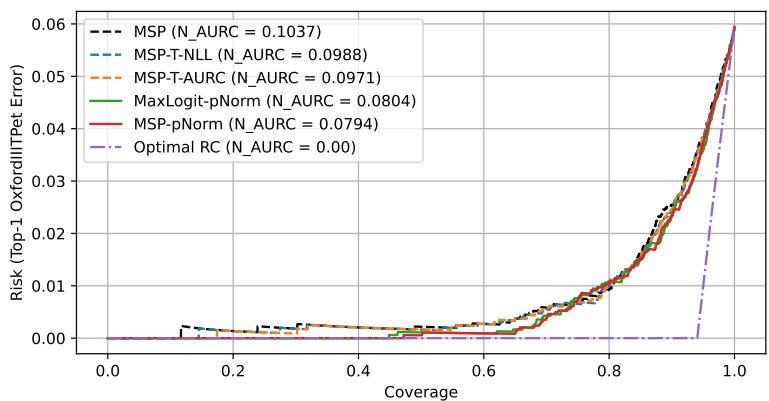

Figure 9: RC curves for a EfficientNet-V2-XL for Oxford-IIIT Pet

As can be seen, there is a significant benefit of using a larger $p$ (especially a tunable one) compared to simply using $p = 2$, especially for MaxLogit-pNorm. Note that, differently from MaxLogit-pNorm, MSP-pNorm requires temperature optimization. This additional tuning is detrimental to data efficiency, which is evidenced by the loss in performance of MSP-pNorm using a tuning set of 1000 samples, as shown in Table 8.

# G   LOGITS TRANSLATION

In Section 4.1 we proposed $p$-normalization applied together with the centralization of the logits. In this section, we aim to provide an ablation of this centralization procedure and the effects of the translation of logits.

First of all, it is worth noting that the softmax function is translation invariant, i.e.,

$$\sigma(\mathbf{z}) = \sigma(\mathbf{z} + \gamma) \quad \forall \gamma \in \mathbb{R}. \tag{16}$$

As the general loss (i.e. the cross-entropy loss) takes as input only the softmax outputs, the logits after convergence might have arbitrarily mean/offsets. Moreover, the following properties become relevant when dealing with selective classification:

- All methods in which the $p$-normalization is applied on the logits are sensitive to any constant summed up to the them;

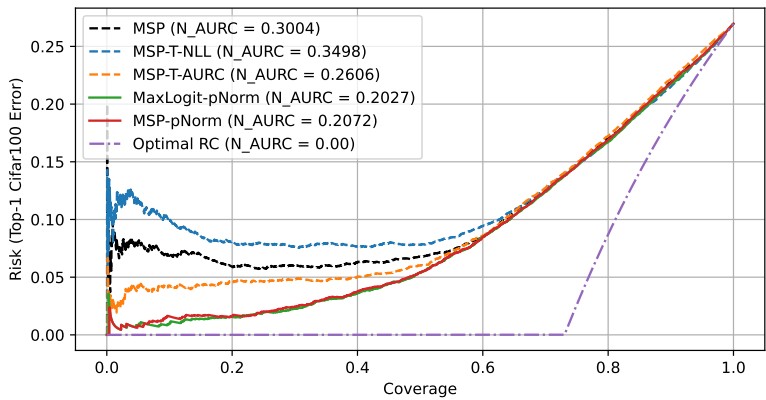

Figure 10: RC curves for a VGG19 for CIFAR-100

Table 5: APG-NAURC (mean ±std) of post-hoc methods across 84 ImageNet classifiers, for a tuning set of 1000 samples

| | Logit Transformation | | | |
|---|---|---|---|---|
| Conf. Estimator | Raw | TS-NLL | TS-AURC | pNorm |
| MSP | 0.00000 ±0.00000 | 0.03657 ±0.00084 | 0.05571 ±0.00164 | 0.06436 ±0.00413 |
| SoftmaxMargin | 0.01951 ±0.00010 | 0.04102 ±0.00052 | 0.05420 ±0.00134 | 0.06238 ±0.00416 |
| MaxLogit | 0.00000 ±0.00000 | - | - | **0.06795** ±0.00077 |
| LogitsMargin | 0.05510 ±0.00059 | - | - | 0.06110 ±0.00084 |
| NegativeEntropy | 0.00000 ±0.00000 | 0.01566 ±0.00182 | 0.05851 ±0.00055 | 0.06485 ±0.00176 |
| NegativeGini | 0.00000 ±0.00000 | 0.03627 ±0.00095 | 0.05617 ±0.00162 | 0.06424 ±0.00390 |

- The sum of the same constant for all samples does not change the ranking between them when the MaxLogit is used as the confidence estimator. However, when a constant different for each sample (such as the centralization) is considered, the ranking might be affected;

- The LogitsMargin is totally insensitive to the translation of the logits;

- All methods using softmax *without* $p$-normalization are insensitive to the translation of the logits.

In order to study the impact of the translation of logits on the MaxLogit-pNorm method, we will start by proposing an alternative post-hoc method:

$$\text{MaxLogit-pNorm-shift}(\mathbf{z}, \Gamma, \gamma) \triangleq \text{MaxLogit}\left(\frac{\mathbf{z} - \Gamma(\mathbf{z}) + \gamma}{||\mathbf{z} - \Gamma(\mathbf{z}) + \gamma||_p}\right), \tag{17}$$

where $\Gamma \colon \mathbb{R}^C \to \mathbb{R}$ is a function of the logits (such as the mean function) and $\gamma \in \mathbb{R}$ is a constant to be optimized together with $p$. The optimization of $\gamma$ is performed with a grid search in the range of [-3,3].

Table 9 shows the APG-NAURC for all 84 models considered in this work on ImageNet when using different possibilites of $\Gamma$ and $\gamma$. Specifically, we considered the cases where $\gamma$ is 0 and when it is optimized in a hold-out set; for $\Gamma$, we considered $\Gamma(\mathbf{z}) = 0$, $\Gamma(\mathbf{z}) = \mu(\mathbf{z})$ (for centralization) and $\Gamma(\mathbf{z}) = \min_j z_j$ (to align the minimum value of all samples to 0, making all logits positive). As can be seen, optimizing $\gamma$ does not provide significant gains and can lead to overfitting in a low data regime; thus, in the main method we discarded this constant. For $\gamma = 0$, choosing $\Gamma(\mathbf{z}) = \mu(\mathbf{z})$ provides the highest gains, which, although relatively small compared to $\Gamma(\mathbf{z}) = 0$, certainly do not harm performance. Since this operation is computationally cheap, does not require optimization, and allows the use of softmax probabilities directly (as mentioned in Section 4.1), we decided to adopt it in the main method.

Figure 12 shows the difference in NAURC when $\Gamma(\mathbf{z}) = \mu(\mathbf{z})$ and when $\Gamma(\mathbf{z}) = 0$ (for $\gamma = 0$), as well as the average across all test samples of the mean of the logits for all methods in which the MaxLogit-pNorm wields gains (i.e., the MSP fallback is not applied). It can be observed that most models already output their logits with almost zero mean, making centralization unnecessary. However, a few models with nonzero logits means present considerable gains in centralization.

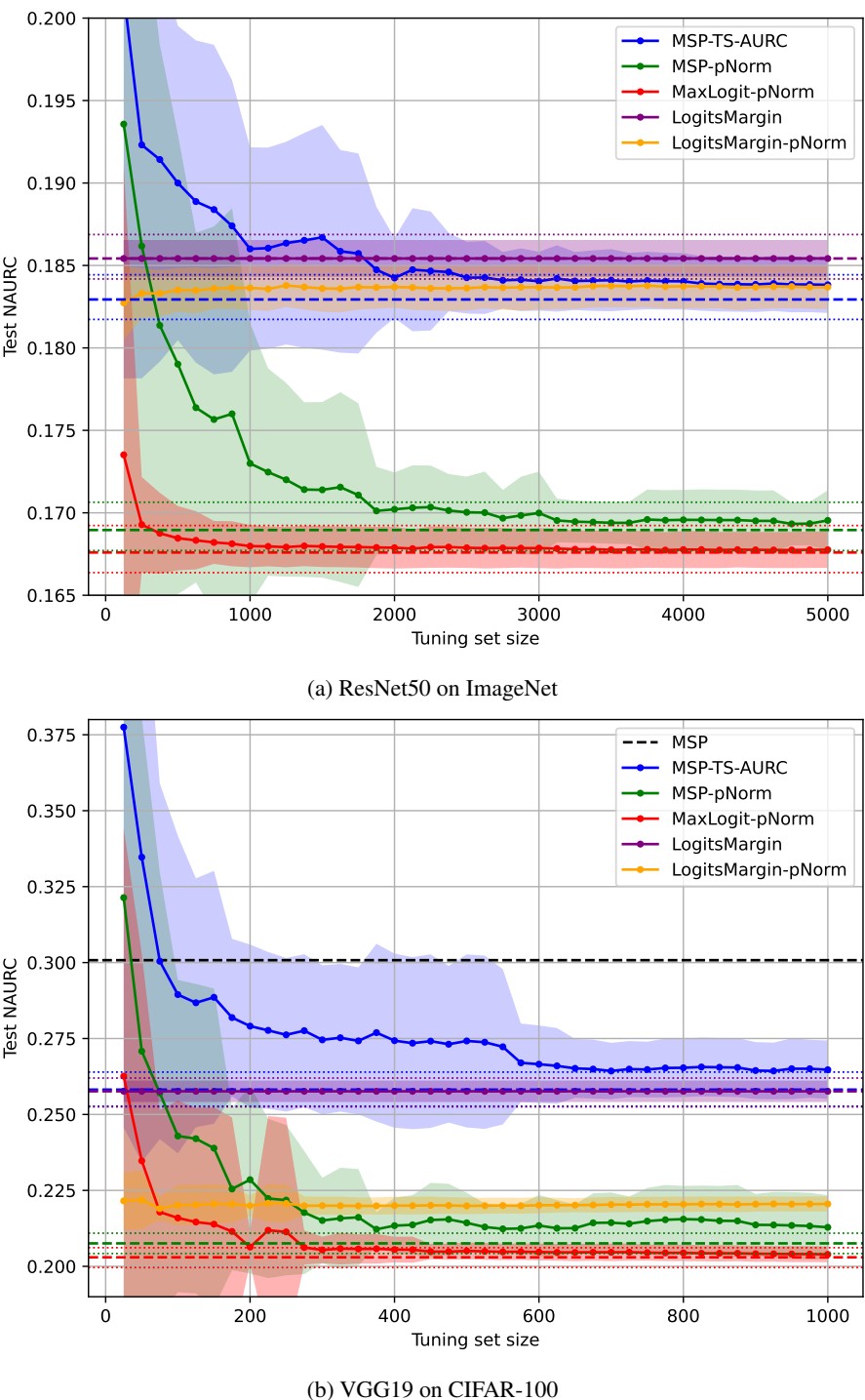

(a) ResNet50 on ImageNet

(b) VGG19 on CIFAR-100

Figure 11: Mean NAURC as a function of the number of samples used for tuning the confidence estimator. Filled regions for each curve correspond to ±1 standard deviation (across 50 realizations). Dashed lines represent the mean of the NAURC achieved when the optimization is made directly on the test set (giving a lower bound on the optimal value), while dotted lines correspond respectively to ±1 standard deviation. (a) ResNet50 on ImageNet. For comparison, the MSP achieves a mean NAURC of 0.3209 (not shown in the figure). (b) VGG19 on CIFAR-100.

Table 6: APG-NAURC (mean ±std) of post-hoc methods across 84 ImageNet classifiers, for a tuning set of 500 samples

| Conf. Estimator | Logit Transformation | | | |
| --- | --- | --- | --- | --- |
| | Raw | TS-NLL | TS-AURC | pNorm |
| MSP | 0.0 ± 0.0 | 0.03614 ±0.00152 | 0.05198 ±0.00381 | 0.05835 ±0.00677 |
| SoftmaxMargin | 0.01955 ±0.00008 | 0.04083 ±0.00094 | 0.05048 ±0.00381 | 0.05601 ±0.00683 |
| MaxLogit | 0.0 ± 0.0 | - | - | **0.06719** ±0.00141 |
| LogitsMargin | 0.05531 ±0.00042 | - | - | 0.06064 ±0.00081 |
| NegativeEntropy | 0.0 ± 0.0 | 0.01487 ±0.00266 | 0.05808 ±0.00066 | 0.06270 ±0.00223 |
| NegativeGini | 0.0 ± 0.0 | 0.03578 ±0.00174 | 0.05250 ±0.00368 | 0.05832 ±0.00656 |

Table 7: APG-NAURC (mean ±std) across 84 ImageNet classifiers, for different values of $p$

| | Confidence Estimator | |
| --- | --- | --- |
| $p$ | MaxLogit-pNorm | MSP-pNorm |
| 0 | 0.00000 ±0.00000 | 0.05769 ±0.00038 |
| 1 | 0.00199 ±0.00007 | 0.05990 ±0.00062 |
| 2 | 0.01519 ±0.00050 | 0.06486 ±0.00054 |
| 3 | 0.05058 ±0.00049 | 0.06748 ±0.00048 |
| 4 | 0.06443 ±0.00051 | 0.06823 ±0.00047 |
| 5 | 0.06805 ±0.00048 | 0.06809 ±0.00048 |
| 6 | 0.06814 ±0.00048 | 0.06763 ±0.00049 |
| 7 | 0.06692 ±0.00053 | 0.06727 ±0.00048 |
| 8 | 0.06544 ±0.00048 | 0.06703 ±0.00048 |
| 9 | 0.06410 ±0.00048 | 0.06690 ±0.00048 |
| Tunable | **0.06863** ±0.00045 | 0.06796 ±0.00051 |

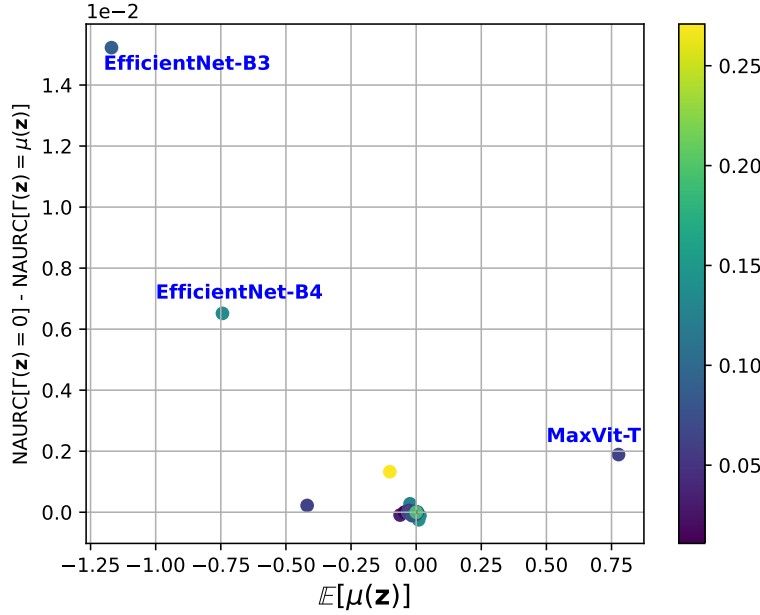

Figure 12: Gains in NAURC when the centralization is applied to the logits in relation to the average of all logits in the test dataset. Colors represent the gain of MaxLogit-pNorm over MSP.

## H COMPARISON WITH OTHER TUNABLE METHODS

In Section 5.1 we compared several logit-based confidence estimators obtained by combining a parameterless confidence estimator with a tunable logit transformation, specifically, TS and $p$-norm normalization. In this section, we consider other

Table 8: APG-NAURC (mean ±std) across 84 ImageNet classifiers, for different values of $p$ for a tuning set of 1000 samples

| | Confidence Estimator | |
| :--- | :--- | :--- |
| $p$ | MaxLogit-pNorm | MSP-pNorm |
| 0 | 0.00000 ±0.00000 | 0.05571 ±0.00164 |
| 1 | 0.00199 ±0.00007 | 0.05699 ±0.00365 |
| 2 | 0.01519 ±0.00050 | 0.06234 ±0.00329 |
| 3 | 0.05058 ±0.00049 | 0.06527 ±0.00340 |
| 4 | 0.06443 ±0.00051 | 0.06621 ±0.00375 |
| 5 | 0.06805 ±0.00048 | 0.06625 ±0.00338 |
| 6 | **0.06814** ±0.00048 | 0.06589 ±0.00332 |
| 7 | 0.06692 ±0.00053 | 0.06551 ±0.00318 |
| 8 | 0.06544 ±0.00048 | 0.06512 ±0.00345 |
| 9 | 0.06410 ±0.00048 | 0.06491 ±0.00329 |
| Tunable | 0.06795 ±0.00077 | 0.06436 ±0.00413 |

Table 9: APG-NAURC (mean ±std) of MaxLogit-pNorm-shift for different selections of $\Gamma$ and $\gamma$. $\gamma^*$ represents the value that optimizes the AURC in the hold-out dataset.

| | 5000 hold-out samples | | 1000 hold-out samples | |
| :--- | :--- | :--- | :--- | :--- |
| $\Gamma(\mathbf{z})$ | $\gamma = 0$ | $\gamma = \gamma^*$ | $\gamma = 0$ | $\gamma = \gamma^*$ |
| 0 | 0.06833 ±0.00044 | 0.06866 ±0.00044 | 0.06760 ±0.00077 | 0.06738 ±0.00091 |
| $\mu(\mathbf{z})$ | **0.06863** ±0.00045 | **0.06867** ±0.00045 | **0.06795** ±0.00077 | **0.06742** ±0.00093 |
| $\min_j z_j$ | 0.06668 ±0.00049 | 0.06658 ±0.00056 | 0.06626 ±0.00073 | 0.06523 ±0.00151 |

previously proposed tunable confidence estimators that do not fit into this framework.

Note that some of these methods were originally proposed seeking calibration, and hence its hyperparameters were tuned to optimize the NLL loss (which is usually suboptimal for selective classification). Instead, to make a fair comparison, we optimized all of their parameters using the AURC metric as the objective metric.

Zhang et al. [2020] proposed *ensemble temperature scaling* (ETS):

$$\text{ETS}(\mathbf{z}) \triangleq w_1 \text{MSP}\left(\frac{\mathbf{z}}{T}\right) + w_2 \text{MSP}(\mathbf{z}) + w_3 \frac{1}{C} \quad (18)$$

where $w_1, w_2, w_3 \in \mathbb{R}^+$ are tunable parameters and $T$ is the temperature previously obtained through the temperature scaling method. The grid for both $w_1$ and $w_2$ was $[0, 1]$ as suggested by the authors, with a step size of 0.01, while the parameter $w_3$ was not considered since the sum of a constant to the confidence estimator cannot change the ranking between samples and consequently cannot change the value of selective classification metrics.

Boursinos and Koutsoukos [2022] proposed the following confidence estimator, referred to here as *Boursinos-Koutsoukos* (BK):

$$\text{BK}(\mathbf{z}) \triangleq a\text{MSP}(\mathbf{z}) + b(1 - \max_{k \in \mathcal{Y}: k \neq \hat{y}} \sigma_k(\mathbf{z})) \quad (19)$$

where $a, b \in \mathbb{R}$ are tunable parameters. The grid for both $a$ and $b$ was $[-1, 1]$ as suggested by the authors, with a step size of 0.01, although we note that the optimization never found $a < 0$ (probably due to the high value of the MSP as a confidence estimator).

Finally, Balanya et al. [2023] proposed *entropy-based temperature scaling* (HTS):

$$\text{HTS}(\mathbf{z}) \triangleq \text{MSP}\left(\frac{\mathbf{z}}{T_H(\mathbf{z})}\right) \quad (20)$$

where $T_H(\mathbf{z}) = \log\left(1 + \exp(b + w \log \bar{H}(\mathbf{z}))\right)$, $\bar{H}(\mathbf{z}) = -(1/C) \sum_{k \in \mathcal{Y}} \sigma_k(\mathbf{z}) \log \sigma_k(\mathbf{z})$, and $b, w \in \mathbb{R}$ are tunable parameters. The grids for $b$ and $w$ were, respectively, $[-3, 1]$ and $[-1, 1]$, with a step size of 0.01, and we note that the optimal parameters were always strictly inside the grid.

The results for these post-hoc methods are shown in Table 10 and Table 11. Interestingly, BK, which can be seen as a tunable linear combination of MSP and SoftmaxMargin, is able to outperform both of them, although it still underperforms MSP-TS. On the other hand, ETS, which is a tunable linear combination of MSP and MSP-TS, attains exactly the same performance as MSP-TS. Finally, HTS, which is a generalization of MSP-TS, is able to outperform it, although it still underperforms most methods that use $p$-norm tuning (see Table 2). In particular, MaxLogit-pNorm shows superior performance to all of these methods, while requiring much less hyperparameter tuning.

Table 10: APG-NAURC of additional tunable post-hoc methods across 84 ImageNet classifiers

| Method | APG-NAURC |
| --- | --- |
| BK | 0.03932 ±0.00031 |
| ETS | 0.05768 ±0.00037 |
| HTS | 0.06309 ±0.00034 |
| MaxLogit-pNorm | **0.06863** ±0.00045 |

Table 11: APG-NAURC of additional tunable post-hoc methods across 84 ImageNet classifiers for a tuning set with 1000 samples

| Method | APG-NAURC |
| --- | --- |
| BK | 0.03795 ±0.00067 |
| ETS | 0.05569 ±0.00165 |
| HTS | 0.05927 ±0.00280 |
| MaxLogit-pNorm | **0.06795** ±0.00077 |

Methods with a larger number of tunable parameters, such as PTS [Tomani et al., 2022] and HnLTS [Balanya et al., 2023], are only viable with a differentiable loss. As these methods are proposed for calibration, the NLL loss is used; however, as previous works have shown that this does not always improve and sometimes even harm selective classification [Zhu et al., 2022, Galil et al., 2023], these methods were not considered in our work. The investigation of alternative methods for optimizing selective classification (such as proposing differentiable losses or more efficient zero-order methods) is left as a suggestion for future work. In any case, note that using a large number of hyperparameters is likely to harm data efficiency.

We also evaluated additional parameterless confidence estimators proposed for selective classification [Hasan et al., 2023], such as LDAM [He et al., 2011] and the method in [Leon-Malpartida et al., 2018], both in their raw form and with TS/pNorm optimization, but none of these methods showed any gain over the MSP. Note that the Gini index, sometimes proposed as a post-hoc method [Hasan et al., 2023] (and also known as DOCTOR's $D_\alpha$ method [Granese et al., 2021]) has already been covered in Section 3.2.

## I  CALIBRATION RESULTS

If the confidence estimation $g(x)$ of a model can be treated as a probability, as is the case with the MSP, it is natural to desire that it truly reflects the probability of a prediction to be correct. A model is said to be perfectly *calibrated* if:

$$\mathbb{P}[\hat{y} = y | g(x) = p] = p, \forall p \in [0, 1] \tag{21}$$

One popular framework to measure calibration in a finite dataset is to use binning. If we group predictions into $M$ interval bins with same size, and if $B_m$ is a set of indices of samples whose prediction confidence belongs to the interval $\left(\frac{m-1}{M}, \frac{m}{M}\right]$, we calculate the accuracy of bin $B_m$ as:

$$\text{acc}(B_m) = \frac{1}{|B_m|} \sum_{i \in B_m} \mathbb{1}[\hat{y}_i = y_i] \tag{22}$$

where $\hat{y}_i$ and $y_i$ are the predicted and the true classes of sample $i$ and $|B_m|$ is the number of samples in the bin. The average confidence of the same bin is calculated as:

$$\text{conf}(B_m) = \frac{1}{|B_m|} \sum_{i \in B_m} g_i(x) \tag{23}$$

From these definitions, the most popular metric for measuring the calibration is the Expected Calibration Error [Naeini et al., 2015], defined as:

$$\text{ECE}(g) \triangleq \sum_{m=1}^{M} \frac{|B_m|}{n} \left| \text{acc}(B_m) - conf(B_m) \right| \tag{24}$$

It is important to re-emphasize that calibration and metrics such as ECE are defined in a context where $g(x)$ can be treated as a probability. Hence, we only present the results for uncertainty quantifiers that have this property/intention. The ECE values for all considered methods (optimized for the AURC) for which $g(x)$ can be considered as a probability are presented in Table 12. Additionally, Figure 13 shows the reliability diagrams [Guo et al., 2017] of different classifiers of ImageNet. For comparison, since MaxLogit-pNorm can only return values between 0 and 1, we also present its reliability curve in Figure 13, even though its values should not be interpreted as a probability. As can be seen, the models (EfficientNetV2-XL and WideResNet50-2) with "broken" selective mechanism tend to have the MSP under-confident, and, while the TS-NLL can minimize the ECE, the MSP variation which optimizes selective classification (MSP-pNorm) can achieve bad calibration results, with overconfident predictions.

Table 12: ECE (mean ±std) for post-hoc methods applied to ImageNet classifiers

| Method | ECE |
|---|---|
| MSP | 0.13060 ±0.00014 |
| MSP-TS-NLL | **0.02990** ±0.00109 |
| MSP-TS-AURC | 0.10395 ±0.00341 |
| MSP-pNorm | 0.10786 ±0.04860 |

These results goes against the natural hypothesis that overconfidence is a huge problem in uncertainty estimation of neural networks. Thus, we present further investigations regarding the relation between the selective classification anomaly and the over/underconfidence phenomenon. Figure 14 shows histograms of confidence values for two representative examples of non-improvable and improvable models, with the latter one shown before and after post-hoc optimization. Figure 15 shows the NAURC gain over MSP versus the proportion of samples with high MSP for each classifier. As can be seen, highly confident models tend to have a good MSP confidence estimator, while less confident models tend to have a poor confidence estimator that is easily improvable by post-hoc methods—after which the resulting confidence estimator becomes concentrated on high values.

## J  FULL RESULTS ON IMAGENET

Table 13 presents all the NAURC results for the most relevant methods for all the models evaluated on ImageNet, while Table 14 shows the corresponding AURC results and Table 15 the corresponding AUROC results. $p^*$ denotes the optimal value of $p$ obtained for the corresponding method, while $p^* = F$ denotes MSP fallback.

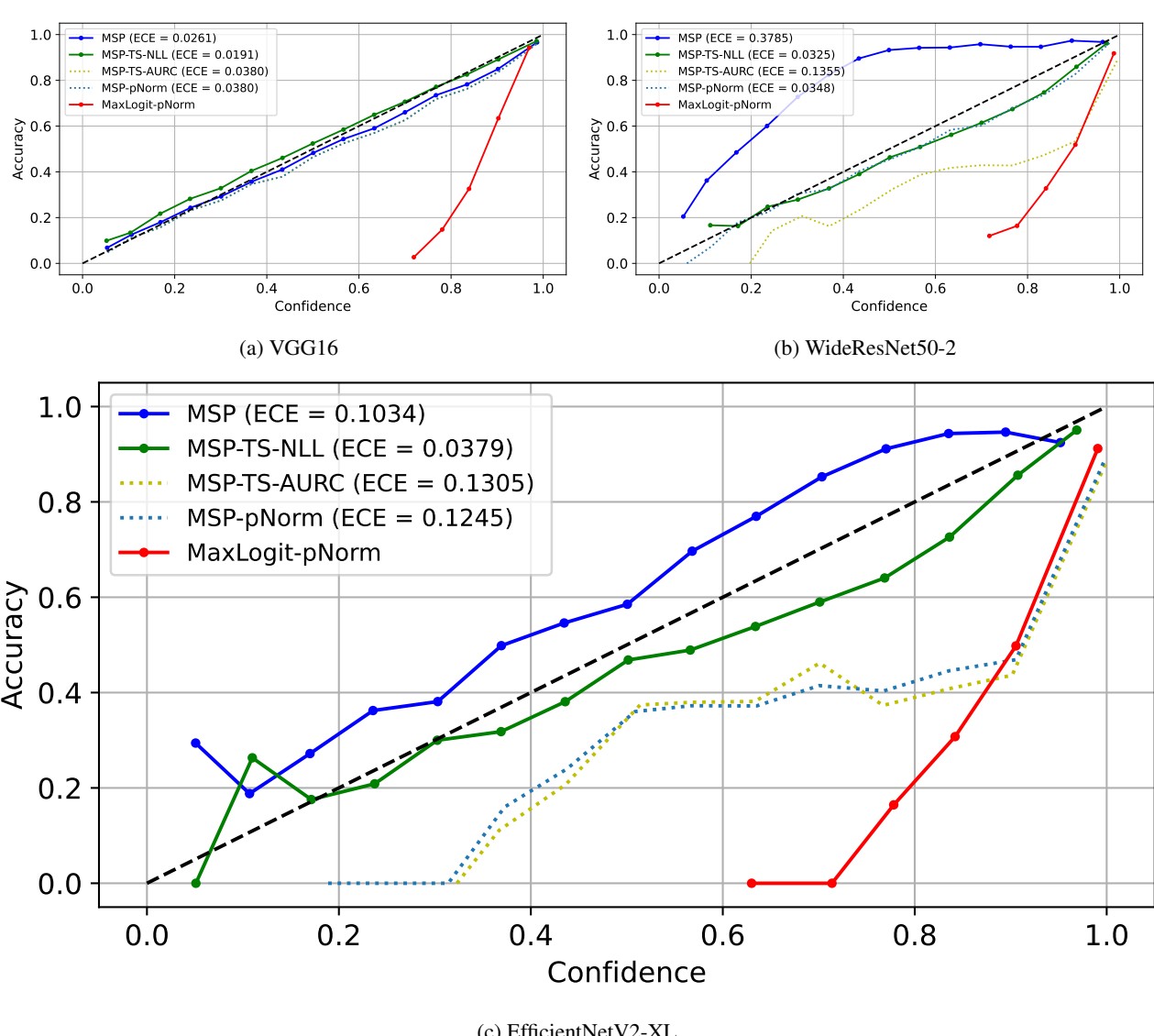

(a) VGG16

(b) WideResNet50-2

(c) EfficientNetV2-XL

Figure 13: Reliability diagrams of different methods applied on VGG16, WideResNet50-2 and EfficientNetV2-XL on ImageNet. Dashed black line indicates perfect calibration. For MaxLogit-pNorm, we do not present the ECE metric since this method is not treated as a probability.

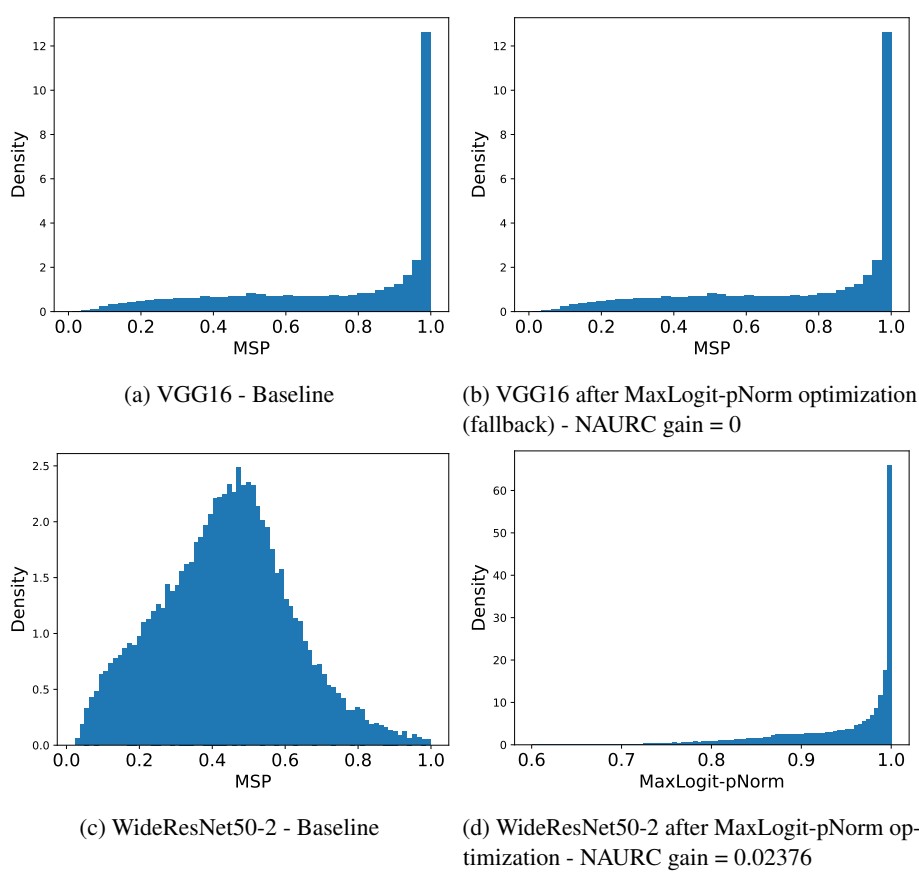

(a) VGG16 - Baseline

(b) VGG16 after MaxLogit-pNorm optimization (fallback) - NAURC gain = 0

(c) WideResNet50-2 - Baseline

(d) WideResNet50-2 after MaxLogit-pNorm optimization - NAURC gain = 0.02376

Figure 14: Histograms of confidence values for VGG16 and WideResNet50-2 before and after post-hoc optimization on ImageNet.

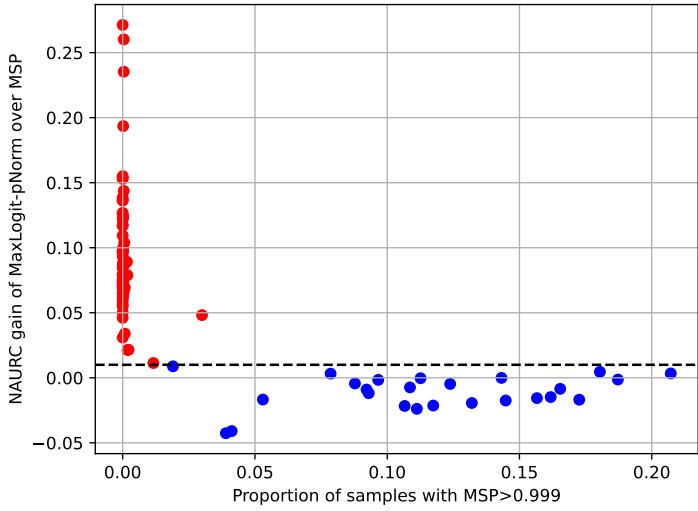

Figure 15: NAURC gain versus the proportion of samples with MSP > 0.999.

Table 13: NAURC (mean ±std) for all models evaluated on ImageNet

| | | Method | | | | | |
|---|---|---|---|---|---|---|---|
| Model | Accuracy[%] | MSP | MSP-TS-NLL | MSP-TS-AURC | LogitsMargin | MSP-pNorm ($p^*$) | MaxLogit-pNorm ($p^*$) |
| alexnet | 0.5654 ± 0.0007 | 0.2229 ± 0.0007 | 0.2245 ± 0.0007 | 0.2226 ± 0.0008 | 0.2593 ± 0.0007 | 0.2226 ± 0.0008 (0) | 0.2402 ± 0.0007 (F) |
| convnext_base | 0.8406 ± 0.0004 | 0.3006 ± 0.0025 | 0.2324 ± 0.0024 | 0.1799 ± 0.0028 | 0.1795 ± 0.0015 | 0.1629 ± 0.0013 (4) | 0.1623 ± 0.0013 (5) |
| convnext_large | 0.8443 ± 0.0005 | 0.2973 ± 0.0025 | 0.2423 ± 0.0024 | 0.1761 ± 0.0059 | 0.1737 ± 0.0013 | 0.1597 ± 0.0014 (5) | 0.1585 ± 0.0012 (5) |
| convnext_small | 0.8363 ± 0.0004 | 0.3001 ± 0.0031 | 0.2288 ± 0.0027 | 0.1751 ± 0.0021 | 0.1750 ± 0.0014 | 0.1596 ± 0.0023 (4) | 0.1578 ± 0.0013 (5) |
| convnext_tiny | 0.8252 ± 0.0006 | 0.2884 ± 0.0020 | 0.2209 ± 0.0020 | 0.1822 ± 0.0052 | 0.1816 ± 0.0015 | 0.1611 ± 0.0014 (4) | 0.1598 ± 0.0014 (6) |
| densenet121 | 0.7442 ± 0.0007 | 0.1908 ± 0.0009 | 0.1919 ± 0.0008 | 0.1907 ± 0.0010 | 0.2083 ± 0.0010 | 0.1907 ± 0.0010 (0) | 0.1975 ± 0.0010 (F) |
| densenet161 | 0.7713 ± 0.0005 | 0.1911 ± 0.0014 | 0.1944 ± 0.0013 | 0.1912 ± 0.0014 | 0.2037 ± 0.0015 | 0.1834 ± 0.0013 (2) | 0.1873 ± 0.0013 (8) |
| densenet169 | 0.7560 ± 0.0006 | 0.1914 ± 0.0015 | 0.1936 ± 0.0013 | 0.1915 ± 0.0016 | 0.2065 ± 0.0015 | 0.1880 ± 0.0015 (2) | 0.1934 ± 0.0020 (F) |
| densenet201 | 0.7689 ± 0.0006 | 0.1892 ± 0.0011 | 0.1911 ± 0.0010 | 0.1888 ± 0.0011 | 0.2026 ± 0.0010 | 0.1851 ± 0.0021 (2) | 0.1896 ± 0.0012 (8) |
| efficientnet_b0 | 0.7771 ± 0.0007 | 0.2123 ± 0.0012 | 0.1974 ± 0.0011 | 0.1906 ± 0.0011 | 0.2008 ± 0.0012 | 0.1761 ± 0.0012 (5) | 0.1764 ± 0.0013 (6) |
| efficientnet_b1 | 0.7983 ± 0.0006 | 0.2203 ± 0.0015 | 0.1897 ± 0.0011 | 0.1848 ± 0.0040 | 0.1900 ± 0.0011 | 0.1751 ± 0.0010 (4) | 0.1740 ± 0.0011 (6) |
| efficientnet_b2 | 0.8060 ± 0.0008 | 0.2330 ± 0.0015 | 0.2065 ± 0.0012 | 0.1908 ± 0.0036 | 0.1930 ± 0.0008 | 0.1718 ± 0.0009 (4) | 0.1710 ± 0.0010 (6) |
| efficientnet_b3 | 0.8202 ± 0.0006 | 0.2552 ± 0.0017 | 0.2086 ± 0.0013 | 0.1807 ± 0.0018 | 0.1814 ± 0.0014 | 0.1683 ± 0.0028 (5) | 0.1662 ± 0.0016 (6) |
| efficientnet_b4 | 0.8338 ± 0.0005 | 0.2984 ± 0.0011 | 0.2148 ± 0.0013 | 0.1753 ± 0.0016 | 0.1761 ± 0.0010 | 0.1672 ± 0.0012 (4) | 0.1645 ± 0.0011 (6) |
| efficientnet_b5 | 0.8344 ± 0.0006 | 0.2360 ± 0.0018 | 0.1993 ± 0.0016 | 0.1726 ± 0.0014 | 0.1745 ± 0.0009 | 0.1590 ± 0.0025 (4) | 0.1574 ± 0.0009 (5) |
| efficientnet_b6 | 0.8400 ± 0.0006 | 0.2303 ± 0.0012 | 0.1924 ± 0.0015 | 0.1702 ± 0.0019 | 0.1711 ± 0.0011 | 0.1591 ± 0.0047 (5) | 0.1572 ± 0.0011 (6) |
| efficientnet_b7 | 0.8413 ± 0.0006 | 0.2526 ± 0.0017 | 0.2028 ± 0.0015 | 0.1666 ± 0.0031 | 0.1675 ± 0.0008 | 0.1571 ± 0.0053 (4) | 0.1548 ± 0.0008 (6) |
| efficientnet_v2_l | 0.8580 ± 0.0004 | 0.2484 ± 0.0023 | 0.2088 ± 0.0016 | 0.1762 ± 0.0028 | 0.1748 ± 0.0014 | 0.1623 ± 0.0020 (5) | 0.1610 ± 0.0017 (6) |
| efficientnet_v2_m | 0.8513 ± 0.0005 | 0.2919 ± 0.0025 | 0.2264 ± 0.0020 | 0.1782 ± 0.0020 | 0.1781 ± 0.0015 | 0.1648 ± 0.0037 (4) | 0.1628 ± 0.0013 (5) |
| efficientnet_v2_s | 0.8424 ± 0.0005 | 0.2314 ± 0.0017 | 0.1939 ± 0.0012 | 0.1700 ± 0.0013 | 0.1714 ± 0.0014 | 0.1581 ± 0.0014 (4) | 0.1577 ± 0.0013 (5) |
| googlenet | 0.6978 ± 0.0006 | 0.2158 ± 0.0013 | 0.2071 ± 0.0013 | 0.2055 ± 0.0013 | 0.2279 ± 0.0011 | 0.2034 ± 0.0017 (3) | 0.2042 ± 0.0021 (6) |
| inception_v3 | 0.7730 ± 0.0006 | 0.2297 ± 0.0015 | 0.2176 ± 0.0012 | 0.1991 ± 0.0012 | 0.2040 ± 0.0011 | 0.1812 ± 0.0009 (4) | 0.1799 ± 0.0009 (5) |
| maxvit_t | 0.8370 ± 0.0006 | 0.2245 ± 0.0022 | 0.2041 ± 0.0018 | 0.1759 ± 0.0021 | 0.1752 ± 0.0013 | 0.1629 ± 0.0012 (4) | 0.1621 ± 0.0013 (5) |
| mnasnet0_5 | 0.6775 ± 0.0006 | 0.2237 ± 0.0009 | 0.2109 ± 0.0008 | 0.2087 ± 0.0008 | 0.2320 ± 0.0009 | 0.2006 ± 0.0008 (4) | 0.2012 ± 0.0009 (7) |
| mnasnet0_75 | 0.7120 ± 0.0008 | 0.3056 ± 0.0013 | 0.2132 ± 0.0014 | 0.2088 ± 0.0012 | 0.2260 ± 0.0010 | 0.1958 ± 0.0008 (3) | 0.1970 ± 0.0009 (6) |
| mnasnet1_0 | 0.7347 ± 0.0005 | 0.1825 ± 0.0006 | 0.1843 ± 0.0007 | 0.1825 ± 0.0006 | 0.2004 ± 0.0006 | 0.1828 ± 0.0010 (0) | 0.1913 ± 0.0005 (F) |
| mnasnet1_3 | 0.7649 ± 0.0005 | 0.3273 ± 0.0015 | 0.2104 ± 0.0014 | 0.1987 ± 0.0025 | 0.2052 ± 0.0013 | 0.1826 ± 0.0010 (4) | 0.1825 ± 0.0009 (6) |
| mobilenet_v2 | 0.7216 ± 0.0008 | 0.2805 ± 0.0015 | 0.2054 ± 0.0012 | 0.2024 ± 0.0012 | 0.2209 ± 0.0011 | 0.1945 ± 0.0011 (4) | 0.1952 ± 0.0010 (6) |
| mobilenet_v3_large | 0.7529 ± 0.0006 | 0.2185 ± 0.0012 | 0.1954 ± 0.0011 | 0.1932 ± 0.0011 | 0.2101 ± 0.0012 | 0.1879 ± 0.0024 (4) | 0.1872 ± 0.0011 (6) |
| mobilenet_v3_small | 0.6769 ± 0.0006 | 0.1934 ± 0.0009 | 0.1950 ± 0.0008 | 0.1932 ± 0.0009 | 0.2173 ± 0.0010 | 0.1932 ± 0.0009 (0) | 0.2056 ± 0.0009 (F) |
| regnet_x_16gf | 0.8273 ± 0.0005 | 0.2302 ± 0.0026 | 0.2029 ± 0.0021 | 0.1767 ± 0.0034 | 0.1765 ± 0.0016 | 0.1644 ± 0.0016 (4) | 0.1634 ± 0.0014 (5) |
| regnet_x_1_6gf | 0.7969 ± 0.0007 | 0.3275 ± 0.0018 | 0.2106 ± 0.0015 | 0.1914 ± 0.0014 | 0.1954 ± 0.0012 | 0.1749 ± 0.0014 (4) | 0.1744 ± 0.0014 (6) |
| regnet_x_32gf | 0.8304 ± 0.0006 | 0.2416 ± 0.0027 | 0.2169 ± 0.0027 | 0.1770 ± 0.0028 | 0.1774 ± 0.0018 | 0.1615 ± 0.0014 (4) | 0.1607 ± 0.0014 (5) |
| regnet_x_3_2gf | 0.8119 ± 0.0006 | 0.2692 ± 0.0026 | 0.2181 ± 0.0019 | 0.1925 ± 0.0026 | 0.1932 ± 0.0018 | 0.1722 ± 0.0013 (5) | 0.1718 ± 0.0017 (5) |

| Model | | | | | | | |
|---|---|---|---|---|---|---|---|
| regnet_x_400mf | 0.7489 ± 0.0008 | 0.3058 ± 0.0021 | 0.2076 ± 0.0015 | 0.1994 ± 0.0013 | 0.2114 ± 0.0013 | 0.1834 ± 0.0011 (4) | 0.1838 ± 0.0010 (5) |
| regnet_x_800mf | 0.7753 ± 0.0007 | 0.3306 ± 0.0020 | 0.2136 ± 0.0018 | 0.1992 ± 0.0012 | 0.2070 ± 0.0012 | 0.1791 ± 0.0013 (4) | 0.1791 ± 0.0010 (5) |
| regnet_x_8gf | 0.8170 ± 0.0006 | 0.2353 ± 0.0022 | 0.2038 ± 0.0017 | 0.1793 ± 0.0013 | 0.1797 ± 0.0013 | 0.1647 ± 0.0013 (5) | 0.1642 ± 0.0013 (5) |
| regnet_y_128gf | 0.8824 ± 0.0003 | 0.1555 ± 0.0011 | 0.1561 ± 0.0012 | 0.1507 ± 0.0010 | 0.1535 ± 0.0010 | 0.1486 ± 0.0017 (0) | 0.1465 ± 0.0008 (8) |
| regnet_y_16gf | 0.8292 ± 0.0005 | 0.2842 ± 0.0027 | 0.2316 ± 0.0026 | 0.1738 ± 0.0016 | 0.1734 ± 0.0016 | 0.1579 ± 0.0014 (4) | 0.1574 ± 0.0012 (5) |
| regnet_y_1_6gf | 0.8090 ± 0.0007 | 0.2637 ± 0.0015 | 0.2111 ± 0.0009 | 0.1915 ± 0.0081 | 0.1917 ± 0.0009 | 0.1701 ± 0.0009 (4) | 0.1697 ± 0.0009 (5) |
| regnet_y_32gf | 0.8339 ± 0.0005 | 0.2483 ± 0.0031 | 0.2069 ± 0.0026 | 0.1717 ± 0.0016 | 0.1725 ± 0.0014 | 0.1572 ± 0.0011 (4) | 0.1566 ± 0.0012 (5) |
| regnet_y_3_2gf | 0.8198 ± 0.0006 | 0.2335 ± 0.0017 | 0.1985 ± 0.0015 | 0.1834 ± 0.0019 | 0.1853 ± 0.0012 | 0.1684 ± 0.0010 (6) | 0.1688 ± 0.0012 (5) |
| regnet_y_400mf | 0.7581 ± 0.0006 | 0.2575 ± 0.0013 | 0.2141 ± 0.0012 | 0.2055 ± 0.0013 | 0.2173 ± 0.0013 | 0.1850 ± 0.0011 (4) | 0.1855 ± 0.0011 (5) |
| regnet_y_800mf | 0.7885 ± 0.0007 | 0.2478 ± 0.0016 | 0.2035 ± 0.0012 | 0.1913 ± 0.0012 | 0.2000 ± 0.0010 | 0.1734 ± 0.0013 (4) | 0.1731 ± 0.0010 (5) |
| regnet_y_8gf | 0.8283 ± 0.0006 | 0.2337 ± 0.0027 | 0.1972 ± 0.0025 | 0.1756 ± 0.0030 | 0.1741 ± 0.0019 | 0.1604 ± 0.0017 (4) | 0.1597 ± 0.0017 (5) |
| resnet101 | 0.8188 ± 0.0005 | 0.2632 ± 0.0026 | 0.2179 ± 0.0023 | 0.1839 ± 0.0020 | 0.1837 ± 0.0016 | 0.1694 ± 0.0032 (4) | 0.1670 ± 0.0017 (5) |
| resnet152 | 0.8230 ± 0.0007 | 0.2561 ± 0.0019 | 0.2098 ± 0.0021 | 0.1728 ± 0.0020 | 0.1732 ± 0.0012 | 0.1615 ± 0.0037 (4) | 0.1591 ± 0.0012 (5) |
| resnet18 | 0.6976 ± 0.0006 | 0.2001 ± 0.0005 | 0.2016 ± 0.0005 | 0.1996 ± 0.0006 | 0.2204 ± 0.0006 | 0.2000 ± 0.0007 (1) | 0.2094 ± 0.0009 (F) |
| resnet34 | 0.7331 ± 0.0007 | 0.1911 ± 0.0010 | 0.1924 ± 0.0009 | 0.1912 ± 0.0009 | 0.2105 ± 0.0009 | 0.1910 ± 0.0010 (2) | 0.1960 ± 0.0010 (F) |
| resnet50 | 0.8084 ± 0.0006 | 0.3216 ± 0.0024 | 0.2105 ± 0.0017 | 0.1839 ± 0.0022 | 0.1852 ± 0.0011 | 0.1699 ± 0.0031 (4) | 0.1676 ± 0.0011 (5) |
| resnext101_32x8d | 0.8283 ± 0.0006 | 0.4204 ± 0.0038 | 0.2538 ± 0.0036 | 0.1849 ± 0.0048 | 0.1834 ± 0.0012 | 0.1641 ± 0.0008 (4) | 0.1632 ± 0.0007 (5) |
| resnext101_64x4d | 0.8325 ± 0.0005 | 0.3962 ± 0.0031 | 0.2371 ± 0.0029 | 0.1777 ± 0.0024 | 0.1771 ± 0.0018 | 0.1630 ± 0.0016 (4) | 0.1606 ± 0.0015 (5) |
| resnext50_32x4d | 0.8119 ± 0.0007 | 0.2698 ± 0.0022 | 0.2214 ± 0.0023 | 0.1882 ± 0.0029 | 0.1877 ± 0.0016 | 0.1712 ± 0.0014 (4) | 0.1696 ± 0.0015 (5) |
| shufflenet_v2_x0_5 | 0.6058 ± 0.0005 | 0.2192 ± 0.0009 | 0.2221 ± 0.0009 | 0.2180 ± 0.0009 | 0.2406 ± 0.0009 | 0.2152 ± 0.0014 (4) | 0.2159 ± 0.0009 (7) |
| shufflenet_v2_x1_0 | 0.6936 ± 0.0008 | 0.1976 ± 0.0009 | 0.2014 ± 0.0010 | 0.1972 ± 0.0009 | 0.2117 ± 0.0009 | 0.1932 ± 0.0010 (4) | 0.1931 ± 0.0010 (7) |
| shufflenet_v2_x1_5 | 0.7303 ± 0.0007 | 0.2856 ± 0.0014 | 0.2122 ± 0.0014 | 0.2072 ± 0.0013 | 0.2231 ± 0.0011 | 0.1964 ± 0.0010 (4) | 0.1969 ± 0.0011 (6) |
| shufflenet_v2_x2_0 | 0.7621 ± 0.0007 | 0.2824 ± 0.0010 | 0.2044 ± 0.0014 | 0.1950 ± 0.0030 | 0.2028 ± 0.0012 | 0.1786 ± 0.0010 (5) | 0.1781 ± 0.0011 (6) |
| squeezenet1_0 | 0.5810 ± 0.0005 | 0.2340 ± 0.0005 | 0.2362 ± 0.0006 | 0.2318 ± 0.0006 | 0.2621 ± 0.0007 | 0.2318 ± 0.0006 (0) | 0.2751 ± 0.0010 (F) |
| squeezenet1_1 | 0.5820 ± 0.0005 | 0.2221 ± 0.0005 | 0.2238 ± 0.0006 | 0.2209 ± 0.0005 | 0.2530 ± 0.0008 | 0.2209 ± 0.0005 (0) | 0.2620 ± 0.0011 (F) |
| swin_b | 0.8358 ± 0.0006 | 0.2804 ± 0.0032 | 0.2444 ± 0.0039 | 0.1801 ± 0.0036 | 0.1780 ± 0.0014 | 0.1645 ± 0.0015 (4) | 0.1617 ± 0.0013 (5) |
| swin_s | 0.8321 ± 0.0005 | 0.2343 ± 0.0015 | 0.2151 ± 0.0016 | 0.1813 ± 0.0024 | 0.1817 ± 0.0013 | 0.1675 ± 0.0012 (4) | 0.1656 ± 0.0012 (5) |
| swin_t | 0.8147 ± 0.0005 | 0.2174 ± 0.0022 | 0.1962 ± 0.0021 | 0.1820 ± 0.0023 | 0.1859 ± 0.0016 | 0.1690 ± 0.0013 (4) | 0.1677 ± 0.0013 (5) |
| swin_v2_b | 0.8415 ± 0.0005 | 0.2515 ± 0.0030 | 0.2232 ± 0.0027 | 0.1786 ± 0.0030 | 0.1784 ± 0.0011 | 0.1644 ± 0.0012 (4) | 0.1633 ± 0.0011 (5) |
| swin_v2_s | 0.8372 ± 0.0004 | 0.2333 ± 0.0016 | 0.2060 ± 0.0015 | 0.1704 ± 0.0027 | 0.1711 ± 0.0010 | 0.1593 ± 0.0010 (4) | 0.1578 ± 0.0012 (5) |
| swin_v2_t | 0.8208 ± 0.0005 | 0.2183 ± 0.0015 | 0.1930 ± 0.0014 | 0.1768 ± 0.0019 | 0.1793 ± 0.0010 | 0.1649 ± 0.0011 (4) | 0.1636 ± 0.0011 (5) |
| vgg11 | 0.6905 ± 0.0006 | 0.1922 ± 0.0011 | 0.1929 ± 0.0011 | 0.1918 ± 0.0011 | 0.2154 ± 0.0012 | 0.1918 ± 0.0011 (0) | 0.2142 ± 0.0014 (F) |
| vgg11_bn | 0.7037 ± 0.0007 | 0.1896 ± 0.0006 | 0.1907 ± 0.0005 | 0.1893 ± 0.0006 | 0.2113 ± 0.0008 | 0.1893 ± 0.0006 (0) | 0.2131 ± 0.0007 (F) |
| vgg13 | 0.6995 ± 0.0005 | 0.1899 ± 0.0009 | 0.1907 ± 0.0008 | 0.1895 ± 0.0009 | 0.2114 ± 0.0010 | 0.1895 ± 0.0009 (0) | 0.2099 ± 0.0013 (F) |
| vgg13_bn | 0.7160 ± 0.0006 | 0.1892 ± 0.0008 | 0.1904 ± 0.0008 | 0.1891 ± 0.0009 | 0.2105 ± 0.0009 | 0.1891 ± 0.0008 (0) | 0.2088 ± 0.0010 (F) |
| vgg16 | 0.7161 ± 0.0007 | 0.1839 ± 0.0006 | 0.1851 ± 0.0006 | 0.1839 ± 0.0007 | 0.2051 ± 0.0005 | 0.1839 ± 0.0007 (0) | 0.2020 ± 0.0012 (F) |
| vgg16_bn | 0.7339 ± 0.0006 | 0.1823 ± 0.0007 | 0.1838 ± 0.0006 | 0.1823 ± 0.0008 | 0.2003 ± 0.0008 | 0.1823 ± 0.0007 (0) | 0.1967 ± 0.0008 (F) |
| vgg19 | 0.7238 ± 0.0005 | 0.1831 ± 0.0008 | 0.1842 ± 0.0007 | 0.1831 ± 0.0008 | 0.2046 ± 0.0008 | 0.1836 ± 0.0009 (0) | 0.1990 ± 0.0009 (F) |
| vgg19_bn | 0.7424 ± 0.0006 | 0.1843 ± 0.0013 | 0.1856 ± 0.0011 | 0.1844 ± 0.0013 | 0.2019 ± 0.0013 | 0.1844 ± 0.0013 (0) | 0.2007 ± 0.0013 (F) |
| vit_b_16 | 0.8108 ± 0.0006 | 0.2343 ± 0.0012 | 0.2102 ± 0.0010 | 0.1810 ± 0.0016 | 0.1833 ± 0.0009 | 0.1676 ± 0.0009 (4) | 0.1662 ± 0.0009 (5) |

| | | | | | | |
|---|---|---|---|---|---|---|
| vit_b_32 | 0.7596 ± 0.0004 | 0.2279 ± 0.0012 | 0.2093 ± 0.0011 | 0.1913 ± 0.0012 | 0.1950 ± 0.0012 | 0.1726 ± 0.0011 (4) | 0.1715 ± 0.0010 (5) |
| vit_h_14 | 0.8855 ± 0.0005 | 0.1717 ± 0.0016 | 0.1674 ± 0.0016 | 0.1551 ± 0.0015 | 0.1573 ± 0.0012 | 0.1504 ± 0.0022 (4) | 0.1494 ± 0.0009 (6) |
| vit_l_16 | 0.7966 ± 0.0007 | 0.2250 ± 0.0019 | 0.2149 ± 0.0016 | 0.1853 ± 0.0017 | 0.1871 ± 0.0011 | 0.1657 ± 0.0007 (4) | 0.1655 ± 0.0009 (4) |
| vit_l_32 | 0.7699 ± 0.0007 | 0.2451 ± 0.0017 | 0.2276 ± 0.0015 | 0.1906 ± 0.0013 | 0.1931 ± 0.0005 | 0.1673 ± 0.0004 (4) | 0.1674 ± 0.0004 (4) |
| wide_resnet101_2 | 0.8252 ± 0.0006 | 0.2795 ± 0.0027 | 0.2280 ± 0.0027 | 0.1789 ± 0.0014 | 0.1785 ± 0.0013 | 0.1624 ± 0.0013 (5) | 0.1612 ± 0.0012 (5) |
| wide_resnet50_2 | 0.8162 ± 0.0007 | 0.3592 ± 0.0032 | 0.2289 ± 0.0030 | 0.1864 ± 0.0027 | 0.1865 ± 0.0015 | 0.1684 ± 0.0016 (4) | 0.1668 ± 0.0013 (5) |
| efficientnetv2_xl | 0.8556 ± 0.0005 | 0.4402 ± 0.0032 | 0.3506 ± 0.0039 | 0.1957 ± 0.0027 | 0.1937 ± 0.0023 | 0.1734 ± 0.0030 (5) | 0.1693 ± 0.0018 (6) |
| vit_l_16_384 | 0.8709 ± 0.0005 | 0.1472 ± 0.0010 | 0.1474 ± 0.0009 | 0.1465 ± 0.0010 | 0.1541 ± 0.0010 | 0.1465 ± 0.0010 (0) | 0.1508 ± 0.0010 (F) |
| vit_b_16_sam | 0.8022 ± 0.0005 | 0.1573 ± 0.0011 | 0.1570 ± 0.0011 | 0.1564 ± 0.0011 | 0.1629 ± 0.0012 | 0.1564 ± 0.0011 (0) | 0.1580 ± 0.0016 (F) |
| vit_b_32_sam | 0.7371 ± 0.0004 | 0.1694 ± 0.0008 | 0.1689 ± 0.0008 | 0.1683 ± 0.0008 | 0.1798 ± 0.0008 | 0.1683 ± 0.0008 (0) | 0.1702 ± 0.0010 (F) |

Table 14: AURC (mean ±std) for all models evaluated on ImageNet

| Model | Accuracy[%] | | | Method | | | |
|---|---|---|---|---|---|---|---|
| | | MSP | MSP-TS-NLL | MSP-TS-AURC | LogitsMargin | MSP-pNorm ($p^*$) | MaxLogit-pNorm ($p^*$) |
| alexnet | 0.5654 ± 0.0007 | 0.1841 ± 0.0005 | 0.1846 ± 0.0005 | 0.1840 ± 0.0005 | 0.1958 ± 0.0005 | 0.1840 ± 0.0005 (0) | 0.1896 ± 0.0005 (F) |
| convnext_base | 0.8406 ± 0.0004 | 0.0573 ± 0.0004 | 0.0473 ± 0.0003 | 0.0397 ± 0.0004 | 0.0396 ± 0.0003 | 0.0372 ± 0.0003 (4) | 0.0371 ± 0.0002 (5) |
| convnext_large | 0.8443 ± 0.0005 | 0.0553 ± 0.0004 | 0.0474 ± 0.0003 | 0.0380 ± 0.0008 | 0.0376 ± 0.0003 | 0.0356 ± 0.0002 (5) | 0.0355 ± 0.0002 (5) |
| convnext_small | 0.8363 ± 0.0004 | 0.0591 ± 0.0005 | 0.0484 ± 0.0004 | 0.0404 ± 0.0003 | 0.0404 ± 0.0002 | 0.0381 ± 0.0003 (4) | 0.0378 ± 0.0002 (5) |
| convnext_tiny | 0.8252 ± 0.0006 | 0.0620 ± 0.0004 | 0.0513 ± 0.0003 | 0.0451 ± 0.0008 | 0.0450 ± 0.0003 | 0.0418 ± 0.0003 (4) | 0.0416 ± 0.0003 (6) |
| densenet121 | 0.7442 ± 0.0007 | 0.0779 ± 0.0004 | 0.0781 ± 0.0004 | 0.0779 ± 0.0004 | 0.0817 ± 0.0004 | 0.0779 ± 0.0004 (0) | 0.0794 ± 0.0004 (F) |
| densenet161 | 0.7713 ± 0.0005 | 0.0667 ± 0.0004 | 0.0673 ± 0.0004 | 0.0667 ± 0.0004 | 0.0692 ± 0.0004 | 0.0651 ± 0.0004 (2) | 0.0659 ± 0.0004 (8) |
| densenet169 | 0.7560 ± 0.0006 | 0.0730 ± 0.0005 | 0.0735 ± 0.0004 | 0.0730 ± 0.0005 | 0.0762 ± 0.0005 | 0.0723 ± 0.0005 (2) | 0.0734 ± 0.0006 (F) |
| densenet201 | 0.7689 ± 0.0006 | 0.0673 ± 0.0004 | 0.0676 ± 0.0004 | 0.0672 ± 0.0004 | 0.0700 ± 0.0004 | 0.0664 ± 0.0006 (2) | 0.0673 ± 0.0004 (8) |
| efficientnet_b0 | 0.7771 ± 0.0007 | 0.0685 ± 0.0004 | 0.0656 ± 0.0004 | 0.0643 ± 0.0004 | 0.0663 ± 0.0004 | 0.0614 ± 0.0004 (5) | 0.0615 ± 0.0004 (6) |
| efficientnet_b1 | 0.7983 ± 0.0006 | 0.0615 ± 0.0005 | 0.0560 ± 0.0004 | 0.0551 ± 0.0007 | 0.0560 ± 0.0004 | 0.0533 ± 0.0004 (4) | 0.0532 ± 0.0004 (6) |
| efficientnet_b2 | 0.8060 ± 0.0008 | 0.0607 ± 0.0005 | 0.0561 ± 0.0004 | 0.0533 ± 0.0007 | 0.0537 ± 0.0004 | 0.0500 ± 0.0004 (4) | 0.0499 ± 0.0004 (6) |
| efficientnet_b3 | 0.8202 ± 0.0006 | 0.0587 ± 0.0003 | 0.0512 ± 0.0003 | 0.0466 ± 0.0004 | 0.0467 ± 0.0003 | 0.0446 ± 0.0004 (5) | 0.0443 ± 0.0003 (6) |
| efficientnet_b4 | 0.8338 ± 0.0005 | 0.0599 ± 0.0003 | 0.0472 ± 0.0003 | 0.0412 ± 0.0004 | 0.0413 ± 0.0003 | 0.0400 ± 0.0003 (4) | 0.0396 ± 0.0003 (6) |
| efficientnet_b5 | 0.8344 ± 0.0006 | 0.0502 ± 0.0004 | 0.0446 ± 0.0003 | 0.0406 ± 0.0004 | 0.0409 ± 0.0003 | 0.0386 ± 0.0003 (4) | 0.0383 ± 0.0003 (5) |
| efficientnet_b6 | 0.8400 ± 0.0006 | 0.0473 ± 0.0003 | 0.0417 ± 0.0003 | 0.0385 ± 0.0003 | 0.0386 ± 0.0003 | 0.0369 ± 0.0008 (5) | 0.0366 ± 0.0003 (6) |
| efficientnet_b7 | 0.8413 ± 0.0006 | 0.0500 ± 0.0003 | 0.0428 ± 0.0003 | 0.0375 ± 0.0004 | 0.0377 ± 0.0002 | 0.0362 ± 0.0007 (4) | 0.0358 ± 0.0002 (6) |
| efficientnet_v2_l | 0.8580 ± 0.0004 | 0.0432 ± 0.0003 | 0.0380 ± 0.0002 | 0.0337 ± 0.0003 | 0.0336 ± 0.0002 | 0.0319 ± 0.0002 (5) | 0.0317 ± 0.0003 (6) |
| efficientnet_v2_m | 0.8513 ± 0.0005 | 0.0517 ± 0.0003 | 0.0427 ± 0.0003 | 0.0361 ± 0.0003 | 0.0361 ± 0.0002 | 0.0343 ± 0.0005 (4) | 0.0340 ± 0.0002 (5) |
| efficientnet_v2_s | 0.8424 ± 0.0005 | 0.0466 ± 0.0003 | 0.0412 ± 0.0002 | 0.0377 ± 0.0002 | 0.0379 ± 0.0002 | 0.0360 ± 0.0002 (4) | 0.0359 ± 0.0002 (5) |
| googlenet | 0.6978 ± 0.0006 | 0.1053 ± 0.0005 | 0.1031 ± 0.0005 | 0.1027 ± 0.0005 | 0.1083 ± 0.0005 | 0.1022 ± 0.0006 (3) | 0.1024 ± 0.0007 (6) |
| inception_v3 | 0.7730 ± 0.0006 | 0.0737 ± 0.0003 | 0.0713 ± 0.0003 | 0.0676 ± 0.0003 | 0.0686 ± 0.0003 | 0.0640 ± 0.0003 (4) | 0.0638 ± 0.0003 (5) |
| maxvit_t | 0.8370 ± 0.0006 | 0.0475 ± 0.0004 | 0.0444 ± 0.0003 | 0.0402 ± 0.0004 | 0.0401 ± 0.0003 | 0.0383 ± 0.0002 (4) | 0.0382 ± 0.0002 (5) |
| mnasnet0_5 | 0.6775 ± 0.0006 | 0.1177 ± 0.0004 | 0.1144 ± 0.0003 | 0.1138 ± 0.0004 | 0.1199 ± 0.0004 | 0.1116 ± 0.0004 (4) | 0.1118 ± 0.0004 (7) |
| mnasnet0_75 | 0.7120 ± 0.0008 | 0.1201 ± 0.0006 | 0.0977 ± 0.0006 | 0.0967 ± 0.0005 | 0.1008 ± 0.0005 | 0.0935 ± 0.0004 (3) | 0.0938 ± 0.0005 (6) |
| mnasnet1_0 | 0.7347 ± 0.0005 | 0.0801 ± 0.0003 | 0.0805 ± 0.0003 | 0.0801 ± 0.0003 | 0.0842 ± 0.0003 | 0.0802 ± 0.0003 (0) | 0.0821 ± 0.0003 (F) |
| mnasnet1_3 | 0.7649 ± 0.0005 | 0.0972 ± 0.0005 | 0.0732 ± 0.0004 | 0.0708 ± 0.0005 | 0.0721 ± 0.0004 | 0.0675 ± 0.0003 (4) | 0.0675 ± 0.0003 (6) |
| mobilenet_v2 | 0.7216 ± 0.0008 | 0.1090 ± 0.0006 | 0.0913 ± 0.0004 | 0.0906 ± 0.0005 | 0.0949 ± 0.0004 | 0.0887 ± 0.0004 (4) | 0.0889 ± 0.0004 (6) |
| mobilenet_v3_large | 0.7529 ± 0.0006 | 0.0801 ± 0.0004 | 0.0752 ± 0.0004 | 0.0747 ± 0.0004 | 0.0783 ± 0.0004 | 0.0736 ± 0.0005 (4) | 0.0734 ± 0.0004 (6) |
| mobilenet_v3_small | 0.6769 ± 0.0006 | 0.1100 ± 0.0004 | 0.1104 ± 0.0004 | 0.1100 ± 0.0005 | 0.1163 ± 0.0005 | 0.1100 ± 0.0005 (0) | 0.1133 ± 0.0004 (F) |
| regnet_x_16gf | 0.8273 ± 0.0005 | 0.0519 ± 0.0004 | 0.0477 ± 0.0003 | 0.0436 ± 0.0005 | 0.0435 ± 0.0003 | 0.0416 ± 0.0003 (4) | 0.0415 ± 0.0002 (5) |
| regnet_x_1_6gf | 0.7969 ± 0.0007 | 0.0814 ± 0.0005 | 0.0603 ± 0.0003 | 0.0568 ± 0.0004 | 0.0575 ± 0.0004 | 0.0538 ± 0.0004 (4) | 0.0537 ± 0.0004 (6) |
| regnet_x_32gf | 0.8304 ± 0.0006 | 0.0526 ± 0.0005 | 0.0488 ± 0.0004 | 0.0426 ± 0.0005 | 0.0427 ± 0.0004 | 0.0402 ± 0.0003 (4) | 0.0401 ± 0.0003 (5) |
| regnet_x_3_2gf | 0.8119 ± 0.0006 | 0.0645 ± 0.0006 | 0.0558 ± 0.0004 | 0.0515 ± 0.0005 | 0.0516 ± 0.0004 | 0.0480 ± 0.0003 (5) | 0.0480 ± 0.0004 (5) |

| Model | | | | | | | |
|---|---|---|---|---|---|---|---|
| regnet_x_400mf | 0.7489 ± 0.0008 | 0.1008 ± 0.0007 | 0.0795 ± 0.0005 | 0.0778 ± 0.0005 | 0.0804 ± 0.0005 | 0.0743 ± 0.0004 (4) | 0.0744 ± 0.0004 (5) |
| regnet_x_800mf | 0.7753 ± 0.0007 | 0.0926 ± 0.0006 | 0.0695 ± 0.0005 | 0.0667 ± 0.0005 | 0.0682 ± 0.0004 | 0.0627 ± 0.0004 (4) | 0.0627 ± 0.0004 (5) |
| regnet_x_8gf | 0.8170 ± 0.0006 | 0.0567 ± 0.0005 | 0.0515 ± 0.0003 | 0.0475 ± 0.0004 | 0.0475 ± 0.0003 | 0.0451 ± 0.0003 (5) | 0.0450 ± 0.0003 (5) |
| regnet_y_128gf | 0.8824 ± 0.0003 | 0.0244 ± 0.0001 | 0.0245 ± 0.0001 | 0.0239 ± 0.0003 | 0.0242 ± 0.0001 | 0.0236 ± 0.0002 (0) | 0.0234 ± 0.0001 (8) |
| regnet_y_16gf | 0.8292 ± 0.0005 | 0.0596 ± 0.0003 | 0.0514 ± 0.0003 | 0.0425 ± 0.0003 | 0.0424 ± 0.0003 | 0.0400 ± 0.0003 (4) | 0.0399 ± 0.0002 (5) |
| regnet_y_1_6gf | 0.8090 ± 0.0007 | 0.0648 ± 0.0004 | 0.0557 ± 0.0003 | 0.0524 ± 0.0013 | 0.0524 ± 0.0004 | 0.0487 ± 0.0003 (4) | 0.0486 ± 0.0003 (5) |
| regnet_y_32gf | 0.8339 ± 0.0005 | 0.0522 ± 0.0005 | 0.0460 ± 0.0004 | 0.0406 ± 0.0003 | 0.0408 ± 0.0003 | 0.0384 ± 0.0002 (4) | 0.0384 ± 0.0002 (5) |
| regnet_y_3_2gf | 0.8198 ± 0.0006 | 0.0553 ± 0.0004 | 0.0496 ± 0.0004 | 0.0472 ± 0.0004 | 0.0475 ± 0.0004 | 0.0447 ± 0.0003 (6) | 0.0448 ± 0.0003 (5) |
| regnet_y_400mf | 0.7581 ± 0.0006 | 0.0860 ± 0.0005 | 0.0769 ± 0.0004 | 0.0751 ± 0.0004 | 0.0776 ± 0.0004 | 0.0708 ± 0.0004 (4) | 0.0709 ± 0.0004 (5) |
| regnet_y_800mf | 0.7885 ± 0.0007 | 0.0706 ± 0.0005 | 0.0623 ± 0.0004 | 0.0600 ± 0.0004 | 0.0616 ± 0.0004 | 0.0566 ± 0.0004 (4) | 0.0566 ± 0.0004 (5) |
| regnet_y_8gf | 0.8283 ± 0.0006 | 0.0521 ± 0.0004 | 0.0464 ± 0.0004 | 0.0431 ± 0.0005 | 0.0428 ± 0.0003 | 0.0407 ± 0.0003 (4) | 0.0406 ± 0.0003 (5) |
| resnet101 | 0.8188 ± 0.0005 | 0.0606 ± 0.0005 | 0.0532 ± 0.0005 | 0.0476 ± 0.0004 | 0.0476 ± 0.0003 | 0.0452 ± 0.0005 (4) | 0.0448 ± 0.0004 (5) |
| resnet152 | 0.8230 ± 0.0007 | 0.0577 ± 0.0004 | 0.0503 ± 0.0004 | 0.0444 ± 0.0005 | 0.0445 ± 0.0003 | 0.0426 ± 0.0006 (4) | 0.0422 ± 0.0003 (5) |
| resnet18 | 0.6976 ± 0.0006 | 0.1015 ± 0.0004 | 0.1018 ± 0.0004 | 0.1013 ± 0.0004 | 0.1066 ± 0.0004 | 0.1014 ± 0.0004 (1) | 0.1038 ± 0.0005 (F) |
| resnet34 | 0.7331 ± 0.0007 | 0.0828 ± 0.0005 | 0.0831 ± 0.0004 | 0.0828 ± 0.0005 | 0.0872 ± 0.0004 | 0.0828 ± 0.0004 (2) | 0.0839 ± 0.0005 (F) |
| resnet50 | 0.8084 ± 0.0006 | 0.0750 ± 0.0005 | 0.0559 ± 0.0004 | 0.0513 ± 0.0004 | 0.0515 ± 0.0003 | 0.0489 ± 0.0004 (4) | 0.0485 ± 0.0003 (5) |
| resnext101_32x8d | 0.8283 ± 0.0006 | 0.0813 ± 0.0007 | 0.0553 ± 0.0006 | 0.0445 ± 0.0007 | 0.0443 ± 0.0003 | 0.0413 ± 0.0003 (4) | 0.0411 ± 0.0003 (5) |
| resnext101_64x4d | 0.8325 ± 0.0005 | 0.0754 ± 0.0005 | 0.0511 ± 0.0004 | 0.0420 ± 0.0004 | 0.0419 ± 0.0003 | 0.0398 ± 0.0003 (4) | 0.0394 ± 0.0003 (5) |
| resnext50_32x4d | 0.8119 ± 0.0007 | 0.0646 ± 0.0005 | 0.0564 ± 0.0005 | 0.0508 ± 0.0006 | 0.0507 ± 0.0004 | 0.0479 ± 0.0003 (4) | 0.0476 ± 0.0004 (5) |
| shufflenet_v2_x0_5 | 0.6058 ± 0.0005 | 0.1571 ± 0.0004 | 0.1580 ± 0.0004 | 0.1567 ± 0.0004 | 0.1636 ± 0.0005 | 0.1559 ± 0.0004 (4) | 0.1561 ± 0.0004 (7) |
| shufflenet_v2_x1_0 | 0.6936 ± 0.0008 | 0.1028 ± 0.0004 | 0.1037 ± 0.0004 | 0.1027 ± 0.0004 | 0.1064 ± 0.0004 | 0.1017 ± 0.0004 (4) | 0.1016 ± 0.0004 (7) |
| shufflenet_v2_x1_5 | 0.7303 ± 0.0007 | 0.1057 ± 0.0005 | 0.0889 ± 0.0004 | 0.0877 ± 0.0005 | 0.0914 ± 0.0004 | 0.0853 ± 0.0004 (4) | 0.0854 ± 0.0005 (6) |
| shufflenet_v2_x2_0 | 0.7621 ± 0.0007 | 0.0893 ± 0.0004 | 0.0732 ± 0.0004 | 0.0712 ± 0.0008 | 0.0729 ± 0.0005 | 0.0679 ± 0.0004 (5) | 0.0677 ± 0.0004 (6) |
| squeezenet1_0 | 0.5810 ± 0.0005 | 0.1773 ± 0.0004 | 0.1781 ± 0.0004 | 0.1767 ± 0.0004 | 0.1862 ± 0.0004 | 0.1767 ± 0.0004 (0) | 0.1903 ± 0.0005 (F) |
| squeezenet1_1 | 0.5820 ± 0.0005 | 0.1729 ± 0.0004 | 0.1735 ± 0.0004 | 0.1725 ± 0.0004 | 0.1826 ± 0.0004 | 0.1725 ± 0.0004 (0) | 0.1855 ± 0.0005 (F) |
| swin_b | 0.8358 ± 0.0006 | 0.0563 ± 0.0005 | 0.0509 ± 0.0005 | 0.0413 ± 0.0005 | 0.0410 ± 0.0003 | 0.0389 ± 0.0003 (4) | 0.0385 ± 0.0003 (5) |
| swin_s | 0.8321 ± 0.0005 | 0.0508 ± 0.0003 | 0.0479 ± 0.0003 | 0.0427 ± 0.0004 | 0.0427 ± 0.0003 | 0.0406 ± 0.0002 (4) | 0.0403 ± 0.0003 (5) |
| swin_t | 0.8147 ± 0.0005 | 0.0546 ± 0.0005 | 0.0511 ± 0.0004 | 0.0487 ± 0.0004 | 0.0494 ± 0.0003 | 0.0466 ± 0.0003 (4) | 0.0463 ± 0.0003 (5) |
| swin_v2_b | 0.8415 ± 0.0005 | 0.0498 ± 0.0005 | 0.0457 ± 0.0004 | 0.0392 ± 0.0004 | 0.0392 ± 0.0002 | 0.0372 ± 0.0002 (4) | 0.0370 ± 0.0002 (5) |
| swin_v2_s | 0.8372 ± 0.0004 | 0.0488 ± 0.0003 | 0.0447 ± 0.0004 | 0.0394 ± 0.0004 | 0.0395 ± 0.0002 | 0.0377 ± 0.0002 (4) | 0.0375 ± 0.0002 (5) |
| swin_v2_t | 0.8208 ± 0.0005 | 0.0525 ± 0.0003 | 0.0484 ± 0.0003 | 0.0458 ± 0.0004 | 0.0462 ± 0.0003 | 0.0438 ± 0.0003 (4) | 0.0436 ± 0.0003 (5) |
| vgg11 | 0.6905 ± 0.0006 | 0.1029 ± 0.0005 | 0.1031 ± 0.0004 | 0.1028 ± 0.0005 | 0.1089 ± 0.0004 | 0.1028 ± 0.0005 (0) | 0.1086 ± 0.0005 (F) |
| vgg11_bn | 0.7037 ± 0.0007 | 0.0959 ± 0.0004 | 0.0962 ± 0.0004 | 0.0958 ± 0.0004 | 0.1013 ± 0.0005 | 0.0958 ± 0.0004 (0) | 0.1017 ± 0.0005 (F) |
| vgg13 | 0.6995 ± 0.0005 | 0.0980 ± 0.0004 | 0.0982 ± 0.0004 | 0.0979 ± 0.0004 | 0.1033 ± 0.0004 | 0.0979 ± 0.0004 (0) | 0.1030 ± 0.0005 (F) |
| vgg13_bn | 0.7160 ± 0.0006 | 0.0901 ± 0.0003 | 0.0904 ± 0.0003 | 0.0900 ± 0.0003 | 0.0952 ± 0.0004 | 0.0900 ± 0.0003 (0) | 0.0948 ± 0.0004 (F) |
| vgg16 | 0.7161 ± 0.0007 | 0.0888 ± 0.0004 | 0.0890 ± 0.0004 | 0.0887 ± 0.0004 | 0.0938 ± 0.0004 | 0.0887 ± 0.0004 (0) | 0.0931 ± 0.0005 (F) |
| vgg16_bn | 0.7339 ± 0.0006 | 0.0804 ± 0.0004 | 0.0808 ± 0.0003 | 0.0804 ± 0.0004 | 0.0845 ± 0.0004 | 0.0804 ± 0.0004 (0) | 0.0837 ± 0.0004 (F) |
| vgg19 | 0.7238 ± 0.0005 | 0.0851 ± 0.0004 | 0.0853 ± 0.0003 | 0.0851 ± 0.0004 | 0.0901 ± 0.0004 | 0.0852 ± 0.0006 (0) | 0.0888 ± 0.0004 (F) |
| vgg19_bn | 0.7424 ± 0.0006 | 0.0772 ± 0.0004 | 0.0775 ± 0.0003 | 0.0772 ± 0.0004 | 0.0811 ± 0.0004 | 0.0772 ± 0.0004 (0) | 0.0809 ± 0.0004 (F) |
| vit_b_16 | 0.8108 ± 0.0006 | 0.0590 ± 0.0004 | 0.0549 ± 0.0003 | 0.0499 ± 0.0004 | 0.0503 ± 0.0003 | 0.0477 ± 0.0003 (4) | 0.0474 ± 0.0003 (5) |

| | | | | | | |
|---|---|---|---|---|---|---|
| vit_b_32 | 0.7596 ± 0.0004 | 0.0791 ± 0.0003 | 0.0753 ± 0.0003 | 0.0715 ± 0.0003 | 0.0723 ± 0.0004 | 0.0676 ± 0.0003 (4) | 0.0674 ± 0.0003 (5) |
| vit_h_14 | 0.8855 ± 0.0005 | 0.0253 ± 0.0002 | 0.0248 ± 0.0001 | 0.0235 ± 0.0002 | 0.0237 ± 0.0001 | 0.0230 ± 0.0002 (4) | 0.0229 ± 0.0001 (6) |
| vit_l_16 | 0.7966 ± 0.0007 | 0.0630 ± 0.0004 | 0.0612 ± 0.0003 | 0.0558 ± 0.0004 | 0.0561 ± 0.0003 | 0.0523 ± 0.0003 (4) | 0.0522 ± 0.0003 (4) |
| vit_l_32 | 0.7699 ± 0.0007 | 0.0781 ± 0.0005 | 0.0746 ± 0.0005 | 0.0672 ± 0.0005 | 0.0677 ± 0.0004 | 0.0625 ± 0.0003 (4) | 0.0625 ± 0.0003 (4) |
| wide_resnet101_2 | 0.8252 ± 0.0006 | 0.0606 ± 0.0005 | 0.0524 ± 0.0005 | 0.0446 ± 0.0003 | 0.0446 ± 0.0003 | 0.0420 ± 0.0003 (5) | 0.0418 ± 0.0003 (5) |
| wide_resnet50_2 | 0.8162 ± 0.0007 | 0.0776 ± 0.0007 | 0.0560 ± 0.0006 | 0.0489 ± 0.0004 | 0.0489 ± 0.0004 | 0.0460 ± 0.0004 (4) | 0.0457 ± 0.0004 (5) |
| efficientnetv2_xl | 0.8556 ± 0.0005 | 0.0697 ± 0.0005 | 0.0577 ± 0.0005 | 0.0371 ± 0.0005 | 0.0368 ± 0.0004 | 0.0341 ± 0.0005 (5) | 0.0336 ± 0.0003 (6) |
| vit_l_16_384 | 0.8709 ± 0.0005 | 0.0264 ± 0.0002 | 0.0265 ± 0.0002 | 0.0264 ± 0.0002 | 0.0273 ± 0.0002 | 0.0264 ± 0.0002 (0) | 0.0269 ± 0.0002 (F) |
| vit_b_16_sam | 0.8022 ± 0.0005 | 0.0488 ± 0.0002 | 0.0488 ± 0.0002 | 0.0487 ± 0.0002 | 0.0498 ± 0.0003 | 0.0487 ± 0.0002 (0) | 0.0489 ± 0.0003 (F) |
| vit_b_32_sam | 0.7371 ± 0.0004 | 0.0762 ± 0.0002 | 0.0760 ± 0.0002 | 0.0759 ± 0.0002 | 0.0785 ± 0.0002 | 0.0759 ± 0.0002 (0) | 0.0763 ± 0.0002 (F) |

Table 15: AUROC (mean ±std) for all models evaluated on ImageNet

| Model | Accuracy[%] | Method | | | | | |
|---|---|---|---|---|---|---|---|
| | | MSP | MSP-TS-NLL | MSP-TS-AURC | LogitsMargin | MSP-pNorm ($p^*$) | MaxLogit-pNorm ($p^*$) |
| alexnet | 0.5654 ± 0.0007 | 0.8487 ± 0.0005 | 0.8477 ± 0.0005 | 0.8489 ± 0.0005 | 0.8188 ± 0.0005 | 0.8489 ± 0.0005 (0) | 0.8394 ± 0.0004 (F) |
| convnext_base | 0.8406 ± 0.0004 | 0.8244 ± 0.0009 | 0.8526 ± 0.0009 | 0.8659 ± 0.0008 | 0.8648 ± 0.0008 | 0.8764 ± 0.0007 (4) | 0.8766 ± 0.0008 (5) |
| convnext_large | 0.8443 ± 0.0005 | 0.8251 ± 0.0011 | 0.8492 ± 0.0011 | 0.8685 ± 0.0015 | 0.8681 ± 0.0008 | 0.8787 ± 0.0008 (5) | 0.8792 ± 0.0008 (5) |
| convnext_small | 0.8363 ± 0.0004 | 0.8253 ± 0.0011 | 0.8548 ± 0.0010 | 0.8681 ± 0.0008 | 0.8669 ± 0.0008 | 0.8782 ± 0.0012 (4) | 0.8790 ± 0.0008 (5) |
| convnext_tiny | 0.8252 ± 0.0006 | 0.8241 ± 0.0009 | 0.8557 ± 0.0009 | 0.8651 ± 0.0016 | 0.8633 ± 0.0009 | 0.8780 ± 0.0009 (4) | 0.8786 ± 0.0008 (6) |
| densenet121 | 0.7442 ± 0.0007 | 0.8611 ± 0.0005 | 0.8603 ± 0.0005 | 0.8612 ± 0.0006 | 0.8464 ± 0.0005 | 0.8613 ± 0.0006 (0) | 0.8574 ± 0.0007 (F) |
| densenet161 | 0.7713 ± 0.0005 | 0.8636 ± 0.0009 | 0.8616 ± 0.0008 | 0.8635 ± 0.0009 | 0.8524 ± 0.0009 | 0.8672 ± 0.0008 (2) | 0.8645 ± 0.0008 (8) |
| densenet169 | 0.7560 ± 0.0006 | 0.8640 ± 0.0009 | 0.8626 ± 0.0007 | 0.8639 ± 0.0009 | 0.8508 ± 0.0008 | 0.8650 ± 0.0010 (2) | 0.8613 ± 0.0012 (F) |
| densenet201 | 0.7689 ± 0.0006 | 0.8630 ± 0.0006 | 0.8618 ± 0.0006 | 0.8632 ± 0.0006 | 0.8512 ± 0.0005 | 0.8647 ± 0.0010 (2) | 0.8620 ± 0.0007 (8) |
| efficientnet_b0 | 0.7771 ± 0.0007 | 0.8568 ± 0.0006 | 0.8642 ± 0.0006 | 0.8649 ± 0.0009 | 0.8538 ± 0.0006 | 0.8705 ± 0.0007 (5) | 0.8701 ± 0.0008 (6) |
| efficientnet_b1 | 0.7983 ± 0.0006 | 0.8535 ± 0.0008 | 0.8667 ± 0.0006 | 0.8664 ± 0.0027 | 0.8578 ± 0.0006 | 0.8708 ± 0.0006 (4) | 0.8709 ± 0.0007 (6) |
| efficientnet_b2 | 0.8060 ± 0.0008 | 0.8511 ± 0.0007 | 0.8627 ± 0.0006 | 0.8642 ± 0.0021 | 0.8591 ± 0.0005 | 0.8740 ± 0.0005 (4) | 0.8740 ± 0.0006 (6) |
| efficientnet_b3 | 0.8202 ± 0.0006 | 0.8424 ± 0.0010 | 0.8624 ± 0.0009 | 0.8673 ± 0.0016 | 0.8650 ± 0.0010 | 0.8753 ± 0.0019 (5) | 0.8763 ± 0.0010 (6) |
| efficientnet_b4 | 0.8338 ± 0.0005 | 0.8222 ± 0.0005 | 0.8603 ± 0.0005 | 0.8694 ± 0.0009 | 0.8675 ± 0.0006 | 0.8756 ± 0.0017 (4) | 0.8773 ± 0.0007 (6) |
| efficientnet_b5 | 0.8344 ± 0.0006 | 0.8538 ± 0.0008 | 0.8681 ± 0.0007 | 0.8721 ± 0.0011 | 0.8688 ± 0.0005 | 0.8804 ± 0.0016 (4) | 0.8812 ± 0.0006 (5) |
| efficientnet_b6 | 0.8400 ± 0.0006 | 0.8573 ± 0.0006 | 0.8712 ± 0.0007 | 0.8735 ± 0.0012 | 0.8702 ± 0.0006 | 0.8799 ± 0.0018 (5) | 0.8806 ± 0.0007 (6) |
| efficientnet_b7 | 0.8413 ± 0.0006 | 0.8500 ± 0.0006 | 0.8672 ± 0.0005 | 0.8747 ± 0.0010 | 0.8723 ± 0.0004 | 0.8808 ± 0.0020 (4) | 0.8819 ± 0.0005 (6) |
| efficientnet_v2_l | 0.8580 ± 0.0004 | 0.8458 ± 0.0012 | 0.8635 ± 0.0009 | 0.8702 ± 0.0011 | 0.8690 ± 0.0009 | 0.8781 ± 0.0015 (5) | 0.8789 ± 0.0010 (6) |
| efficientnet_v2_m | 0.8513 ± 0.0005 | 0.8234 ± 0.0012 | 0.8534 ± 0.0009 | 0.8678 ± 0.0009 | 0.8669 ± 0.0009 | 0.8761 ± 0.0021 (4) | 0.8768 ± 0.0009 (5) |
| efficientnet_v2_s | 0.8424 ± 0.0005 | 0.8575 ± 0.0009 | 0.8704 ± 0.0008 | 0.8718 ± 0.0010 | 0.8695 ± 0.0009 | 0.8802 ± 0.0008 (4) | 0.8802 ± 0.0008 (5) |
| googlenet | 0.6978 ± 0.0006 | 0.8488 ± 0.0008 | 0.8541 ± 0.0008 | 0.8549 ± 0.0007 | 0.8361 ± 0.0006 | 0.8557 ± 0.0010 (3) | 0.8549 ± 0.0013 (6) |
| inception_v3 | 0.7730 ± 0.0006 | 0.8481 ± 0.0006 | 0.8539 ± 0.0006 | 0.8591 ± 0.0015 | 0.8529 ± 0.0005 | 0.8693 ± 0.0005 (4) | 0.8696 ± 0.0005 (5) |
| maxvit_t | 0.8370 ± 0.0006 | 0.8584 ± 0.0006 | 0.8648 ± 0.0008 | 0.8688 ± 0.0009 | 0.8664 ± 0.0008 | 0.8770 ± 0.0007 (4) | 0.8772 ± 0.0008 (5) |
| mnasnet0_5 | 0.6775 ± 0.0006 | 0.8463 ± 0.0006 | 0.8533 ± 0.0005 | 0.8541 ± 0.0006 | 0.8340 ± 0.0007 | 0.8580 ± 0.0005 (4) | 0.8574 ± 0.0006 (7) |
| mnasnet0_75 | 0.7120 ± 0.0008 | 0.8025 ± 0.0005 | 0.8507 ± 0.0006 | 0.8523 ± 0.0005 | 0.8373 ± 0.0005 | 0.8591 ± 0.0005 (3) | 0.8582 ± 0.0006 (6) |
| mnasnet1_0 | 0.7347 ± 0.0005 | 0.8666 ± 0.0005 | 0.8654 ± 0.0006 | 0.8665 ± 0.0005 | 0.8511 ± 0.0004 | 0.8664 ± 0.0007 (0) | 0.8606 ± 0.0004 (F) |
| mnasnet1_3 | 0.7649 ± 0.0005 | 0.7943 ± 0.0007 | 0.8548 ± 0.0006 | 0.8570 ± 0.0024 | 0.8494 ± 0.0007 | 0.8659 ± 0.0006 (4) | 0.8657 ± 0.0005 (6) |
| mobilenet_v2 | 0.7216 ± 0.0008 | 0.8165 ± 0.0007 | 0.8552 ± 0.0007 | 0.8561 ± 0.0006 | 0.8397 ± 0.0007 | 0.8599 ± 0.0007 (4) | 0.8592 ± 0.0007 (6) |
| mobilenet_v3_large | 0.7529 ± 0.0006 | 0.8522 ± 0.0006 | 0.8627 ± 0.0006 | 0.8622 ± 0.0008 | 0.8462 ± 0.0007 | 0.8639 ± 0.0010 (4) | 0.8639 ± 0.0007 (6) |
| mobilenet_v3_small | 0.6769 ± 0.0006 | 0.8623 ± 0.0005 | 0.8613 ± 0.0005 | 0.8623 ± 0.0006 | 0.8419 ± 0.0007 | 0.8623 ± 0.0006 (0) | 0.8547 ± 0.0005 (F) |
| regnet_x_16gf | 0.8273 ± 0.0005 | 0.8541 ± 0.0011 | 0.8655 ± 0.0010 | 0.8680 ± 0.0010 | 0.8666 ± 0.0009 | 0.8767 ± 0.0009 (4) | 0.8772 ± 0.0008 (5) |
| regnet_x_1_6gf | 0.7969 ± 0.0007 | 0.7991 ± 0.0009 | 0.8585 ± 0.0007 | 0.8629 ± 0.0010 | 0.8576 ± 0.0007 | 0.8717 ± 0.0007 (4) | 0.8719 ± 0.0008 (6) |
| regnet_x_32gf | 0.8304 ± 0.0006 | 0.8535 ± 0.0011 | 0.8627 ± 0.0011 | 0.8685 ± 0.0008 | 0.8673 ± 0.0008 | 0.8779 ± 0.0008 (4) | 0.8782 ± 0.0008 (5) |
| regnet_x_3_2gf | 0.8119 ± 0.0006 | 0.8339 ± 0.0012 | 0.8577 ± 0.0009 | 0.8607 ± 0.0013 | 0.8589 ± 0.0010 | 0.8720 ± 0.0009 (5) | 0.8722 ± 0.0010 (5) |

| Model | | | | | | |
|---|---|---|---|---|---|---|
| regnet_x_400mf | 0.7489 ± 0.0008 | 0.8582 ± 0.0007 | 0.8585 ± 0.0010 | 0.8457 ± 0.0008 | 0.8660 ± 0.0007 (4) | 0.8655 ± 0.0006 (5) |
| regnet_x_800mf | 0.7753 ± 0.0007 | 0.8557 ± 0.0006 | 0.8589 ± 0.0010 | 0.8501 ± 0.0005 | 0.8686 ± 0.0006 (4) | 0.8683 ± 0.0005 (5) |
| regnet_x_8gf | 0.8170 ± 0.0006 | 0.8651 ± 0.0008 | 0.8674 ± 0.0010 | 0.8656 ± 0.0008 | 0.8766 ± 0.0008 (5) | 0.8770 ± 0.0008 (5) |
| regnet_y_128gf | 0.8824 ± 0.0003 | 0.8826 ± 0.0006 | 0.8834 ± 0.0014 | 0.8799 ± 0.0006 | 0.8846 ± 0.0009 (0) | 0.8857 ± 0.0005 (8) |
| regnet_y_16gf | 0.8292 ± 0.0005 | 0.8577 ± 0.0011 | 0.8696 ± 0.0009 | 0.8685 ± 0.0008 | 0.8798 ± 0.0008 (4) | 0.8800 ± 0.0007 (5) |
| regnet_y_1_6gf | 0.8090 ± 0.0007 | 0.8612 ± 0.0005 | 0.8632 ± 0.0031 | 0.8584 ± 0.0005 | 0.8735 ± 0.0005 (4) | 0.8735 ± 0.0005 (5) |
| regnet_y_32gf | 0.8339 ± 0.0005 | 0.8631 ± 0.0009 | 0.8699 ± 0.0011 | 0.8675 ± 0.0007 | 0.8798 ± 0.0006 (4) | 0.8801 ± 0.0007 (5) |
| regnet_y_3_2gf | 0.8198 ± 0.0006 | 0.8648 ± 0.0007 | 0.8655 ± 0.0021 | 0.8609 ± 0.0007 | 0.8732 ± 0.0006 (6) | 0.8732 ± 0.0007 (5) |
| regnet_y_400mf | 0.7581 ± 0.0006 | 0.8574 ± 0.0006 | 0.8569 ± 0.0008 | 0.8442 ± 0.0006 | 0.8663 ± 0.0006 (4) | 0.8657 ± 0.0006 (5) |
| regnet_y_800mf | 0.7885 ± 0.0007 | 0.8614 ± 0.0006 | 0.8626 ± 0.0006 | 0.8524 ± 0.0006 | 0.8715 ± 0.0007 (4) | 0.8711 ± 0.0006 (5) |
| regnet_y_8gf | 0.8283 ± 0.0006 | 0.8668 ± 0.0012 | 0.8689 ± 0.0012 | 0.8673 ± 0.0011 | 0.8781 ± 0.0010 (4) | 0.8785 ± 0.0010 (5) |
| resnet101 | 0.8188 ± 0.0005 | 0.8602 ± 0.0010 | 0.8655 ± 0.0015 | 0.8633 ± 0.0009 | 0.8747 ± 0.0018 (4) | 0.8757 ± 0.0010 (5) |
| resnet152 | 0.8230 ± 0.0007 | 0.8649 ± 0.0008 | 0.8708 ± 0.0009 | 0.8687 ± 0.0007 | 0.8789 ± 0.0017 (4) | 0.8802 ± 0.0007 (5) |
| resnet18 | 0.6976 ± 0.0006 | 0.8565 ± 0.0003 | 0.8578 ± 0.0003 | 0.8403 ± 0.0003 | 0.8575 ± 0.0004 (1) | 0.8520 ± 0.0005 (F) |
| resnet34 | 0.7331 ± 0.0007 | 0.8610 ± 0.0006 | 0.8617 ± 0.0005 | 0.8456 ± 0.0005 | 0.8619 ± 0.0006 (2) | 0.8587 ± 0.0007 (F) |
| resnet50 | 0.8084 ± 0.0006 | 0.8601 ± 0.0009 | 0.8644 ± 0.0014 | 0.8612 ± 0.0007 | 0.8734 ± 0.0019 (4) | 0.8743 ± 0.0007 (5) |
| resnext101_32x8d | 0.8283 ± 0.0006 | 0.8452 ± 0.0012 | 0.8646 ± 0.0007 | 0.8637 ± 0.0006 | 0.8768 ± 0.0005 (4) | 0.8774 ± 0.0005 (5) |
| resnext101_64x4d | 0.8325 ± 0.0005 | 0.8515 ± 0.0013 | 0.8680 ± 0.0010 | 0.8672 ± 0.0009 | 0.8778 ± 0.0009 (4) | 0.8788 ± 0.0008 (5) |
| resnext50_32x4d | 0.8119 ± 0.0007 | 0.8566 ± 0.0009 | 0.8631 ± 0.0017 | 0.8603 ± 0.0008 | 0.8731 ± 0.0009 (4) | 0.8738 ± 0.0008 (5) |
| shufflenet_v2_x0_5 | 0.6058 ± 0.0005 | 0.8497 ± 0.0006 | 0.8520 ± 0.0005 | 0.8322 ± 0.0006 | 0.8535 ± 0.0008 (4) | 0.8528 ± 0.0006 (7) |
| shufflenet_v2_x1_0 | 0.6936 ± 0.0008 | 0.8575 ± 0.0005 | 0.8598 ± 0.0005 | 0.8464 ± 0.0005 | 0.8623 ± 0.0004 (4) | 0.8621 ± 0.0005 (7) |
| shufflenet_v2_x1_5 | 0.7303 ± 0.0007 | 0.8521 ± 0.0008 | 0.8537 ± 0.0008 | 0.8393 ± 0.0007 | 0.8594 ± 0.0007 (4) | 0.8589 ± 0.0007 (6) |
| shufflenet_v2_x2_0 | 0.7621 ± 0.0007 | 0.8592 ± 0.0007 | 0.8613 ± 0.0026 | 0.8525 ± 0.0006 | 0.8696 ± 0.0006 (5) | 0.8696 ± 0.0007 (6) |
| squeezenet1_0 | 0.5810 ± 0.0005 | 0.8410 ± 0.0004 | 0.8436 ± 0.0003 | 0.8180 ± 0.0004 | 0.8436 ± 0.0003 (0) | 0.8178 ± 0.0005 (F) |
| squeezenet1_1 | 0.5820 ± 0.0005 | 0.8481 ± 0.0003 | 0.8498 ± 0.0003 | 0.8234 ± 0.0005 | 0.8498 ± 0.0003 (0) | 0.8263 ± 0.0006 (F) |
| swin_b | 0.8358 ± 0.0006 | 0.8544 ± 0.0012 | 0.8668 ± 0.0008 | 0.8652 ± 0.0008 | 0.8771 ± 0.0015 (4) | 0.8785 ± 0.0007 (5) |
| swin_s | 0.8321 ± 0.0005 | 0.8613 ± 0.0007 | 0.8657 ± 0.0011 | 0.8629 ± 0.0007 | 0.8748 ± 0.0007 (4) | 0.8755 ± 0.0007 (5) |
| swin_t | 0.8147 ± 0.0005 | 0.8583 ± 0.0008 | 0.8664 ± 0.0017 | 0.8606 ± 0.0008 | 0.8744 ± 0.0008 (4) | 0.8747 ± 0.0007 (5) |
| swin_v2_b | 0.8415 ± 0.0005 | 0.8514 ± 0.0010 | 0.8666 ± 0.0007 | 0.8646 ± 0.0008 | 0.8761 ± 0.0007 (4) | 0.8764 ± 0.0008 (5) |
| swin_v2_s | 0.8372 ± 0.0004 | 0.8589 ± 0.0006 | 0.8724 ± 0.0009 | 0.8700 ± 0.0006 | 0.8802 ± 0.0006 (4) | 0.8809 ± 0.0008 (5) |
| swin_v2_t | 0.8208 ± 0.0005 | 0.8593 ± 0.0006 | 0.8689 ± 0.0017 | 0.8645 ± 0.0006 | 0.8766 ± 0.0007 (4) | 0.8771 ± 0.0006 (5) |
| vgg11 | 0.6905 ± 0.0006 | 0.8616 ± 0.0007 | 0.8619 ± 0.0007 | 0.8419 ± 0.0007 | 0.8619 ± 0.0007 (0) | 0.8484 ± 0.0008 (F) |
| vgg11_bn | 0.7037 ± 0.0007 | 0.8630 ± 0.0004 | 0.8632 ± 0.0004 | 0.8447 ± 0.0005 | 0.8632 ± 0.0004 (0) | 0.8495 ± 0.0004 (F) |
| vgg13 | 0.6995 ± 0.0005 | 0.8622 ± 0.0005 | 0.8624 ± 0.0006 | 0.8438 ± 0.0006 | 0.8624 ± 0.0006 (0) | 0.8503 ± 0.0008 (F) |
| vgg13_bn | 0.7160 ± 0.0006 | 0.8628 ± 0.0005 | 0.8629 ± 0.0005 | 0.8447 ± 0.0006 | 0.8629 ± 0.0005 (0) | 0.8512 ± 0.0006 (F) |
| vgg16 | 0.7161 ± 0.0007 | 0.8660 ± 0.0004 | 0.8661 ± 0.0004 | 0.8476 ± 0.0003 | 0.8661 ± 0.0004 (0) | 0.8552 ± 0.0007 (F) |
| vgg16_bn | 0.7339 ± 0.0006 | 0.8674 ± 0.0003 | 0.8674 ± 0.0003 | 0.8518 ± 0.0005 | 0.8674 ± 0.0003 (0) | 0.8587 ± 0.0005 (F) |
| vgg19 | 0.7238 ± 0.0005 | 0.8657 ± 0.0005 | 0.8657 ± 0.0005 | 0.8478 ± 0.0005 | 0.8654 ± 0.0011 (0) | 0.8562 ± 0.0005 (F) |
| vgg19_bn | 0.7424 ± 0.0006 | 0.8651 ± 0.0008 | 0.8651 ± 0.0008 | 0.8503 ± 0.0008 | 0.8651 ± 0.0008 (0) | 0.8557 ± 0.0008 (F) |
| vit_b_16 | 0.8108 ± 0.0006 | 0.8560 ± 0.0005 | 0.8665 ± 0.0018 | 0.8620 ± 0.0005 | 0.8755 ± 0.0005 (4) | 0.8762 ± 0.0005 (5) |

| | | | | | | |
|---|---|---|---|---|---|---|
| vit_b_32 | $0.7596 \pm 0.0004$ | $0.8559 \pm 0.0006$ | $0.8632 \pm 0.0006$ | $0.8610 \pm 0.0023$ | $0.8553 \pm 0.0008$ | $0.8739 \pm 0.0007$ (4) | $0.8741 \pm 0.0006$ (5) |
| vit_h_14 | $0.8855 \pm 0.0005$ | $0.8745 \pm 0.0009$ | $0.8768 \pm 0.0009$ | $0.8807 \pm 0.0007$ | $0.8782 \pm 0.0007$ | $0.8839 \pm 0.0014$ (4) | $0.8843 \pm 0.0007$ (6) |
| vit_l_16 | $0.7966 \pm 0.0007$ | $0.8590 \pm 0.0007$ | $0.8620 \pm 0.0006$ | $0.8641 \pm 0.0009$ | $0.8604 \pm 0.0005$ | $0.8767 \pm 0.0003$ (4) | $0.8769 \pm 0.0005$ (4) |
| vit_l_32 | $0.7699 \pm 0.0007$ | $0.8542 \pm 0.0004$ | $0.8593 \pm 0.0004$ | $0.8605 \pm 0.0018$ | $0.8562 \pm 0.0004$ | $0.8758 \pm 0.0002$ (4) | $0.8755 \pm 0.0003$ (4) |
| wide_resnet101_2 | $0.8252 \pm 0.0006$ | $0.8371 \pm 0.0009$ | $0.8580 \pm 0.0010$ | $0.8669 \pm 0.0010$ | $0.8658 \pm 0.0007$ | $0.8780 \pm 0.0008$ (5) | $0.8785 \pm 0.0006$ (5) |
| wide_resnet50_2 | $0.8162 \pm 0.0007$ | $0.7915 \pm 0.0014$ | $0.8520 \pm 0.0014$ | $0.8631 \pm 0.0014$ | $0.8606 \pm 0.0010$ | $0.8741 \pm 0.0010$ (4) | $0.8748 \pm 0.0009$ (5) |
| efficientnetv2_xl | $0.8556 \pm 0.0005$ | $0.7732 \pm 0.0014$ | $0.8107 \pm 0.0016$ | $0.8606 \pm 0.0011$ | $0.8604 \pm 0.0011$ | $0.8712 \pm 0.0012$ (5) | $0.8740 \pm 0.0010$ (6) |
| vit_l_16_384 | $0.8709 \pm 0.0005$ | $0.8851 \pm 0.0007$ | $0.8850 \pm 0.0006$ | $0.8855 \pm 0.0006$ | $0.8793 \pm 0.0007$ | $0.8855 \pm 0.0006$ (0) | $0.8835 \pm 0.0007$ (F) |
| vit_b_16_sam | $0.8022 \pm 0.0005$ | $0.8817 \pm 0.0007$ | $0.8819 \pm 0.0007$ | $0.8822 \pm 0.0007$ | $0.8760 \pm 0.0008$ | $0.8822 \pm 0.0007$ (0) | $0.8812 \pm 0.0012$ (F) |
| vit_b_32_sam | $0.7371 \pm 0.0004$ | $0.8752 \pm 0.0005$ | $0.8755 \pm 0.0005$ | $0.8759 \pm 0.0005$ | $0.8655 \pm 0.0005$ | $0.8759 \pm 0.0005$ (0) | $0.8751 \pm 0.0007$ (F) |