# OpenReview forum: "How to Fix a Broken Confidence Estimator: Evaluating Post-hoc Methods for Selective Classification with Deep Neural Networks"
_auai.org/UAI/2024/Conference — UAI 2024 poster_

### Official Review · Reviewer_YMk7 · 2024-03-10

**Q2-1 Originality-Novelty:** 3
**Q2-2 Correctness-Technical Quality:** 3
**Q2-5 Clarity Of Writing:** 3

**Q1 Summary And Contributions:**

The paper targets the problem of selective classification where a model should be able to output a decision to abstain from making a prediction when it is uncertain. The paper focuses on post-hoc calibration of abstention. They perform an extensive empirical evaluation on a variety of datasets and models, benchmarking the different methods for abstention calibration. The propose a simple method that outperforms the state-of-the-art methods in terms of and abstention accuracy and a new metric fixing previous shortcomings. The paper is sound and well-written, and the results are convincing. There need to be some formatting and clarity improvements, and a further investigation of the theoretical justification/intuition behind the proposed method would be beneficial.

**Q2-3 Extent To Which Claims Are Supported By Evidence:**

4: Excellent: all claims are supported by very convincing evidence (in the form of comprehensive experimental evaluation, rigorous mathematical proofs, detailed (pseudo-)code, precise references, well-motivated and realistic assumptions) and the authors deliver what they promise.

**Q2-4 Reproducibility:**

3: Good: key resources (e.g. proofs, code, data) are available and key details (e.g. proofs, experimental setup) are sufficiently well-described for competent researchers to confidently reproduce the main results.

**Q3 Main Strengths:**

- The paper is solving an important problem in confidence calibration, which is relevant to the audience of UAI, and it is clearly written.
- Extensive empirical evaluation on a variety of datasets (in-distribution and out-of-distribution) and models, supporting the claims of the paper.
- Clear overview of the state-of-the-art methods for confidence scoring.
- Proposal of NAURC which fixes the existing methods' evaluation metrics.

**Q4 Main Weakness:**

- It would be useful to also report some variation of expected calibration error (ECE) to compare how the accuracy is matched with the confidence.
- The reviewer understands that the study is empirical, but what is the theoretical justification for the proposed normalization method - or some sort of intuition behind it? MSP is a very intuitive method which makes it likeable in practice, so for a wider adoption, it would be beneficial to have a more thorough explanation of the normalization method. This could be partially addressed by moving the points from Appendix C to the main text.
- Submission of code and data would be beneficial to check for reproducibility.

**Q5 Detailed Comments To The Authors:**

- Ding et al. [2020] in the paragraph for AUROC/AURC should use a different citation style.
- The formatting of the tables could be neater e.g. sometimes you use \tiny for the standard deviation and sometimes there seems to be the padding of zeros.
- Some of the references have a broken URL link in terms of formatting.
- Some of the tables have a bold, winning value and some do not. It would be beneficial to have a consistent style.

**Q9 Complying With Reviewing Instructions:**

Yes

---

> ### Author Rebuttal · Authors · 2024-04-08
>
> We thank the reviewer for the positive comments and the suggestions. For the camera ready version, we will be adding a section in the Appendix reporting the value of ECE for all the methods where it is relevant (where g(x) can be considered as a probability). Moreover, we will be releasing a link for our Github repository, where all the experiments and codes can be found.
>
> About the theoretical insights about normalization: The key idea is that we notice that all the methods that achieve gains are mathematical methods for considering mainly the greatest logits in the evaluation of g(z(x)). This is explicitly in the LogitsMargin method, and in the Appendix C we show some insights for why the temperature scaling and the p-normalization are also making this. Hence, the selective classification performance anomaly detected in some models is (in our hypothesis) a result of these models having its uncertainty information mainly in the largest logits. The reason for why this happens (as well as the reason for why the MSP can be ‘’broken’’ for some models) is being currently investigated and stays as a suggestion for future works. We will include a Discussion section containing further discussions about what is presented in Appendix C and in this answer, bringing hence theoretical insights and the novelties of the paper to the main body.

---

### Official Review · Reviewer_z5Rq · 2024-03-10

**Q2-1 Originality-Novelty:** 2
**Q2-2 Correctness-Technical Quality:** 3
**Q2-5 Clarity Of Writing:** 3

**Q10 Ethical Concerns:**

No ethical concerns.

**Q1 Summary And Contributions:**

The paper presents a way to improve selective classification, by considering post hoc approaches, such as the normalization of last-layer logits. Experimental evaluation shows the effectiveness of such methods whenever the neural network’s confidence is not well estimated.

**Q2-3 Extent To Which Claims Are Supported By Evidence:**

3: Good: the main claims are supported by convincing evidence (in the form of adequate experimental evaluation, proofs, (pseudo-)code, references, assumptions).

**Q2-4 Reproducibility:**

2: Fair: key resources (e.g. proofs, code, data) are unavailable but key details (e.g. proof sketches, experimental setup) are sufficiently well-described for an expert to confidently reproduce the main results.

**Q3 Main Strengths:**

1. Clarity of the paper. The paper is very well organized and the contributions are well framed.

2. Simple yet effective idea. The authors propose the usage of a logit normalization to fix the confidence estimators of different methods. This turns out to be an effective way to improve many of the classifiers.

3. A novel “fair” metric. The authors introduce a novel metric (NAURC) to compare different selective classifiers. This metric solves some of the problems of previous metrics such as E-AURC.

**Q4 Main Weakness:**

The work’s main weaknesses are:

1. Limited theoretical grounds: the work proposes a heuristic approach to fix underconfidence of a neural network estimator, without addressing from a theoretical perspective why this should be actively improving the confidence.

2. Only image data: the work is evaluated only using three image datasets. Although this is quite common in the recent literature on selective classification and the authors provide a well thought set of experiments, I argue that images are not the only kind of data where selective classification might be useful. Including other kinds of data, e.g. tabular, might make the contribution more solid.

3. Missing code: despite quite a few details that allow for some reproducibility, the lack of the code used for experiments is a weakness.

**Q5 Detailed Comments To The Authors:**

The paper is well organized and the core idea is easy to grasp. However, I have a few concerns/questions

1. Regarding the evaluation, you focused mostly on the NAURC/AURC. However, I think the main goal of selective classification (as framed by Geifman et al. 2017, 2019) is either to minimize a certain risk given a target coverage $c$ or maximize coverage given a target risk $e$ (see also [1]). I wonder if you could provide more results in this sense as well, as done for instance in some works such as [2, 3, 4]. For instance, this allows one to understand which is the best methodology in terms of risk (coverage) at a specific target coverage (risk) level, as your postprocessing approach might affect performance differently at different coverage (risk) levels.

2. Moreover, it is not clear whether the improvements derived from applying logit normalization are statistically significant. I would suggest the authors to consider the approach by [5] to address this point (a useful Python package implementing it is autorank [6]).

3. As expressed above, the evaluation of the approach is only provided for image data. This might hinder the generalization of your results. For evaluating the approach on other set of data, e.g. tabular,  I would suggest the authors the resources of [7], a recent paper that benchmarks many of the existing selective classifier approaches for deep neural networks.


[1] Franc, Vojtech, Daniel Prusa, and Vaclav Voracek. "Optimal strategies for reject option classifiers." Journal of Machine Learning Research 24, no. 11 (2023): 1-49.
[2] Fisch, Adam, Tommi Jaakkola, and Regina Barzilay. "Calibrated selective classification." arXiv preprint arXiv:2208.12084 (2022).
[3] Gangrade, Aditya, Anil Kag, and Venkatesh Saligrama. "Selective classification via one-sided prediction." In International Conference on Artificial Intelligence and Statistics, pp. 2179-2187. PMLR, 2021.
[4] Pugnana, Andrea, and Salvatore Ruggieri. "AUC-based Selective Classification." In International Conference on Artificial Intelligence and Statistics, pp. 2494-2514. PMLR, 2023.
[5] Demšar, Janez. "Statistical comparisons of classifiers over multiple data sets." The Journal of Machine learning research 7 (2006): 1-30.
[6] Herbold, Steffen. "Autorank: A python package for automated ranking of classifiers." Journal of Open Source Software 5, no. 48 (2020): 2173.
[7] Pugnana, Andrea, Lorenzo Perini, Jesse Davis, and Salvatore Ruggieri. "Deep Neural Network Benchmarks for Selective Classification." arXiv preprint arXiv:2401.12708 (2024).

**Q9 Complying With Reviewing Instructions:**

Yes

---

> ### Author Rebuttal · Authors · 2024-04-08
>
> We thank the reviewer for the positive comments and the suggestions. We will be releasing a public github repository with all codes and experiments available.
>
> > **Q1** (Other performance metrics)
>
> We took this into consideration when we presented the results (see figure 3c) for the SAC metric (the coverage needed for achieving some desired risk). For being mostly used for critical applications, SC usually will be used to achieve low risk guarantees; hence, for calculating the SAC, we chose an accuracy of 98% as the constraint. Moreover, as being a considerably high accuracy for all models in ImageNet (that usually have the total accuracy around 80%), achieving great gains for it indicates probably a high gain for lower accuracies, as the RC curve being continuous. Furthermore, the AURC can be interpreted as the average of the SAC for all coverages/risks.
>
> > **Q2** (Statistical significance)
>
> For all experiments comparing different methods, we applied different random experiments, reporting the mean and standard deviation. Moreover, we considered 84 ImageNet models as well as some extra models for other datasets, and the results were consistent for all of them.
> Nevertheless, even if the results regarding the gains of MaxLogit-pNorm over other considered methods can be small, we must emphasize that the main goal of this work was not to develop a SOTA method for selective classification, but to show that simple methods can fix some pathologies previously observed in some image classifiers. Indeed, MaxLogit-pNorm was the best method for achieving the highest results as also being extremely data efficient - however, if the results pointed out that some other method is the best, all the other conclusions and results of the paper would maintain, as the gain of all these methods over the baseline are very significant and fix the SC anomalies.
>
> > **W2/Q3** (Only image data)
>
> Indeed, in this work we focused solely on image classifiers. The reason for that was the previously reported anomalies of SC performance for pretrained models on ImageNet. Galil et al. [2023] showed that different models achieve different SC performance; in our work, we show that, actually, after optimization, all models perform well, with the SC metrics well correlated with the accuracy. However, a considerable number of models can’t be optimized, and these models have the MSP already as an excellent uncertainty estimator, indicating thus that the other models suffer from some anomaly. Hence, achieving good post-hoc optimization results demand us to identify such an anomaly in some models in some specific datasets, which may require us to consider a potentially large number of models for each dataset. But even this is still not guaranteed to lead to significant gains, since the cause of these anomalies may be related to the way some image classifiers are trained, i.e., it is possible that the anomaly manifests itself particularly in image classifiers. Understanding the cause of these anomalies and whether they affect other data modalities is a topic we are certainly interested in investigating. In summary, we thank the reviewer for the suggestion and will consider it in our future research.
>
>  > **W1** (Limited theoretical grounds)
>
> Chow [1970] show that, if MSP represents the true posterior P(y|x), then it must be the optimal way to estimate uncertainty. Furthermore, the general maximum likelihood estimation (MLE) used for training classifiers has a strong theoretical background for obtaining the posterior. However, modern deep learning classifiers have been trained with approaches focusing on the accuracy maximization instead of solely MLE. This might reflect in overfitting in the likelihood (as pointed by [Guo 2017] as a reason for miscalibration) and in the use of methods that do not focus on the posterior estimation itself, but modify the loss in order to maximize accuracy. These reasons might explain why only some models have these bad selective classification performance.
> Theoretical insights about normalization: The key idea is that we notice that all the methods that achieve gains are mathematical methods for considering mainly the greatest logits in the evaluation of g(z(x)). This is explicitly in the LogitsMargin method, and in the Appendix C we show some insights for why the temperature scaling and the p-normalization are also making this. Hence, the selective classification performance anomaly detected in some models is (in our hypothesis) a result of these models having its uncertainty information mainly in the largests logits. The reason for why this happens (as well as the reason for why the MSP can be ‘’broken’’ for some models) is being currently investigated and stays as a suggestion for future works. Moreover, we will be including a Discussion section containing further discussions about what is presented in Appendix C and in this answer, bringing hence theoretical insights and the novelties of the paper to the main body.

---

### Official Review · Reviewer_rmaS · 2024-03-23

**Q2-1 Originality-Novelty:** 3
**Q2-2 Correctness-Technical Quality:** 3
**Q2-5 Clarity Of Writing:** 4

**Q1 Summary And Contributions:**

The paper deals with the setting of a deep neural network making selective classification decisions based on estimated confidence so as to minimise errors on the decisions that are made. It is motivated by an empirical observation in prior work that the confidence estimators of many models are  "broken", giving rise to a selective classification performance worse than what may be expected from their overall accuracy. The paper extensively analyses several post-hoc methods of confidence estimation that use the classifier logits (thus avoiding re-training), both those proposed before and new ones, on multiple datasets and a large number of pre-trained models. The authors also propose an improved metric to have a fair evaluation of the selective classification performance independent of the classifier accuracy. From the extensive empirical analysis, it can be concluded that the proposed p-norm normalisation of the logits, followed by selecting the max logit, is able to significantly improve the selective classification performance for the majority of models studied, and for the others a fallback to the previous confidence estimation method (maximum softmax probability) can be made. In addition, the selective classification performance then achieves a correlation of nearly one with the overall accuracy. The proposed method is also data efficient and robust to distribution shift.

**Q2-3 Extent To Which Claims Are Supported By Evidence:**

4: Excellent: all claims are supported by very convincing evidence (in the form of comprehensive experimental evaluation, rigorous mathematical proofs, detailed (pseudo-)code, precise references, well-motivated and realistic assumptions) and the authors deliver what they promise.

**Q2-4 Reproducibility:**

2: Fair: key resources (e.g. proofs, code, data) are unavailable but key details (e.g. proof sketches, experimental setup) are sufficiently well-described for an expert to confidently reproduce the main results.

**Q3 Main Strengths:**

1. The method proposed by the paper is quite simple and efficient, and yet achieves considerable gains in selective classification performance as evident from the thorough and extensive empirical evaluation.
2. The methodology adopted in the paper appears to be technically sound, including the choice of metrics, comparison to adequate baselines, hyper parameter tuning, etc.
3. The claims are strongly supported by the experimental evidence. In particular, the curves showing the NAURC gains and the almost perfect correlation of AURC and accuracy after optimisation are quite convincing.
4. The paper is very well written and was a pleasure to read. Experiments are described in sufficient detail and design choices and assumptions are well motivated.

**Q4 Main Weakness:**

1. More of the related work should be presented in the main body of the paper rather than being almost entirely relegated to the appendix. Otherwise it is hard to understand the novelty of the work.

**Q5 Detailed Comments To The Authors:**

1.More of the related work should be presented in the main body of the paper rather than being almost entirely relegated to the appendix. Otherwise it is hard to understand the novelty of the work. I understand that space is tight, but you may move some less important results to the appendix to make space.
2. It would be nice to clarify if you will be releasing the code.

**Q9 Complying With Reviewing Instructions:**

Yes

---

> ### Author Rebuttal · Authors · 2024-04-08
>
> We thank the reviewer for the positive comments and the suggestions. All the codes and experiments will be released in a public GitHub repository for the camera ready version, and we moved the related work appendix to the main paper, as suggested. Additionally, we will also be including a Discussion section containing further discussions about what is presented in Appendix C, bringing hence theoretical insights and the novelties of the paper to the main body.

---

### Official Review · Reviewer_UL9v · 2024-03-24

**Q2-1 Originality-Novelty:** 2
**Q2-2 Correctness-Technical Quality:** 3
**Q2-5 Clarity Of Writing:** 2

**Q1 Summary And Contributions:**

In the context of selective classification, this paper benchmarks several post-hoc methods that aim to improve uncertainty quantification in pretrained models (in this case neural networks), where the uncertainty can be represented by different quantities that depend on the last layer's softmax score.
The methods being compared are variants of temperature scaling and p-norm logit normalization, and the authors use a variant of the AURC to compare these calibration methods.

**Q2-3 Extent To Which Claims Are Supported By Evidence:**

2: Fair: the main claims are somewhat supported by evidence (but the experimental evaluation may be weak, or does not match entirely with the claims, important baselines may be missing, proofs contain important ideas but lack rigor, algorithmic details are only discussed superficially, references are imprecise, assumptions are not sufficiently motivated or explicated, etc.).

**Q2-4 Reproducibility:**

2: Fair: key resources (e.g. proofs, code, data) are unavailable but key details (e.g. proof sketches, experimental setup) are sufficiently well-described for an expert to confidently reproduce the main results.

**Q3 Main Strengths:**

The experimental methodology is exhaustively described and the point of the authors is well supported by the data. Moreover, the main conclusion, that logit normalisation is preferable to temperature scaling, is interesting (and not evident).

**Q4 Main Weakness:**

The authors only considered one metric, and while I understand that their focus is selective classification, it would have been interesting to consider other metrics (expected calibration error, brier score, etc.).

**Q5 Detailed Comments To The Authors:**

- The paper is missing links to the code (or as a supplementary material)
- the content of appendix C is interesting and could be summarized a bit more in the main part (it's just mentioned once) so that we can understand the method and the difference with TS and Wei et al. better

**Q9 Complying With Reviewing Instructions:**

Yes

---

> ### Author Rebuttal · Authors · 2024-04-08
>
> We thank the reviewer for the positive comments and the suggestions.
>
> Firstly, we emphasize that we considered four selective classification metrics: AURC, SAC and NAURC, which are directly correlated, and AUROC, which has no direct relation to the other three. Moreover, we will add in the Appendix a section reporting the value of ECE for all the methods where it is relevant (where g(x) can be considered as a probability).
>
> Both of the additional comments/suggestions will be addressed for the camera ready version:  all the codes and experiments will be released in a public GitHub repository, and, we will be including a Discussion section containing further discussions about what is presented in Appendix C, bringing hence theoretical insights and the novelties of the paper to the main body.

---

### Official Review · Reviewer_LcgL · 2024-03-26

**Q2-1 Originality-Novelty:** 3
**Q2-2 Correctness-Technical Quality:** 3
**Q2-5 Clarity Of Writing:** 2

**Q1 Summary And Contributions:**

The paper is an empirical study comparing different approaches to post training calibration. The authors show that calibrated models exhibit better correlation among the targeted metrics.

**Q2-3 Extent To Which Claims Are Supported By Evidence:**

3: Good: the main claims are supported by convincing evidence (in the form of adequate experimental evaluation, proofs, (pseudo-)code, references, assumptions).

**Q2-4 Reproducibility:**

2: Fair: key resources (e.g. proofs, code, data) are unavailable but key details (e.g. proof sketches, experimental setup) are sufficiently well-described for an expert to confidently reproduce the main results.

**Q3 Main Strengths:**

This work provides an interesting comparison of calibrating methods using publicly available models.

I think the contributions are nicely summarised by the authors on page 2.

The empirical evaluation is extensive.

**Q4 Main Weakness:**

There’s no justification why the authors observe this behaviour. I didn’t expect any exact theoretical arguments but I was thinking about some arguments expressed in the mathematical notation. I left questions in the detailed comments section.

**Q5 Detailed Comments To The Authors:**

Are the authors able to reproduce this behaviour on a toy dataset? This would make the paper stronger.

Will authors open source the code? It is crucial for such empirical studies.

Can authors provide guidelines where post training calibration can hurt the performance?

Are these methods needed when ensembling? One model is rarely used in practice.

Minor: It would be also good to compare the methods to some statistical technique, such as MLE error bars using approximate Fisher on a toy experiment.

**Q9 Complying With Reviewing Instructions:**

Yes

---

> ### Author Rebuttal · Authors · 2024-04-08
>
> We thank the reviewer for the comments and suggestions. First of all, we will be releasing a link to a Github repository with all codes and experiments in the camera ready version.
>
> > Can authors provide guidelines where post training calibration can hurt the performance?
>
> Post-hoc methods seeking calibration (in the sense presented by [Guo et al. 2017], where the MSP must return a ‘correct’ probability) were previously shown [Galil et al. 2023] to sometimes hurt selective classification performance (see also [Zhu et al. 2022]). When the post-hoc methods are focused directly on selective classification performance, such as the ones presented in our work, what we have shown is that, for some models, all the tuning methods achieve better performance than the MSP, while for some other models all the methods are worse or equal to the MSP. Previous theoretical works show that, if the MSP can represent the true posterior P(y|x), then it must be optimal for selective classification [Chow 1970]. Hence, our hypothesis is that, for these ‘improvable’ models, there is some pathology in the MSP for representing the true posterior, probably caused by some modern training recipe that focus mostly in accuracy and not in achieving the best posterior ([Guo et al. 2017] arguments that the overfitting in the NLL loss might lead to bad uncertainty estimation). Indeed, In Appendix I we show that this is correlated with underconfidence. Nevertheless, the problem of our methods hurting the performance is easily solved when we allow the MSP fallback: when the method is harmful, this seems to indicate that the MSP is already great. Thus, in these cases the model might choose to use the MSP. Lastly, one reason that a method might hurt performance is overfitting in the hold-out set. This problem is addressed in our data efficiency experiment.
>
> > There’s no justification why the authors observe this behaviour. I didn’t expect any exact theoretical arguments but I was thinking about some arguments expressed in the mathematical notation
>
> As discussed in the last answer, the anomaly behavior of the MSP might be caused by some training methods that do not focus on estimating the posterior. About why our methods can heal this problem, some insights in mathematical notation are given in Appendix C (please see also W1 in our response to Reviewer z5Rq). We will include a Discussion section expanding on these explanations.
>
> > Are these methods needed when ensembling? One model is rarely used in practice.
>
> While ensembling is a popular technique for uncertainty quantification, previous works have shown that some popular ensemble methods might not be that useful for selective classification [Cattelan and Silva 2022, Galil et al. 2023]. Furthermore, applying post-hoc methods together with ensembles have shown to be positive when seeking calibration [Rahaman et al., NeurIPS 2021]. Hence, we believe that our methods might still be useful when dealing with ensembles. Unfortunately, performing experiments with ensembles require the access to multiple variations of the same model, and as dealing mostly with pretrained models we have no access to it. Training and analyzing ensemble models with post-hoc methods for selective classification demands time and remains as a suggestion for future works. We thank the reviewer for the suggestion.
>
> > It would be also good to compare the methods to some statistical technique, such as MLE error bars using approximate Fisher on a toy experiment.
>
> If the reviewer considers small datasets as toy datasets, we performed some extra experiments in Cifar100 and OxfordPets in Appendix E, obtaining positive results. If the author is referring to some simulated data for theoretical analysis solely, the results might not necessarily be similar to the ones observed in our Image datasets, since we are mainly reporting an anomaly in these datasets and presenting the possibility to fix it. As discussed in the previous answers, the MSP should be the best uncertainty quantifier, but something is making it not optimal (or even good) for a lot of models. Hence, replicating this results in some simulated dataset might require us to understand the origin of the anomaly (since we need these pathologies so we can fix them). Theoretical results and justifications about these anomalies and our methods are currently being investigated and might result in future works, which will probably have experiments in toy datasets for analyzing hypotheses. In conclusion, we thank the reviewer for the suggestion and will certainly address it in our future work, but it would not be feasible for us to address it in this paper.

---

### Meta-Review · Area_Chair_FAeU · 2024-04-16

This paper provides an empirical study of selective classification via post-hoc abstention.  The reviews unanimously are in favor of acceptance (2 x borderline accept, 2 x accept, 1 x strong accept), finding the paper to be experimentally through and to address an important problem (confidence estimation / confidence-based rejection in NNs, which can improve safety).  The primary criticisms were that other metrics could have been examined, such as ECE, that only image data sets are considered, and that no explanation is offered as to why p-norm normalization is so effective.  I believe that the authors have addresses these concerns (or have made good faith promises to do so in the camera-ready version).

AC Note: please fix the formatting of some references so that the text does not span the two columns.